# Elucidating the origin of chiroptical activity in chiral 2D perovskites through nano-confined growth

Sunihl Ma [1], Young-Kwang Jung [1], Jihoon Ahn[1], Jihoon Kyhm [2], Jeiwan Tan[1], Hyungsoo Lee [1], Gyumin Jang[1], Chan Uk Lee[1], Aron Walsh [1,3] & Jooho Moon [1✉]

Chiral perovskites are being extensively studied as a promising candidate for spintronic- and polarization-based optoelectronic devices due to their interesting spin-polarization properties. However, the origin of chiroptical activity in chiral perovskites is still unknown, as the chirality transfer mechanism has been rarely explored. Here, through the nano-confined growth of chiral perovskites ($MBA_2PbI_{4(1-x)}Br_{4x}$), we verified that the asymmetric hydrogen-bonding interaction between chiral molecular spacers and the inorganic framework plays a key role in promoting the chiroptical activity of chiral perovskites. Based on this understanding, we observed remarkable asymmetry behavior (absorption dissymmetry of $2.0 \times 10^{-3}$ and anisotropy factor of photoluminescence of $6.4 \times 10^{-2}$ for left- and right-handed circularly polarized light) in nanoconfined chiral perovskites even at room temperature. Our findings suggest that electronic interactions between building blocks should be considered when interpreting the chirality transfer phenomena and designing hybrid materials for future spintronic and polarization-based devices.

[1] Department of Materials Science and Engineering, Yonsei University, 50 Yonsei-ro Seodaemun-gu, Seoul 03722, Republic of Korea. [2] Technology Support Center Korea, Institute of Science and Technology, Seoul 02792, Republic of Korea. [3] Department of Materials, Imperial College London, London SW7 2AZ, UK. ✉email: jmoon@yonsei.ac.kr

Chiral photonics based on chiroptical phenomena have attracted tremendous scientific interest in a wide variety of fields, such as opto-spintronics[1,2], optical information processing[3], biological science[4], chiral biosensing[5,6], and quantum computing[7,8]. Chiral materials, which are commonly found in natural organic compounds, exhibit nonlinear optical responses depending on the polarization state of the circularly polarized light (CPL) owing to their inherently non-centrosymmetric nature. In particular, chiral organic materials have been widely exploited in optoelectronic devices based on the polarization phenomenon. Although organic chiral materials that retain various physical shapes are ubiquitous, the wavelength ranges in which chiroptical phenomena are revealed are limited to the near-ultraviolet (UV) region[9,10]. In addition, the poor charge-transfer capability of organic materials impedes their practical application to optoelectronic devices.

In 2017, our group rediscovered low-dimensional organic–inorganic hybrid perovskites (OIHPs) as a new class of chiral semiconductors, which have been also recognized as a novel platform for photovoltaics and light-emitting diodes (LEDs)[11]. We firstly reported that OIHPs with chiral organic ammonium cations exhibit circular dichroism (CD) depending on the different absorptions of left-handed circularly polarized light (LCP, $\sigma^+$) and right-handed circularly polarized light (RCP, $\sigma^-$). Since then, various chiral OIHPs in the forms of nanocrystals[12,13], co-gels[14], nanoplatelets[15], and thin films[16,17] have been much reported owing to their unusual spin-related optoelectronic properties, such as strong spin-orbit coupling[18], large Rashba splitting, long spin life time exceeding 1 ns[18,19], and long spin diffusion length over 85 nm[20]. For example, Long et al. demonstrated that 3% of circularly polarized photoluminescence (CPPL) was achieved at the temperature of 2 K even in the absence of an external magnetic field by varying the average number of inorganic layers[21]. Despite the superior chiroptical performance observed in chiral OIHPs, the chirality transfer mechanism from chiral bulky organic cations to achiral inorganic framework is still equivocal. To fully exploit the great potential of chiral OHIPs for spin-related quantum optics and spintronics, a clear understanding of the chiroptical activity origin is highly demanded.

To elucidate the origin of the chiroptical activity in chiral OIHPs, four different mechanisms involved in the chirality transfer phenomena have been suggested in organic–inorganic hybrid systems: (i) crystallization into a chiral crystal structure induced by chiral organic molecules[22,23], (ii) chiral distortion on the surface of inorganic semiconductors[24,25], (iii) chiral dislocations[26], and (iv) electronic interactions between the chiral organic molecules and inorganic semiconductors[27]. Since the chiral OIHPs with the Sohncke chiral space group of $P2_12_12_1$ were reported in 2003[28,29], their chiroptical phenomena have been interpreted based on the crystal structure–property relationship. Although the spatial interactions between chiral bulky organic molecules and the achiral inorganic framework (i.e., aforementioned mechanisms (i), (ii), and (iii)) provide a straightforward explanation of chirality transfer, the electronic interactions between the chiral organic molecules and achiral inorganic framework (i.e., mechanism (iv), which is less studied) should also be elaborately scrutinized. Very recently, it was demonstrated that a large $\pi$ bond ($\Pi_6^6$) with delocalized electrons of the organic spacer could effectively modify the electronic configuration of quasi-two-dimensional (2D) OIHPs via the coupling effect between the $\pi$-electron and $p$-orbital of the iodide in the inorganic framework[30]. Thus, via the delicate control of the electronic interaction between the chiral organic molecules and achiral inorganic framework, it is highly expected that the origin of the chirality transfer in chiral OIHPs could be clearly elucidated.

In this study, we systematically investigated the effects of electronic interaction between chiral organic spacer cations and inorganic framework on the chiroptical activity of chiral 2D OIHPs by modulating the interaction of the $\pi$-electron in chiral organic spacer cations. Through the spatially confined growth of chiral 2D OIHPs inside the varying nanopore-sized templates, the level of micro-strain on the 2D crystal lattice can be precisely controlled. The chiroptical activity (i.e., CD and CPPL) of chiral 2D OIHPs dramatically varied depending on the degree of micro-strain due to the varying hydrogen-bonding interactions between chiral organic spacer and achiral inorganic framework, which is originated from the redistributed conformation of benzene rings in the methylbenzylamine (MBA$^+$) cation. The resulting strain-imposed chiral 2D OIHPs grown inside the nanoporous template exhibited not only significantly magnified chirality but also handedness switching. Consequently, our experimental observations combined with theoretical simulation clearly demonstrated that controlling the electronic interaction between the chiral organic spacer cations and inorganic framework becomes a key enabler to design future chiral materials for high-performance spin-polarization-based optoelectronic devices.

## Results

**Chiroptical activity of nanoconfined chiral 2D OIHPs.** To examine the origin of chiroptical activity in chiral OIHPs, we utilized the strain-engineered chiral 2D OIHPs, which are grown within nanoporous anodized aluminum oxide (AAO) templates with varying pore sizes (Supplementary Figure 1)[31]. It is well-established that crystallization under nanoconfinement significantly differs from constraint-free growth by which the crystal structure of chiral OHIPs could be drastically altered by imposing a micro-strain onto the lattice[32]. It is worth noting that we began our investigation by focusing on a specific halide composition ($x = 0.325$) where the distinct phase transition occurs (see Supplementary Note 1 for the detailed reason and validity for choosing the halide composition)[16]. The same precursor solutions were deposited on a glass substrate (hereafter denoted as planar or substrate with 0 nm pore size) and AAO templates with different pore sizes, followed by spin coating and a subsequent annealing process. It is worth noting that the overlayer grown on top of the AAO template will disturb our experimental subject (i.e., nanoconfined chiral 2D OIHPs in the pore of AAO template: imposing the micro-strain into the lattice of chiral 2D OIHPs). Therefore, it is necessary to prevent the formation of such an overlayer by using a precursor solution with low concentration. As shown in Supplementary Figure 2, nanoconfined chiral 2D OIHPs in AAO templates grew as a single crystal without grain boundary, which is consistent with our observation in the previous reports[31]. The horizontal growth (parallel to the substrate) of single-crystalline chiral 2D OIHPs is effectively hampered by the pore walls of AAO templates as expected.

To shed light on the influence of the nanoconfined growth on chiroptical activity, the chiral 2D OIHPs thin films grown under different substrate conditions were characterized by CD measurements. As shown in Fig. 1a, the R-configuration chiral 2D OIHPs exhibited completely different CD spectra depending on the grown substrate types. Interestingly, AAO-templated chiral 2D OIHPs revealed exceptionally enhanced CD signals compared to their planar counterparts, even though the same precursor solutions were utilized (i.e., the same concentration of chiral organic spacer in chiral 2D OIHPs lattice). To quantitatively assess the effects of nanoconfined growth on chiroptical activity, the asymmetry factor (g-factor, $g_{CD}$) was calculated from the CD spectra and plotted in Fig. 1b using Eq. (1):

$$g_{CD} = \frac{CD}{(32980 \times extinction)} \qquad (1)$$

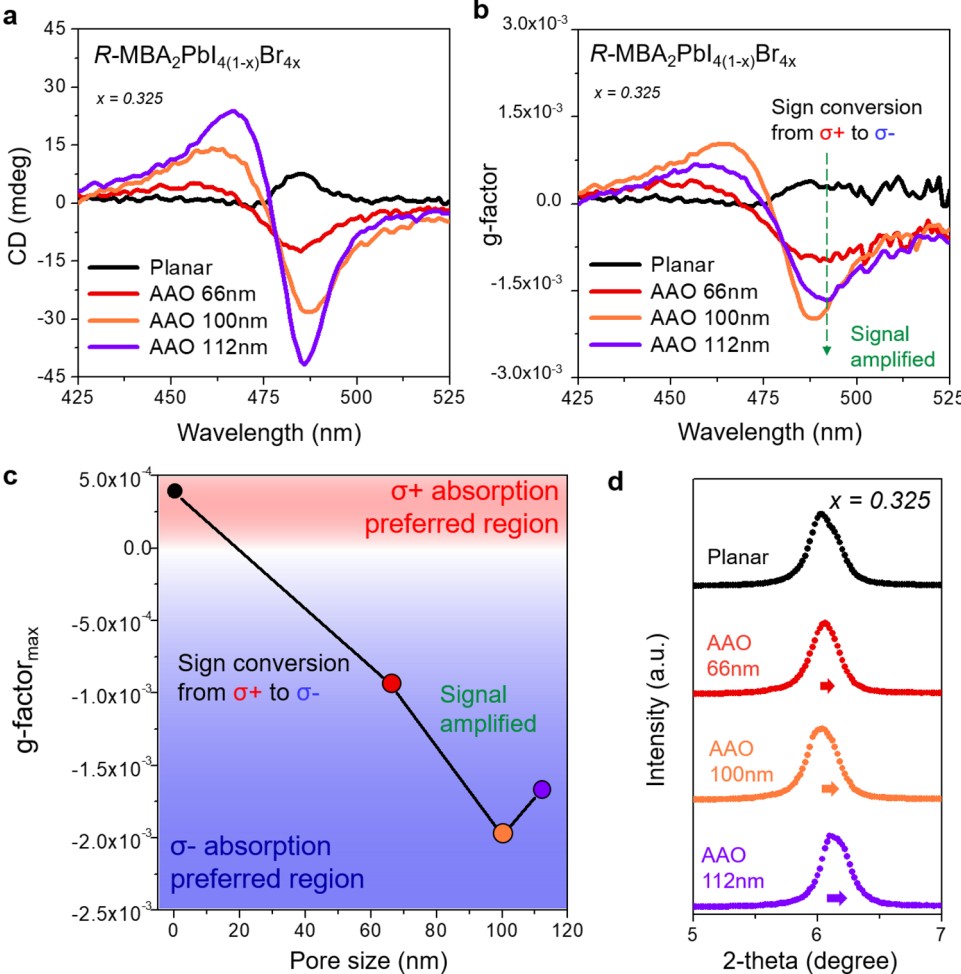

**Fig. 1 Chiroptical activity and in-depth structural studies of chiral 2D OIHPs grown under different substrate conditions. a** CD spectra of chiral 2D OIHPs thin films, (**b**) calculated g-factor from the CD spectra, and (**c**) plot of local maximum g-factor at the first extinction band edge as a function of AAO pore sizes. **d** thin-film XRD patterns of chiral 2D OIHPs grown on different substrate conditions with bromide composition of $x = 0.325$.

where CD and extinction are obtained from the CD and absorption spectra, respectively. As shown in Fig. 1c, the local maximum value of $g_{CD}$ around the first extinction band edge was significantly enhanced from $g_{CD} = 3.8 \times 10^{-4}$ for planar chiral 2D OIHPs to $g_{CD} = -2.0 \times 10^{-3}$ for 100 nm pore-sized AAO-templated chiral 2D OIHPs corresponding to a 5.12-fold improvement.

In addition to the huge amplification of the absolute $g_{CD}$ value in Fig. 1c, we also observed two striking conversion phenomena in the AAO-templated chiral 2D OIHPs grown under the nanoconfined condition. The first one is the sign conversion of the Cotton effect from positive for planar chiral 2D OIHPs to negative for AAO-templated chiral 2D OIHPs (green dashed arrow in Fig. 1b). Because the sign of the CD signal for chiral 2D OIHPs is determined by the handedness of the chiral organic spacer (whether it is an *S*- or *R*-configuration), this peculiar behavior has yet to be reported in chiral 2D OIHPs. The second one is the spectral shape conversion of the Cotton effect from unisignate for planar chiral 2D OIHPs to bisignate for AAO-templated chiral 2D OIHPs. The origin of the bisignate CD signal can be explained by the oscillator coupling theory where two (or more) different chromophores are located nearby in space and have a proper chiral mutual orientation[33,34]. In order to eliminate the interference induced by the optical anisotropic properties of AAO templates, CD measurement was also carefully investigated with empty AAO substrates with various pore sizes (Supplementary Figure 3).

Although the CD spectra of bare AAO templates exhibit huge CD signal (nearly 100 mdeg) due to the overestimated scattering contribution of transmitted CD measurement, which is common in nanostructured materials with definite spatial orientations[35], the empty AAO templates show no CD signal in the wavelength range above 350 nm. This implies that the effect of the optical anisotropy from the bare AAO templates can be completely excluded.

Recently, Di Bari and co-workers have reported that several organic thin films with macroscopic anisotropy can exhibit unexpected CD signal with a strong dependence on the light propagation direction (angle of incident light during the chiroptical measurement)[36,37], which stems from the optical interference of thin film's linear birefringence (LB) and linear dichroism (LD) (hereafter LDLB effect) rather than excitonic effects. Therefore, when investigating the chiroptical activities of thin films with macroscopic anisotropy, we need to consider a basic concept of Mueller matrix analysis; because the observed CD signal ($CD_{obs}$) is the sum of various contributions as represented by the Eq. (2):

$$CD_{obs} \approx CD_{true} + \frac{1}{2}(LD' \cdot LB - LD \cdot LB') \qquad (2)$$

where the first term refers to genuine CD, while the second term accounts for LDLB effect contribution (the signal of which is taken along an arbitrary axis defined in the laboratory frame and where the prime indicates a 45° axis rotation). It is necessary to exclude the

influence of LDLB contribution to explain the true effect of spatial confined growth on chiroptical activity of chiral 2D perovskites. Since the LDLB effect contribution is inverted upon sample flipping (i.e., flipping the sample by 180° with respect to the light propagation axis), the $CD_{true}$ term can be separately obtained by taking semi-sum of the two CD spectra with different measurement directions (i.e., front and back).

$$CD_{true} = 0.5 \times (CD_{obs, front} + CD_{obs, back}) \qquad (3)$$

The effect of nanoconfined growth in AAO template (i.e., huge amplification of CD signal) can be clearly observed in $CD_{true}$ spectra (Supplementary Fig. 4b), where the effect of the optical anisotropy resulted from the macroscopic nature is completely excluded. Consequently, it can be concluded that the observed chiroptical activities in the AAO-templated chiral 2D OIHPs (e.g., huge amplification of the absolute $g_{CD}$ value, sign conversion, and spectral shape change of the Cotton effect) are attributed to the effect of spatial confined growth of chiral 2D OIHPs rather than optical anisotropy from the bare AAO templates and macroscopic anisotropy of chiral 2D OIHPs.

**Micro-strain analysis of nanoconfined chiral 2D OIHPs.** To understand the origin of the two striking conversion phenomena observed in the AAO-templated chiral 2D OIHPs (i.e., sign conversion and spectral shape conversion of the Cotton effect), we obtained X-ray diffraction (XRD) patterns for chiral 2D OIHPs grown on different substrates. As shown in Fig. 1d, all the XRD patterns for chiral OIHPs (with a bromide ratio of 0.325) showed a single peak at $2\theta \approx 6.2°$ regardless of the grown substrate types, which corresponds to the iodide-determinant phase. Furthermore, the XRD for chiral OIHPs grown in AAO templates exhibited only marginal peak shifts within a range of $2\theta \approx 0.2°$ compared to the planar condition. This observation implies that unprecedented chiroptical phenomena of templated chiral 2D OIHPs cannot be explained in terms of the dichotomy between optically active iodide-determinant phase (chiral space group of $P2_12_12_1$) and optically non-active bromide-determinant phase (thermodynamically unfavorable phase)[38,39], which is based on the prevailing crystal structure-dependent chirality transfer mechanism.

To gain in-depth knowledge of the enhanced chiroptical phenomena observed by AAO-templated chiral 2D OIHPs (i.e., other than the crystalline structure-dependent chiroptical behavior), we performed local strain (micro-strain) analysis using a modified Williamson–Hall method (Fig. 2a). We calculated the degree of micro-strain in the lattice of chiral 2D OIHPs by analyzing peak broadening in the XRD patterns. For the accurate assessment of the micro-strain imposed by AAO templates, the XRD patterns for chiral 2D OIHPs with single iodine halide composition (i.e., $MBA_2PbI_4$) grown in different substrate conditions were obtained. The degree of micro-strain values imposed by different substrate conditions (planar; i.e., substrate with 0 nm pore size, AAO template with pore size of 66 nm, 100 nm, and 112 nm) were carefully calculated by the comparison with the strain-freely grown single crystalline $MBA_2PbI_4$. Detailed information on strain analysis is provided in Supplementary Note 2 and Supplementary Fig. 5.

In our previous report, we found that the magnitude of micro-strain imposed onto the lattice of 3D $MAPbI_{(3-x)}Cl_x$ and $CsPbI_3$ is inversely proportional to the pore size of AAO templates due to the reduction in the spatial capacity required for growth[31,40]. However, for the confined grown chiral 2D OIHPs within AAO templates, the magnitude of micro-strain rather revealed a zigzag tendency, as shown in Fig. 2b, than a linear dependency on the pore size. It is worth noting that the presence of −6% micro-strain in the lattice of OIHPs was observed in 0 nm pore size

condition, even though the thin films were freely grown on the planar substrate without spatial confinement. However, it is not uncommon that the local lattice strain exists in OIHPs thin films grown on the planar substrate due to the lattice mismatch, atomic size misfit, mismatch of thermal expansion, or lattice defects[41–43]. As mentioned above, we have calculated the degree of micro-strain by the comparison with the strain-free single-crystalline data (i.e., $MBA_2PbI_4$ single crystal as standard material), the obtained value of −6% micro-strain from the thin films of chiral 2D OIHPs grown in the planar substrate is reasonable.

The unexpected nonlinear behavior of calculated micro-strain values for AAO-templated chiral 2D OIHPs as a function of pore size can be possibly understood by the flexible structural nature of 2D OIHPs compared to their 3D counterparts. As illustrated in Supplementary Fig. 6, 3D OIHPs consist of a robust inorganic framework in which the lattice is connected with ionic–covalent mixed bonding. It would be difficult to break this strong chemical bonding between Pb–I, so compressive stress developed during confined growth not only reduces the unit cell size (e.g., induced micro-strain represented by the black arrow in Supplementary Fig. 6) but also results in octahedral tilting and lattice distortion of the Pb–I–Pb bond angle (expressed by the green arrow in Supplementary Fig. 6) while maintaining the inorganic framework structure. However, 2D OIHPs exhibit corner-sharing layered structures comprising of alternately stacked double layers of inorganic framework and two large organic spacers. Because two organic spacer cations are weakly bound by noncovalent π–π interactions, the stacking conformation of benzene rings (e.g., distance and angle between two benzene rings) can be readily varied depending on the imposed micro-strain level. As a result, the AAO-templated chiral 2D OIHPs tend to exhibit a zigzag tendency of micro-strain magnitude as a function of the template pore size, whereas the stacking conformation of organic spacer cations varies instead of inorganic framework distortion. Therefore, it is likely that the zigzag tendency observed in chiral 2D OIHPs manifests as a consequence of the lowest energy optimization of the π–π stacking structure for given spatial restriction conditions (i.e., pore size of the AAO template). The first-principles density function theory (DFT) calculation, which was conducted for $R$-$MBA_2PbI_4$ as a model compound, accentuate that the relative distance and angle between chiral organic spacers in chiral 2D OIHPs could be drastically adjusted by the π–π stacking conformation change during the nanoconfined growth inside AAO templates (see Supplementary Note 3 and Supplementary Figs. 7 and 8 for calculation details).

In previous literature, it is well known that the spectral peak position of photoluminescence (PL) could shift with varying the degree of micro-strain[44]. Therefore, to confirm the existence of the micro-strain in the lattice of chiral 2D OIHPs induced by the AAO templates, we conducted PL measurement with the chiral 2D OIHPs grown in different substrate conditions (e.g., planar, 66 nm, 100 nm, and 112 nm pore-sized AAO templates). As shown in Supplementary Fig. 9a, the PL spectra of chiral 2D OIHPs reveal completely different emission behavior depending upon the grown substrate. The chiral 2D OIHPs grown in AAO templates (regardless of pore sizes) exhibit significantly enhanced PL intensity than planar substrate condition due to the reduced defect density and suppressed trap states. These results indicate that, as previously reported, the confined growth in AAO templates efficiently controls the crystallization kinetic of OIHPs, resulting a high-quality single crystal in AAO templates[31,40]. Furthermore, to correlate the spectral peak shift of PL according to the induced micro-strain in the lattice of chiral 2D OIHPs, we carefully calculated the energy of PL emission obtained from each substrate conditions. The PL emission energy as a function of AAO template pore sizes was plotted in Supplementary Figure 9b.

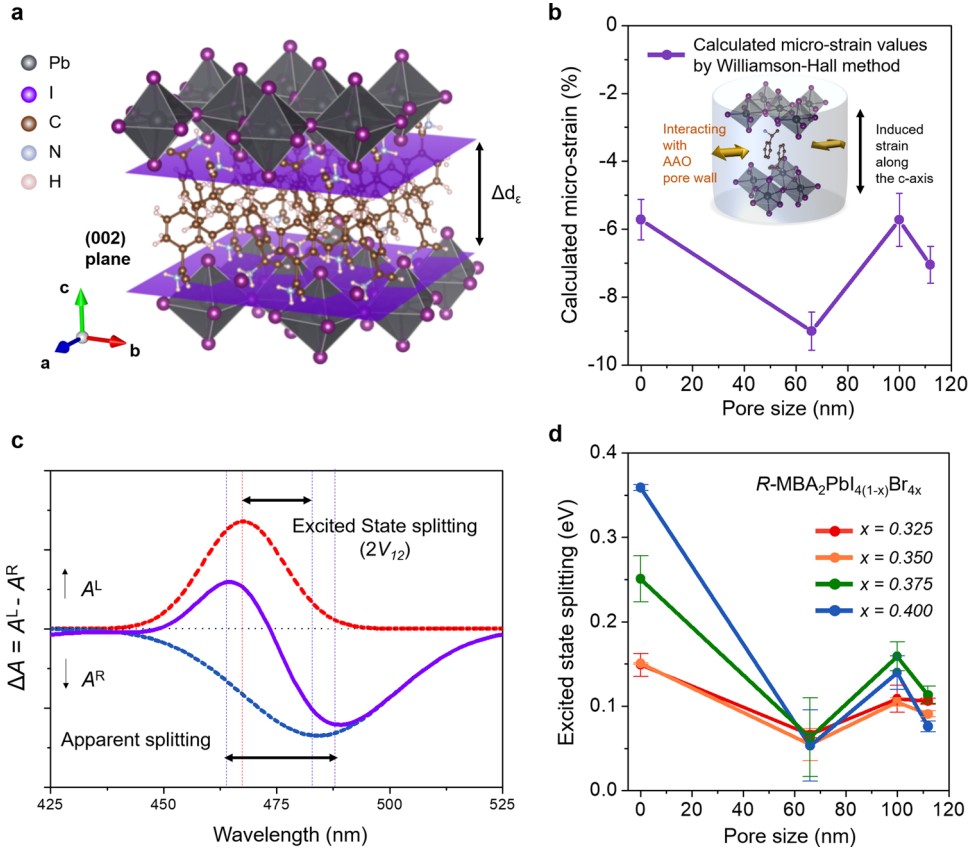

**Fig. 2 Estimation of micro-strain in chiral 2D OIHPs and stacking conformation variation induced by micro-strain. a** Schematic illustration of unit cell of R-MBA$_2$PbI$_4$ and induced change of $d$-spacing between the (002) planes. **b** Magnitude of calculated micro-strain values as a function of AAO template pore size. The inset represents the schematic illustration of nanoconfined growth of chiral 2D OIHPs in AAO template. The black arrow indicates the change in the lattice parameter along the $c$-axis. **c** Deconvolution results obtained from the CD spectra. The solid purple line represents obtained CD spectra from the chiral 2D OIHPs. The red and blue dot-line indicate the absorption of LCP and RCP, respectively. **d** Plot of excited state splitting values versus AAO template pore size. Error bars indicate the standard deviation. Note that the 0 nm pore size condition represents chiral 2D OIHPs grown on planar substrate without template.

Interestingly, the plot of PL emission energy also exhibits zigzag tendency as a function of the template pore size, which is similar to the calculated micro-strain plot. This implies that the PL emission shift is originated from the imposed micro-strain in the lattice of chiral 2D OIHPs. Such a coincident tendency (i.e., similar zigzag tendency in both PL emission energy and calculated micro-strain) reconfirmed the existence of the micro-strain imposed by AAO templates.

**Relationship between micro-strain and chiroptical activity.** Before establishing a plausible interpretation for observed significantly enhanced chiroptical activity of chiral 2D OIHPs in the presence of micro-strain, we must carefully check the generality of such abnormal chiroptical behaviors in the various composition range. The chiral 2D OIHPs thin films with different bromide ratio (from $x = 0.350$ to $x = 0.400$ with an interval of 0.025) was fabricated on different substrate conditions in the same manner as mentioned above. Interestingly, the abnormal chiroptical behavior (i.e., amplified CD signal, sign conversion, and spectra shape change in the Cotton effect) was also observed for the chiral 2D OIHPs thin films with all the compositions (from $x = 0.350$ to $0.400$ as well as $x = 0.325$) (Supplementary Fig. 10). We also conducted the CD measurement and XRD analysis with racemic MBA$_2$PbBr$_x$I$_{4-x}$ ($x = 0.325$) to clearly confirm the effect of micro-strain on the chiroptical activity of chiral 2D OIHPs. As shown in Supplementary Fig. 11, the CD spectra of racemic

compound do not exhibit any notable chiroptical response in the range of 425–525 nm regardless of the grown substrate conditions (e.g., planar, 66 nm, 100 nm, 112 nm pore-sized AAO templates), implying that the abnormal chiroptical behavior in chiral 2D OIHPs is not due to the optical anisotropy of the AAO substrate itself, but rather resulted from the promoted chirality transfer from organic spacer cations to achiral inorganic frameworks. The XRD spectra of racemic samples with different AAO templates condition also show no noticeable difference, suggesting that the nanoconfined growth in AAO templates do not influence on the crystallization process without impairing the preferred orientation and quality of chiral 2D OIHPs (Supplementary Fig. 12). These observations imply that the chirality transfer could be effectively promoted by micro-strain imposed by AAO templates, resulting in the modulation of chiroptical activity in chiral 2D OIHPs even at higher bromide composition.

The total CD spectra at the peak position result from the sum of the multiple excitonic transition in the optical spectrum (Fig. 2c)[34]. If two transition dipole moments are located close enough in space but are not coplanar, the coupling between two exciton transitions generate the splitting of excited states into two levels separated by $2V_{12}$, which is referred to as the excited state splitting[45,46]. The strength of the interaction can be calculated by the Coulomb dipole–dipole Eq. (4):

$$V_{12} = \frac{\mu_1 \mu_2}{r_{12}^3}[\overrightarrow{e_1} \cdot \overrightarrow{e_2} \cdot 3(\overrightarrow{e_1} \cdot \overrightarrow{e_{12}})(\overrightarrow{e_2} \cdot \overrightarrow{e_{12}})] \quad (4)$$

where $\mu_1$, $\mu_2$ and $r_{12}$ are the intensity of each transition dipole and distance between the two transition dipoles, while $\vec{e_i}$ is the corresponding unit vector. In this respect, the intensity of characteristic CD signal is proportional to the rotational strength ($R$) by the following Eq. (5), which considers the interaction between the electric transition dipole moments ($\mu_1$ and $\mu_2$) and also includes the terms relating to the coupling of electric transition dipole moment ($\mu$) and magnetic transition dipole moment ($m$) by Rosenfield mechanism[47,48].

$$R \propto r_{1,2} \cdot \mu_1 \times \mu_2 + \text{Im}\{(\mu_1 \mp \mu_2) \cdot (m_1 \mp m_2)\} \quad (5)$$

It is worth to noting that the abnormal chiroptical behaviors (i.e., amplified CD signal, sign conversion, and spectra shape change in the Cotton effect) observed in AAO-templated chiral 2D OIHPs occurs at ~475 nm (near the band edge extinction of chiral 2D OIHPs), which is far from the wavelength region where the exciton transition of chiral MBA$^+$ cations occurs (~260 nm). Therefore, the CD signal of chiral 2D OIHPs in Fig. 1a should be interpreted as a result of excitonic transition behavior in the lead halide inorganic framework where the chirality was induced by the chirality transfer phenomena. It is logical to conclude that the efficiency (or degree) of the chirality transfer can greatly vary depending upon the imposed micro-strain. In this manner, we propose the stepwise chirality transfer mechanism to interpret the unprecedent chiroptical activity of chiral 2D OIHPs in AAO templates: i) conformational stacking order of chiral organic cations (i.e., angle and length between the MBA$_1$ and MBA$_2$) changes due to the imposed micro-strain, ii) the electronic interaction between the chiral organic molecules and achiral inorganic framework was enhanced (or reduced), iii) the chirality transfer from the chiral organic cation to inorganic framework was promoted (or suppressed).

**Chirality transfer facilitated by micro-strain.** To verify the relationship between the imposed micro-strain and the efficiency of chirality transfer in chiral 2D OIHPs, we evaluated the excited state splitting values from the deconvoluted CD spectra where a bisgnate CD signal appeared around the extinction band edge $\lambda_0$. Using a multiple-peak fitting function, several peaks in chiral 2D OIHPs with various halide composition were identifiable (Supplementary Fig. 13; see Supplementary Note 4 for the detailed procedures and validity). The experimentally determined excited state splitting values for chiral 2D OIHPs, $2V_{12}$, are plotted in Fig. 2d as a function of the pore size of the AAO template (pore size = 0 for the planar substrate). Interestingly, in the entire composition range (from $x = 0.325$ to $x = 0.400$), the corresponding variation in excited state splitting also exhibited a zigzag tendency, which is similar to the calculated micro-strain results (Fig. 2b) and PL emission shift (Supplementary Fig. 9b). Such a coincident tendency (i.e., similar zigzag tendency in calculated micro-strain, PL emission shift as well as excited state splitting) can support that unprecedentedly observed chiroptical conversion behavior in AAO-templated chiral 2D OIHPs results from facilitated chirality transfer phenomena from chiral organic cations (MBA$^+$) to achiral inorganic framework (lead halide) induced by the micro-strain.

To support our scenario of facilitated chirality transfer phenomena by imposed micro-strain, we analyzed the structural properties of MBA$_2$PbI$_4$ as a function of micro-strain. It is worth mentioning that MBA$_2$PbI$_{4(1-x)}$Br$_{4x}$ thin films exhibit sharp XRD diffraction peaks assignable to the (002 $l$) planes regardless of the growing substrates and bromide composition, indicating the highly preferred orientation along the $c$-axis (Fig. 1d and Supplementary Figure 14). The observed peak shift in the iodide-determinant phase (toward higher 2θ degree) suggests

that confined growth induces lattice shrinkage along the $c$-axis. In addition, the horizontal growth of chiral 2D OIHPs is effectively inhibited by the pore wall (parallel to substrate) (as shown in Supplementary Fig. 2), the direction of imposed micro-strain is the out-of-plane direction (i.e., parallel to pore wall). Consequently, we need to focus on the shrinkage range in the out-of-plane direction (i.e., negative uniaxial strain and positive biaxial strain; yellow region in Fig. 3b, c) to properly interpret our DFT calculations. To correlate the imposed micro-strain and the degree of chirality transfer, the specific structural parameters such as the intra-octahedron distortions (i.e. $\Delta d$ and $\sigma^2$) were measured from our DFT-optimized structures as has been suggested by the Mitzi group[49]. $\Delta d$ represents the bond length distortion defined as $\Delta d = \sum (d_i - d_0)^2 / 6d_0^2$ ($d_i$ implies the six Pb–I bond lengths and $d_0$ is the average Pb–I bond length), and $\sigma^2$ is the bond angle variance defined as $\sigma2 = \sum_{i=1}^{12} (\theta_i - 90)^2 / 11$, where $\theta_i$ denotes the individual $cis$ I–Pb–I bond angles (Fig. 3a). Remarkably, in the compressive micro-strain-imposed region (as highlighted with yellow color in Fig. 3b), both $\Delta d$ and $\sigma^2$ increased sharply, implying that the degree of intra-octahedron distortion becomes larger when the lattice shrinkage occurs along the c-axis.

To elucidate how the micro-strain (nanoconfined growth in AAO templates) influences the degree of the intra-octahedron distortion (efficiency of the chirality transfer), we have also calculated the hydrogen-bonding length between NH$_3^+$ groups of chiral organic spacer and nearest iodine atom of the inorganic framework (Fig. 3a). Notably, there are four different distinguishable hydrogen bondings in the unit cell of MBA$_2$PbI$_4$ (denoted as HB$_{top1}$, HB$_{top2}$, HB$_{bot1}$, and HB$_{bot2}$ in Fig. 3a). As shown in Fig. 3c, the asymmetric nature of hydrogen bonding (variance and difference between the hydrogen-bonding length) was amplified when the lattice shrinkage occurs along the c-axis (region of negative uniaxial strain and positive biaxial strain as highlighted in Fig. 3c). The calculation results support that the asymmetric behavior of hydrogen bonding between the chiral organic molecules and inorganic framework can be amplified depending on the degree of micro-strain, which can promote the efficient chirality transfer process by increasing the chiral distortion in the inorganic framework. These results are consistent with a recent study by Jana et al, which discovered that asymmetric hydrogen-bonding interactions between the chiral spacer cation and lead bromide-based layers cause symmetry-breaking in the inorganic layer[45]. Indeed, asymmetric hydrogen-bonding interaction between the NH$_3^+$ of chiral spacer cations and inorganic layers are crucial for determining the associated electronic structure of chiral 2D OIHPs, thereby giving rise to chiroptical responses, such as CD and CPPL.

**CPPL in nanoconfined chiral 2D OIHPs.** Finally, we measured CPPL on chiral 2D OIHPs thin films with different growing substrates (Fig. 4a; see the Method section for details for the optical setup) to investigate the effects of asymmetric hydrogen-bonding induced by confined growth on the electronic structure of chiral 2D OIHPs. The CPPL spectra give useful information on the ground state of chiral 2D OIHPs, whereas CD spectroscopy can provide information regarding the electronic structure of the excited state of materials[34]. However, as the CPPL spectra can be highly influenced by the spin and energy relaxation process, it is difficult to derive the information of electronic structure by itself alone. Therefore, to probe the electronic structure of chiral 2D OIHPs, it is necessary to investigate these complementary spectroscopies (both CD and CPPL) rather than using each separately.

As shown in Fig. 4b, the room-temperature CPPL spectra for the 100 nm pore-sized AAO template-grown chiral 2D OIHPs

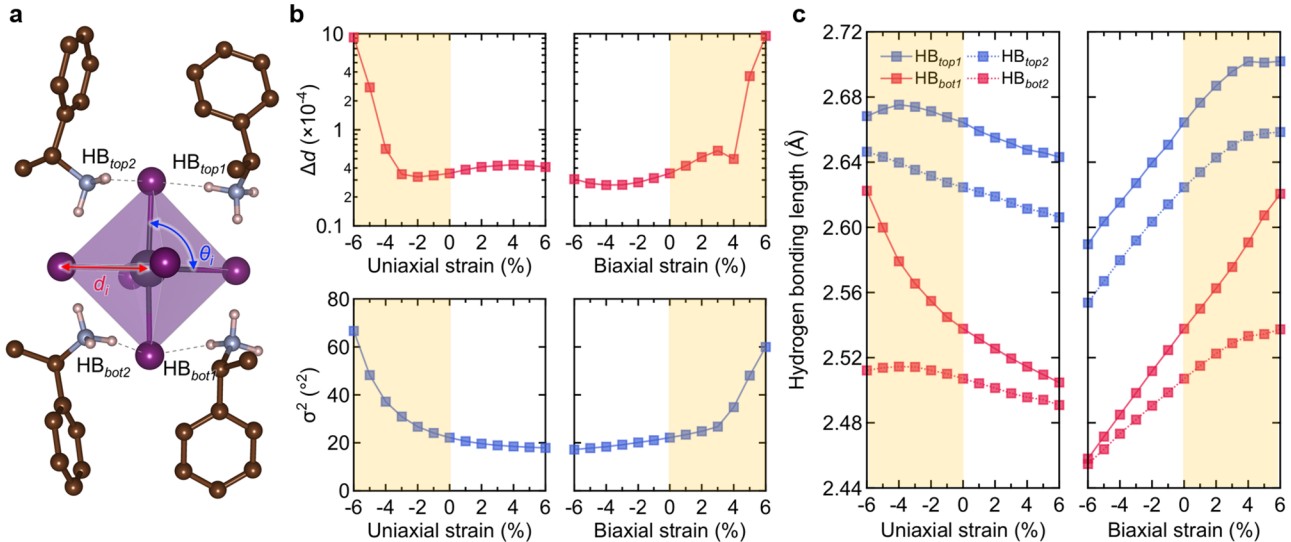

**Fig. 3 Structure properties of chiral 2D OIHPs and the influence of micro-strain on the specific structural parameters by theoretical calculation.** **a** Schematic illustration of intra-octahedron distortions and hydrogen bondings in DFT-optimized structure. **b** Bond length distortion and bond angle variance as a function of micro-strain. **c** Hydrogen-bonding length between the four different distinguishable hydrogen bondings in the unit cell of $MBA_2PbI_4$ (denoted as $HB_{top1}$, $HB_{top2}$, $HB_{bot1}$, and $HB_{bot2}$) and inorganic framework under uniaxial strain and biaxial strain. Highlighted yellow regions indicate the range where the lattice shrinkage occurs along the c-axis.

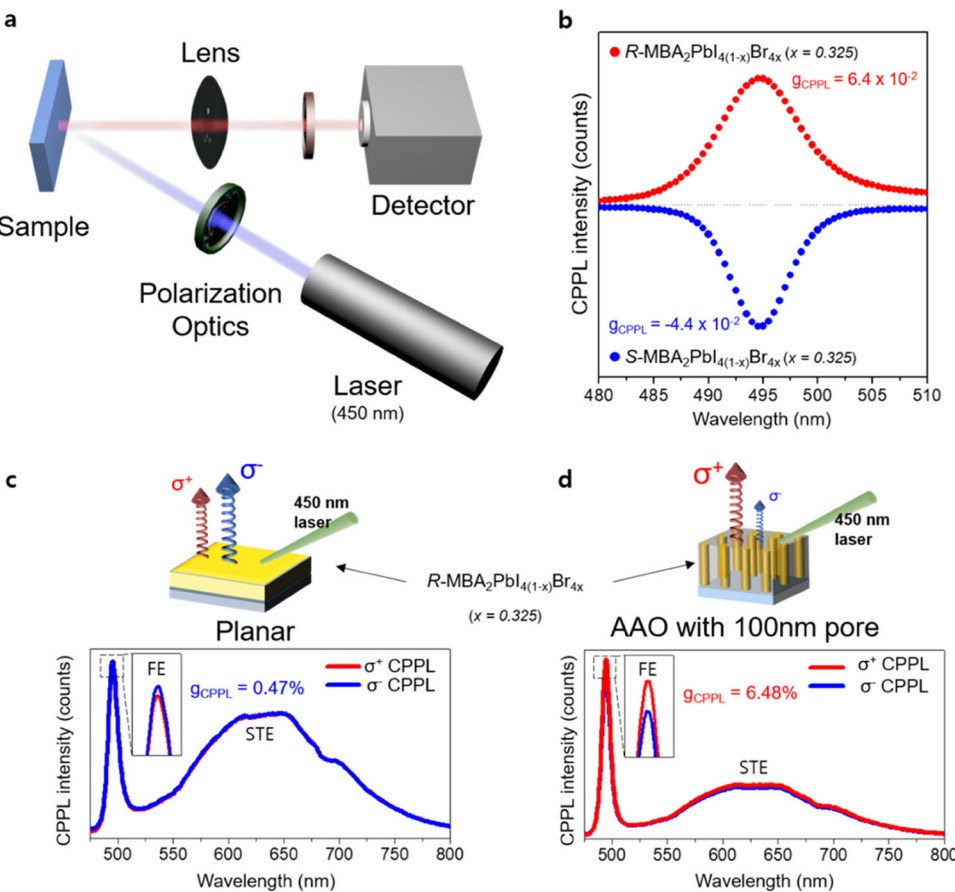

**Fig. 4 Effects of asymmetric hydrogen-bonding interaction on the electronic structure of chiral 2D OIHPs.** **a** Schematic illustration of optical setup for measuring the CPPL of chiral 2D OIHPs thin films. **b** Magnified CPPL spectra (near FE emission wavelength region) of $R$-$MBA_2PbI_{4(1-x)}Br_{4x}$ chiral 2D OIHPs ($x = 0.325$) grown in the AAO template with a 100 nm pore size. Full range of CPPL spectra (including FE and STE emission wavelength region) of $R$-$MBA_2PbI_{4(1-x)}Br_{4x}$ chiral 2D OIHPs ($x = 0.325$) grown on **c**, planar substrate condition, and (**d**), AAO with 100 nm pore size.

(i.e., having the largest $g_{CD}$ value among the samples) exhibited a clear free excitonic (FE) emission signal (centered at 495 nm), which was slightly below the first extinction band edge (ca. 475 nm with the composition of $x = 0.325$). To quantify the anisotropic emission characteristic of chiral 2D OIHPs, the asymmetry factor of CPPL ($g_{CPPL}$) was calculated from the CPPL spectra using the following definition:

$$g_{CPPL} = \frac{I_L - I_R}{\frac{1}{2}(I_L + I_R)} \qquad (6)$$

where $I_L$ and $I_R$ are the intensities of LCP and RCP light photoluminescence, respectively. The highest $g_{CPPL}$ of $6.4 \times 10^{-2}$ was obtained for $R$-configuration chiral 2D OIHPs grown inside the 100 nm pore-sized AAO template, which is, to the best of our knowledge, the reported highest polarization value in chiral OIHPs at room temperature. The $S$-configuration chiral 2D OIHPs grown within the 100 nm pore-sized AAO template revealed a similar CPPL spectra with an opposite sign and slightly lower value of $-4.4 \times 10^{-2}$. Because the order of magnitude and the signal-appearing wavelength region of CD and CPPL are similar, we can conclude that the observed chiroptical phenomena are based on the same ground and excited states corresponding to the first extinction band edge in chiral 2D OIHPs.

Due to the heavy atoms in OIHPs, large spin-orbit coupling (SOC) of OIHPs can lift the degeneracy of spin state and lead to large Rashba splitting if the structure lacks inversion symmetry[50,51]. Furthermore, in the presence of applied magnetic field (about 1 T − 5 T), the CPPL can be observed even in racemic compounds and 3D OIHPs without chirality transfer phenomena due to the field-induced population changes among the spin sublevels[18,21]. Compare to the unpolarized light, which consists of many electromagnetic waves polarized in different directions (i.e., net electric and magnetic field are zero), CPL is polarized only in one direction by passing through the polarizing filter, so that both the electric and magnetic field exist. Although the effective magnitude of external magnetic field for magneto-CPPL is quite large (as aforementioned; 1 T − 5 T) compared to the magnetic field in CPL, the excitation by CPL source can give rise to CPPL due to the field-effect rather chirality transfer phenomena. To exclude the effect of Rashba splitting due to large SOC of OIHPs on CPPL spectra, which might arise from the experimental procedure (because of the magnetic field in CPL), and to clarify the origin of different emission rates of RCP and LCP, the CPPL measurement was also performed in the same manner (using circular polarized light as an excitation source) for racemic compounds grown on AAO templates with a pore size of 100 nm. As shown in Supplementary Figure 15, the racemic compounds grown in AAO templates do not show any different emission behavior between the RCP and LCP. The CPPL spectra of racemic compounds grown on AAO templates suggested that the Rashba effect and coherent lifting of the spin degeneracy induced by the SOC of OIHPs did not occur in the absence of chirality transfer phenomena (i.e., in the absence of chiral organic molecules). This implies that the enhanced asymmetric factor of CPPL ($g_{CPPL}$) in chiral 2D OIHPs results from the facilitated chirality transfer phenomena rather than the Rashba effect itself (induced by the large SOC of OIHPs). Very recently, the Mitzi group found that the Rashba-Dresselhaus spin-splitting is a consequence of the chirality transfer phenomena[52]. Based on their findings and experimental results, the other effects such as the Rashba effect or Rashba-Dresselhaus spin-splitting (whether it occurs or not), which might contribute to the measured CPPL, cannot be explained separately. Rather, such effects should be included as a consequence of the chirality transfer phenomena facilitated by nanoconfined growth in AAO templates.

The full wavelength ranges of the CPPL spectra for chiral 2D OIHPs grown in different substrate conditions are also shown in Fig. 4c, d. Despite the same configuration for the chiral 2D OIHPs (i.e., $R$-MBA$_2$PbI$_{4(1-x)}$Br$_{4x}$ with a composition of $x = 0.325$), sign conversion from $\sigma^-$ CPPL for the planar sample to $\sigma^+$ for the AAO template sample was clearly observed, which is consistent with the sign conversion in the CD spectra (Fig. 1a, b). Furthermore, it is not uncommon for OIHPs to exhibit a weak and a broad PL band emission in the range from 500 to 800 nm. Such broad photoluminescence spectra have been widely reported in OIHPs, which is attributed to self-trapped excitons (STEs). STEs arise from the highly distorted layered framework of OIHPs, which are mediated by strong electron–phonon coupling and defects generated during crystal growth[53,54]. Therefore, the intensity ratio of narrow FEs emission (centered at 495 nm) to broadband STE emission (from 500 to 800 nm) can represent the degree of structural distortion. If the ratio of $I_{FE}/I_{STE}$ is lower than 1.0, the PL emission of OIHPs is governed by the electron–phonon coupling effect raised by structural distortion. Interestingly, the intensity ratio, $I_{FE}/I_{STE}$, drastically increased from 1.43 for the planar chiral 2D OIHPs to 3.19 for the 100 nm pore-sized AAO-templated chiral 2D OIHPs, implying that the PbX$_4$ inorganic framework structure is not distorted by the micro-strain imposed during nanoconfined growth. It is possible that the CPPL spectra (different emission rates between the RCP and LCP) and charge carrier dynamics can be changed with the incident angle of the excitation laser. To verify the effect of nanoconfined growth on charge carrier dynamics, we additionally investigated the CPPL spectra of chiral 2D perovskite in AAO templates with 100 nm pore size by varying the incident angle of the excitation laser (Supplementary Fig. 16a). As shown in Supplementary Figure 16b, despite the huge difference in incident angle (between 45° and 60°), the CPPL intensity exhibit only a negligible difference in FE transition, implying that the carrier and recombination dynamics is independent of the incident angle of the excitation laser. Therefore, we assume that the different chiroptical behavior (i.e., different anisotropy factor of photoluminescence of $4.7 \times 10^{-3}$ for planar and $6.48 \times 10^{-2}$ for AAO template in Fig. 4c, d, respectively) is attributed to different charge carrier dynamics induced by nanoconfined growth rather than the incident angle variation of the excitation laser.

## Discussion

Consequently, it can be concluded that the amplified chiroptical activity in AAO-templated chiral 2D OIHPs is more affected by the asymmetric hydrogen-bonding interaction between MBA cations and the inorganic framework rather than the structural distortion in the inorganic framework itself. Our findings suggest that the degree of chirality transfer (efficient chirality transfer phenomena) from chiral organic spacers to the achiral inorganic framework can be facilitated by enhancing the asymmetric nature of hydrogen-bonding interaction between chiral organic molecules and inorganic frameworks. While the chirality transfer mechanism in chiral 2D OIHPs has been frequently explained based on the inherent chiral crystal structure formation, the dipolar or electronic interaction mechanism should also be considered to gain a better understanding of chiroptical phenomena and boost the chiroptical activities of chiral 2D OIHPs.

In summary, we report the control of chiroptical phenomena (CD and CPPL) in chiral 2D OIHPs thin films through strain engineering. By adopting nanoconfined growth inside AAO templates, the conformational stacking of the $\pi$-electron in the chiral organic spacer cations were readily modulated. The induced conformational stacking variation promoted the asymmetric hydrogen-bonding interaction between the chiral organic

cations and inorganic framework, resulting in excellent asymmetry behavior with $g_{CD}$ of $2.0 \times 10^{-3}$ and $g_{CPPL}$ of $6.4 \times 10^{-2}$. To the best of our knowledge, the $g_{CPPL}$ exceeding the $10^{-2}$ order of magnitude at room temperature is the reported highest value so far for chiral 2D OIHPs. Furthermore, by using quantum mechanical calculations and the dipole–dipole interaction theory, we successfully elucidated the origin of unprecedented chiroptical phenomena (i.e., sign conversion of the Cotton effect, spectral shape change of the Cotton effect, and amplified CD intensity). Our experimental and theoretical calculation results clearly demonstrated that the electronic interaction between the molecular organic spacers and extended inorganic building blocks plays a critical role in interpreting the chirality transfer in chiral 2D OIHPs.

## Methods

**Materials preparation**. 1.2 mmol of organic amine (R- and S-MBA > 98%; Sigma-Aldrich, St. Louis, MO) and 1.3 mmol of hydroiodic acid in the form of an aqueous solution (57 wt% stabilized with 1.5% hypophosphorous acid; Alfa Aesar, Ward Hill, MA) were mixed with 0.5 mL of absolute ethanol (Merk, Darmstadt, Germany). After vigorous stirring for 12 h, the solution was fully evaporated at 80 °C in a vacuum to synthesize the chiral organic ammonium halide salts.

**Fabrication of chiral perovskites thin films on different substrates**. To fabricate the planar thin films, the synthesized chiral organic ammonium cation salts were dissolved in N,N-dimethyformamide (DMF; anhydrous, Sigma-Aldrich) along with $PbI_2$ (99.999%; Sigma-Aldrich) and $PbBr_2$ (99.999%; Sigma-Aldrich) at designated ratios to satisfy the chemical formulas of $(R\text{- or }S\text{-MBA})_2PbI_{4(1-x)}Br_{4x}$ where $x = 0.325, 0.350, 0.375$ and $0.400$. Dimethyl sulfoxide (DMSO; > 99.5%; Sigma-Aldrich) was added to the solutions to obtain a compact and dense morphology. Then, DMF was added to the solutions to make the total concentration in the solutions 20 wt%. The resulting solutions were spin coated onto a glass substrate at 3000 rpm for 30 s for planar chiral 2D OIHPs films. The solution-coated substrate was then annealed on a hot plate at 65 °C for 30 min. To fabricate the AAO-templated chiral 2D OIHPs thin films, the same precursor solution was used. The precursor solution was deposited on the AAO substrate with different pore sizes followed by evacuation at 125 Torr for 3 min under a vacuum to promote the infiltration of the precursor solution into the AAO pores. The samples were further spun at 6000 rpm for 60 s and then annealed at 65 °C for 30 min in a glove box.

**Characterization**. The surface morphologies of the AAO templates were confirmed by field emission scanning electron microscopy (FE-SEM, JSM-7001F, JEOL Ltd, Tokyo, Japan). By using ImageJ software (Wayne Rasband, National Institutes of Health, USA), the pore sizes of the AAO templates were examined. The polymorphism and crystallinities of the chiral 2D OIHPs on different substrates were determined using SmartLab (Rigaku) with a Cu Kα radiation source (0.15406 nm). CD data and extinction spectra were obtained using a CD spectrometer (J-815, JASCO, Easton, MD). The background was calibrated by air, and the spectra were obtained at a scan rate of 50 nm/min with a data pitch of 1 nm. CPPL spectra were measured using a PL spectrometer (FP-8500, JASCO). To measure the circularly polarized response of the chiral 2D OIHPs thin films, the Ti:sapphire laser (450 nm, Mai-tai, Spectra-Physics) was used, and linear light was converted to circularly polarized laser light by using a polarizer and compensator purchased from Newport Co.

**Computational details**. Based on the Kohn–Sham density functional theory[55], we performed total energy and electronic structure calculations and adopted periodic boundary conditions to represent the extended crystal. The Vienna Ab Initio Simulation Package (VASP)[56,57] was used with Projector augmented-wave (PAW)[58,59] method where the valence states of H, C, N, Pb, and I are treated explicitly by $1(1s^1)$, $4(2s^2 2p^2)$, $5(2s^2 2p^3)$, $14(5d^{10} 6s^2 6p^2)$, $7(5s^2 5p^5)$ electrons, respectively. The Perdew–Burke–Ernzerhof exchange-correlation functional (PBE)[60] with the Grimme D3[61] scheme for vdW corrections (i.e., PBE + D3) were employed, since lattice constants calculated from PBE + D3 show the most excellent agreement with lattice constants that were experimentally measured. A comparison with other functionals and vdW corrections can be found in Supplementary Figure 7. The plane-wave kinetic energy cutoff of 700 eV and a $6 \times 6 \times 2$ Γ-centered k-mesh were adopted during all calculations. The convergence criteria were set to $10^{-6}$ eV and $10^{-3}$ eV Å$^{-1}$ for total energy and atomic forces, respectively.

## Data availability

All data generated or analyzed during this study are included in this published article and its Supplementary Information. Source data are provided with this paper.

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

## Acknowledgements

This work was supported by the National Research Foundation (NRF) of Korea grant (No. 2018M3D1A1058793 and 2021R1A3B1068920) funded by the Ministry of Science and ICT. This research was also supported by the Yonsei Signature Research Cluster Program of 2021 (2021-22-0002).

## Author contributions

The fabrication process of the experimental samples was developed by C.U.L., H.L., and G.J. under the supervision of J.T. and S.M. XRD characterization was performed by S.M. S.M., J.A., and J. K., performed optical experiments and analyzed data under the supervision of J.M. Y.-K.J., and A.W. performed the density functional theory calculations, S.M. and J.M. wrote the manuscript with contributions from all other co-authors.

## Competing interests

The authors declare no competing interests.
