## [Peer Review File · Nature Communications]

Elucidating the origin of chiroptical activity in chiral 2D perovskites through nano-confined growthREVIEWER COMMENTS

Reviewer #1 (Remarks to the Author):

Sinihl Ma et al from Yonsey University investigate chirality transfer in chiral perovskites with the view of their engineering for polarization-based optoelectronic devices. The authors apply the classical oscillator coupling theory adapted for the organic-inorganic (aka hybrid, <https://onlinelibrary.wiley.com/doi/abs/10.1002/adma.201807628>) low-dimensional perovskite crystals with various compositions: MBA_2PbI_4 as well as $\text{MBA}_2\text{PbI}_4(1-x)\text{Br}_4x$ with $x = 160, 0.325, 0.350$. They observed quite high optical activity asymmetry for absorption and fluorescence for nanoconfined perovskite. The conclusion of the study that chirality transfer phenomena have direct effect on the perovskite structure and can be utilized in the design of chiral photonic and spintronic devices.

Overall the manuscript describes a timely well-designed study with significant potential impact. It can be accepted for the publication in N.Comm with the following revisions and edits.

p.2. :“ Chiral photonics based 40 on chiroptical phenomena...” One of the significant streams of studies in chiral photonics is about biosensing (e.g. <https://pubs.acs.org/doi/abs/10.1021/nl900726s>; https://www.sciencedirect.com/science/article/pii/S0001868621000178?dgcid=rss_sd_all; <https://pubs.rsc.org/en/content/chapterhtml/2016/bk9781849739832-00001?isbn=978-1-84973-983-2>). Since biosensing applications are important for perovskites as well, It makes sense mentioning it in the intro part.

p.7 “...but also results in octahedral tilting and lattice distortion of the Pb–I–Pb bond angle...” This is the most significant finding of the manuscript that, in my view, underlines all the other calculations shown after that. The authors demonstrated that the constrains of the AAO template induce chiral deformation of the lattice. The deformations of the crystal lattice had been shown before to cause enhancement of the chiroptical activity (<https://science.sciencemag.org/content/359/6373/309.editor-summary> ; <https://pubs.acs.org/doi/10.1021/nl504369x>) How is it different from the prior studies?

p.13, Figure 4. The CPPL data are very interesting and complementary. However, both CD and potentially CPPL spectra of AAO templated MBA_2PbI_4 can depend on the incident angle. It will likely to change the spectra and thus the small changes in FE peaks can be ascribed to the change in the incident angle.

Reviewer #2 (Remarks to the Author):

In this manuscript, Ma and coauthors have investigated the chirality transfer mechanism in chiral perovskites. They concluded that the multi-polar interaction between chiral molecular spacers and the inorganic framework plays a key role in promoting the chiroptical activity of chiral perovskites via the in-depth investigation of chiroptical phenomena-based oscillator coupling theory and theoretical calculations. Chiral perovskites would find promising applications in spintronic- and polarization-based optoelectronic devices and understanding the chirality transfer mechanism would be essential for the material design and device construction. Therefore, this study is important to the chiral perovskite community. Nevertheless, the conclusions cannot fully supported by the provided experimental evidences. I cannot recommend its publication at the current form.

1 The detailed material characterizations for the strain-imposed chiral 2D perovskites are missing. Are those strain-imposed chiral 2D perovskites in the pores of AAO single crystals? What is the size distribution of the as-grown 2D perovskites on AAO substrates? The authors also need to provide the morphology information on the as-grown samples.

- 2 If the as-grown chiral 2D perovskite samples are not single crystals, how the results the authors obtained will change? How the non-uniform size influences the conclusions the authors obtained?
- 3 The authors claimed that there is strain in the samples grown on AAO substrates. Can the authors directly provide the experimental evidences and estimate the strain from them? For example, PL spectrum would change under external strain.
- 4 In addition to strain, there might be other factors that alter the chiroptical behaviors such as quantum confinement effect itself. How can the authors exclude other possibilities?
- 5 My most concern is the optical anisotropy of the samples grown on AAO substrates. For those samples, large optical anisotropy could be expected, which would greatly affect the accuracy of the measured CD signals and CPL. If this occurs, any conclusion obtained in this manuscript needs to be reconsidered. Or how can the authors exclude the influence from the optical anisotropy of the samples?
- 6 The CD spectra of perovskite grown on planar substrates exhibit unisignated CD signal at the first excitonic transition. Can the author explain the reason for uncommon behavior (not bisignated Cotton effect but unisignated CD)?
- 7 The authors are suggested to provide the results of rac-MBA samples grown on both planar and AAO substrate for comparisons.
- 8 I noticed that the authors used circularly polarized light to excite samples for CPPL measurement. Under such case, other effect such as Rashba effect might also contribute to the measured CPPL, which is not the intrinsic CPPL due to the chirality transfer.
- 9 Some important recent works on chiral 2D perovskites are missing. The authors are suggested to introduce and cite them in the revised manuscript.

Reviewer #3 (Remarks to the Author):

Please see attached referee report (PDF file)

Manuscript number: Nature Communications manuscript NCOMMS-21-30935-T

Title: Multi-Polar Interaction: The Origin of Chiroptical Activity in Chiral 2D Perovskites

Authors: Sunihl Ma et al.

Synopsis:

The major scientific problem which this study attempts to address is the mechanism of chiro-optical activities within chiral layered organic-inorganic hybrid perovskite films (OIHPs).

Ma *et al.* report experimental results entailing the growth of chiral OIHPs on, or inside, anodized aluminum oxide (AAO) nanopore substrates. The material grown is chiral $\text{MBA}_2\text{PbI}_{4(1-x)}\text{Br}_{4x}$ (with x either equal to zero, or $x = 0.325-0.35$). Circular dichroism (CD) and dissymmetry spectra measured on these samples exhibit different spectral features depending on the nature of the substrate used for sample growth (Fig1): While planar substrates show a mono-signate CD response at the exciton peak, films grown in or on nanopore substrates, with pore dimensions 66nm, 100nm, and 112nm, show bi-signate CD spectra and a significant enhancement in the peak dissymmetry relative to planar grown films. Additionally the authors claim to determine the micro-strain of the films grown in/on AAO substrates as a function of the AAO nano-pore size; strains of up to $\sim -9\%$ are reported. Plots provided show data for 0 nm pore size, which shows -

6% strain. The authors conclude the work by claiming “control of chiro-optical phenomena” via “strain engineering” (line 351).

The authors’ claims are largely based on an interpretation in terms of a model wherein nano-confined growth on nano-pore substrates, with its resultant micro-strain, is assumed to modify the relative distance and angle between the chiral organic molecules in the crystal structure. This modification is in turn assumed to affect the CD spectra by way of Davydov splitting (Eq. 2): Namely, the inter-cation dipole-dipole interaction is assumed to determine the splitting between excited state transitions associated with the MBA cations. These transitions are assigned within deconvolved CD spectra (Fig 2c). Based on this deconvolution procedure applied to the measured CD spectra, values are determined for the Davydov splitting and correlated with pore-size and micro-strain; ab-initio band structures calculations are presented to show the effect of the micro-strain on the band structure. The authors also present circular polarized PL measurements on the films.

Analysis

I was excited by the title and abstract of this manuscript. The authors attempt to address a difficult and important problem (the origin/mechanism of chiro-optical phenomena in chiral metal halide systems) and the abstract promises analysis in terms of a very interesting new idea, namely, multipolar interactions involving the chiral cations in these systems. Not mentioned in the abstract was the technique used of growth in nano-pore substrates; this is a very clever idea.

However after reading the work I find that the claims made are not well supported by the work, it seems very preliminary. In addition, the manuscript is missing some essential information and the presentation needs to be substantially improved.

Here I summarize what I see as the major technical issues; further below I will make comments on the presentation of the manuscript and list other minor technical issues I noted while reading the manuscript.

Major technical issues:

1. **Analysis of micro-strain of films grown on AAO nanopore substrates.** Fig 2b shows the micro-strain determined for films grown on AAO nanopore substrates. The authors describe in the SI the procedure whereby these substrates were fabricated. But no-where is there a definition of an AAO substrate with pore size of 0 nm as shown in Fig 2b.
 - a. I assume that a pore size of 0 nm means “there are no pores”.
 - b. *How then is it possible that the micro-strain shown for the 0nm pore size in Fig 2b is -6%?* It is difficult to believe that this is a correct analysis. If I understand correctly what the 0 nm pore size means, then this sample comprises a planar film. Given the weak non-covalent interaction between layers it is highly unlikely that these films are epitaxial so the -6% strain of this “0 nm pore” sample is quite a mystery. Indeed there is no correlation between the derived micro-strain and the nano-pore size (see plot in Fig2b). The Authors’ own description on line 167, is that the nano-pore size is of “unexpected irrelevance”.

- c. Yet the interpretation of the XRD pattern in terms of the micro-strain is a foundational element of the author's interpretation and claims.
2. CD and dissymmetry spectra are shown in Fig 1 and in the SI, and details of the deconvolution procedure they use to assign transitions are described in the SI (Suppl Note 4). However, no absorption or extinction spectra are shown. It is problematic to evaluate statements about the nature of the spectral features in the CD spectra and how they are deconvolved and assigned without seeing these spectra.
3. The analysis in terms of Davydov splitting of the MBA cation transitions is a concept based in Frenkel exciton theory and it is a very interesting idea.
 - a. But the MBA cation exciton transition is at - 260nm (See Fig S3 in DOI: 10.1039/c7mh00197e from this group); it is never explained adequately here how this transition would be relevant to the exciton transition at - 475nm which is associated with the lead-iodide framework.
 - b. To be clear, the authors claim on line 275 *et seq* that Fig 3b “suggests” that the MBA HOMO forms a “resonant state with the inorganic PbI₄ framework” but this is an assertion, and it is certainly not proven. The assertion does not seem to be backed up by the energy level calculations in Fig 3 and it contradicts what the Authors state about the frontier orbitals being associated with the inorganic sub-lattice (in line 262-262).
 - c. If Davydov splitting in nano-confined MBAPbX₄ is responsible for the emergence of bi-signate CD spectra, as claimed here, why are bi-signate CD spectra (sometimes) observed in planar thin film samples as reported for example in DOI: 10.1039/c7mh00197e from this group?
4. The authors claim that the CD “intensity” is given by Eq. 3. I do not understand the basis for this statement. It is not explained or derived and I am not aware of a literature precedent.

Comments on the presentation and additional technical issues:

In this section I comment on aspects of the manuscript presentation that I think ought to be corrected; as well as some other technical issues I noted while reading the manuscript.

1. Given the importance of the nano-confined growth in nanopore substrates in this study, this rather clever technique ought to be mentioned at least in the abstract-- if not in the title; however, nothing is said about the technique until page 4 of the manuscript.
2. The actual material system studied (chiral MBA₂PbI_{4(1-x)}Br_{4x}) is not stated until page 6 of the manuscript: It is not stated in the abstract or title; leading one to wonder what x in line 107 on page 4 refers to. The impression given in the title is that a very general analysis will be forthcoming; but that was not the case.
3. Line 34 in the abstract is rather unclear: “(different absorption of 2.0×10^{-3} and distinct photoluminescence of 6.4×10^{-2} for left- and right-handed circularly polarized light)” There are terms for what the authors are trying to describe here: “dissymmetry” and “anisotropy factor”. I suggest these terms be used.
4. Line 78 why “inconspicuous”?
5. Line 79 “it is” -> “it was”

6. Page 4 line 122: “intensities” is not a correct term to use here.
7. Page 6, line 148-152. I find it hard to accept the general statement made, when only one XRD peak is shown. Properly, an analysis would need to be made of the XRD patterns to actually determine the space groups before making such a broad claim: “This observation implies that unprecedented chiroptical phenomena of templated chiral 2D OIHPs cannot be explained in terms of the dichotomy between optically active chiral space group of $P2_12_12_1$ (iodide-determinant phase) and nonchiral space group (bromide-determinant phase)”
8. The analysis of micro-strain based on XRD linewidths, shown in Fig 2, is described in the SI in Supplementary Note 2. There the authors state that Scherrer broadening is neglected since chiral OIHPs “have grain size larger than a few hundreds of nanometers”. How is that statement relevant to chiral OIHP films grown in nanopore substrates with pore sizes less than ~100nm?
9. Page 9 and Fig 2 show the “experimentally determined Davydov splitting”. But Fig 2c and Fig S7 are not properly labelled and are not properly explained:
 - a. There are three lines in Fig 2c; none of them are labelled or explained in the legend or in the figure caption.
 - b. Supplementary Note 4 contains a description of the deconvolution procedure and contains Fig S7, an expanded spectral view of the same quantities that are plotted in Fig 2c. But Fig S-7 also has no legends and the caption has no explanation of what the lines plotted refer to.
 - c. The text in Suppl. Note 4 describes the figure in an unclear manner. The text on lines 198-200 states: “The dotted and bold lines (in Fig. S8) indicate the preferential absorption peak of LCP or RCP corresponding to each excitonic transition (Gaussians) and measured CD spectra, respectively.” I gather that the bold lines show CD but the dotted lines show absorption. If so, why is the vertical axis labelled CD and not absorption? There should be two separate axes, one for CD, and a separate vertical axis for absorption.
 - d. The text on lines 198-200 in Suppl. Note 4 should refer to Fig S7, not Fig.S8.

Page 12 contains a statement on line 303-306 which reads, “The CPPL spectra give useful information on the ground state of chiral 2D OIHPs, whereas CD

spectroscopy can provide information regarding the electronic structure of the excited state of materials”. This statement is oversimplified and is actually confusing/misleading. The ground state of chiral OIHPs (or any system) has no excitons. Both CD and CPPL give information on transitions that involve the ground state and various excited states. It is true that CD (and absorption spectroscopy generally) can access information about transitions involving higher energy states than PL spectroscopies can access generally; but PL spectra such as CPPL are heavily influence by spin and energy relaxation processes and are therefore often difficult to interpret meaningfully.

Reviewer #4 (Remarks to the Author):

In the article "Multi-Polar Interaction: The Origin of Chiroptical Activity in Chiral 2D Perovskites" the

authors investigated 2D hybrid organic-inorganic perovskites having chiro-optical activity. In particular they focus on $\text{MBA}_2\text{PbI}_4(1-x)\text{Br}_{4x}$ systems. The study involves both experimental characterization and theoretical explanations, mainly based on Density Functional Theory (DFT) calculations.

Starting from experimental characterization of the optical activity due to the chirality of the structure, they proposed a theoretical interpretation in terms of an indirect interaction (there is no chemical bonding) between organic spacers (chiral molecular cations) and the framework.

We discussions seem appropriate and supported both from experiments and calculations.

I believe that the article will be interesting for the large community working on hybrid perovskites, and certainly, 2D Chiral HIOPs are an hot topic in material science. Therefore, I recommend publication in Nat. Comm.

I have a minor comment:

Formula (3), line 239, should not be called quadruple product.

A quadruple product involves 4 vectors, and it can be either scalar or vector quadruple product. In this case, V_{12} is simply a scalar quantity (Davydov splitting), which acts as a scale factor.

Therefore, I would refer to a triple product (the quantity in square brackets), scaled by V_{12} .

Response Letter

Journal: Nature Communication

Manuscript number: NCOMMS-21-30935-T

Title: “Multi-Polar Interaction: The Origin of Chiroptical Activity in Chiral 2D Perovskites”

Author(s): Sunihl Ma¹, Young-Kwang Jung¹, Jihoon Ahn¹, Jihoon Kyhm², Jeiwan Tan,¹ Hyungsoo Lee,¹ Gyumin Jang,¹ Chan Uk Lee,¹ Aron Walsh^{1,3}, and Jooho Moon^{1,*}

<Reviewer 1>

Sunihl Ma et al from Yonsei University investigate chirality transfer in chiral perovskites with the view of their engineering for polarization-based optoelectronic devices. The authors apply the classical oscillator coupling theory adapted for the organic-inorganic (aka hybrid, <https://onlinelibrary.wiley.com/doi/abs/10.1002/adma.201807628>) low-dimensional perovskite crystals with various compositions: MBA_2PbI_4 as well as $\text{MBA}_2\text{PbI}_4(1-x)\text{Br}_4x$ with $f_x = 0.325, 0.350$. They observed quite high optical activity asymmetry for absorption and fluorescence for nanoconfined perovskite. The conclusion of the study that chirality transfer phenomena have direct effect on the perovskite structure and can be utilized in the design of chiral photonic and spintronic devices. Overall the manuscript describes a timely well-designed study with significant potential impact. It can be accepted for the publication in N.Comm with the following revisions and edits.

Remark:

We would like to thank the reviewer for evaluating our work. Our response to the reviewer's comments can be found below.

Comment 1:

p.2. : “Chiral photonics based 40 on chiroptical phenomena....”

One of the significant streams of studies in chiral photonics is about biosensing (e.g. <https://pubs.acs.org/doi/abs/10.1021/nl900726s>;

https://www.sciencedirect.com/science/article/pii/S0001868621000178?dgcid=rss_sd_all;

[https://pubs.rsc.org/en/content/chapterhtml/2016/bk9781849739832-00001?isbn=978-1-](https://pubs.rsc.org/en/content/chapterhtml/2016/bk9781849739832-00001?isbn=978-1-84973-983-2)

84973-983-2). Since biosensing applications are important for perovskites as well. It makes sense mentioning it in the intro part.

Author's Response:

We agree with the reviewer that the one of the important mainstream topics for chiral photonic studies are bio responsive imaging [R1] and chiral biosensing [R2]. Therefore, as the reviewer commented, the statement related to the biosensing and potential application of chiral perovskite have been added. We have also cited some references that reviewer recommended. We thank the reviewer for careful comment.

References cited in this response:

- R1 Heffern, M. C., Matosziuk, L. M. & Meade, T. J. Lanthanide probes for bioresponsive imaging. *Chem. Rev.* **114**, 4496–4539 (2014).
- R2 Paiva-Marques WA, Gomez FR, Oliveira Jr ON, Ricardo Mejia-Salazar J. Chiralplasmonics and their potential for point-of-care biosensing applications. *Sensors* **20**, 944 (2020)

Revision made (colored in blue):

(in Page 2, Introduction)

Chiral photonics based on chiroptical phenomena have attracted tremendous scientific interest in a wide variety of fields, such as opto-spintronics,^{1,2} optical information processing,³ biological science,⁴ **chiral biosensing**,^{5,6} and quantum computing.^{7,8} ...

(References; 5 and 6 are added)

5. Chen W, *et al.* Nanoparticle Superstructures Made by Polymerase Chain Reaction: Collective Interactions of Nanoparticles and a New Principle for Chiral Materials. *Nano Lett.* **9**, 2153-2159 (2009).
6. Wen Y, He MQ, Yu YL, Wang JH. Biomolecule-mediated chiral nanostructures: a review of chiral mechanism and application. *Adv. Colloid Interface Sci.* **289**, 102376 (2021).

Comment 2:

p.7. : "...but also results in octahedral tilting and lattice distortion of the Pb–I–Pb bond angle..." This is the most significant finding of the manuscript that, in my view, underlines all the other calculations shown after that. The authors demonstrated that the constrains of the AAO template induce chiral deformation of the lattice. The deformations of the crystal lattice had been shown before to cause enhancement of the chiroptical activity (<https://science.sciencemag.org/content/359/6373/309.editor-summary>;

<https://pubs.acs.org/doi/10.1021/nl504369x>)

How is it different from the prior studies?

Author's Response:

We appreciate the reviewer for important comment regarding the conceptual novelty of our strategy based on strain-engineering for enhancing the chiroptical activity. Although we observed the lattice distortion of chiral 2D perovskites induced by nanoconfined growth in AAO templates (XRD patterns in Fig. 1d and calculated micro-strain data obtained by modified Williamson-Hall methods in Fig. 2b), we elucidate that **the resulting octahedral tilting and lattice distortion in achiral inorganic framework can be amplified by the asymmetric hydrogen bonding between the achiral inorganic framework and chiral organic cations, which is the fundamental origin of amplified chiroptical activity in nanoconfined chiral 2D perovskites.** Indeed, we focused on the conformational stacking change of benzene rings in chiral methylbenzylamine (MBA) cations due to the weak bonding nature of noncovalent π – π interactions. Because the stacking conformation (*e.g.*, distance and angles between the benzene rings) of benzene ring in MBA organic spacer cations varies, the AAO templated chiral 2D OIHPs tend to exhibit a zigzag tendency of micro-strain magnitude rather than a linear dependency on the pore size as shown in previously reported 3D perovskites [R3, R4]. In addition, based on the first-principles density functional theory (DFT) calculation results, we concluded that **the variation of hydrogen bonding interaction between the achiral inorganic framework and chiral organic molecules (not the structural variation) is likely a key origin of the amplified chiroptical activity of chiral 2D perovskites.**

In the previous report that reviewer commented [R5], the chiroptical activity of Co₃O₄ nanoparticles (NPs) is attributed to the lattice distortion on surface of NPs caused by molecular imprinting (*i.e.*, attaching chiral surface ligand such as cysteine). Although the chirality transfer phenomena are propagating from chiral ligand to the lattice of the inorganic NPs core, the

chirality transfer mechanism is based on the **spatially rearrangement of atoms in achiral system** (not originated from the inherent chiral crystal structure). On the other hand, the incorporation of the chiral MBA molecules (not adsorption on the surface) into the lattice results in an **inherent chiral crystal structure** (chiral space group of $P2_12_12_1$ for MBA_2PbI_4). Therefore, the origin of chiroptical phenomena in these two systems are based on the completely different chirality transfer mechanisms (e.g., *surface chiral ligand induced atomic rearrangement versus inherent chiral crystal structure formation*). In the case of semiconductor nanowires (NWs) in previous report (*Nano Lett.* **15**, 1710-1715 (2015)), the chirality transfer mechanism is based on the chiral lattice distortion, which is similar to the above-mentioned chirality transfer mechanism in NPs (**spatial rearrangement of atoms**).

Consequently, we assert that our results are completely different from the previous studies that reviewer mentioned. Furthermore, we would like to emphasize that our discovery and understanding of chiroptical activity in chiral 2D perovskites are unprecedented for the following reasons (*vide infra*):

(a) New strategy for improving chiroptical activity in chiral 2D perovskites

Although the attractive spin-related and chiroptical properties of chiral 2D perovskites have led to a recent research interest, the numerous studies have been limited to only suggest new composition of chiral perovskites (e.g., substitution of chiral organic molecules, halide composition engineering, and dimension control). On the contrary, to the best our knowledge, **this is the first report on enhancing the chiroptical activity of chiral 2D perovskite without any compositional change.** Even though the same precursor solutions were utilized (*i.e.*, the same ratio between the chiral molecules to inorganic lead halide precursor; the same dimension of perovskites, and the same halide composition), nanoconfined chiral 2D perovskites revealed exceptionally enhanced chiroptical activity (circular dichroism (CD) and circularly polarized photoluminescence (CPPL)) compared to their planar counterparts.

(b) Elucidating the origin of chiroptical activity other than crystal structure

Due to the structural flexibility of 2D perovskites, the chiroptical activities in chiral perovskite are usually interpreted as the chirality transfer mechanism based on inherent chiral crystal structure. Although the crystal structure-property relationship provides a straightforward explanation of chirality transfer phenomena, significantly enhanced chiroptical activity in our experimental results cannot be fully explained by the prevailing chirality transfer

mechanism. Therefore, we suspect that the huge enhancement of chiroptical activity might originate from the other underestimated mechanism; not from crystal structure-chiroptical property relationship. Very recently, Mitzi et al. discovered that the induced CD of chiral 2D OIHPs do not always accompany structural chirality transfer from organic to inorganic layers as manifested by centrosymmetric breaking in inorganic frameworks [R6]. Their experimental results and theoretical calculation are consistent with our findings. For the first time, we observed that the hydrogen bonding interaction between chiral molecular spacers and the inorganic framework can be modified by imposing micro-strain into the lattice of 2D perovskite. Combining experimental results and DFT calculation, we verified that the chirality transfer mechanism based on the electronic interaction plays a key role in promoting the chiroptical activity of chiral perovskites.

References cited in this response:

- R3 Kwon, H. C., Kim, A., Lee, H., Lee, D., Jeong, S., Moon, J. Parallelized Nanopillar Perovskites for Semitransparent Solar Cells Using an Anodized Aluminum Oxide Scaffold. *Adv. Energy Mater.* **6**, 1601055 (2016)..
- R4 Ma, S., Kim, S. H., Jeong, B., Kwon, H. C., Yun, S. C., Jang, G., *et al.* Strain-Mediated Phase Stabilization: a New Strategy for Ultrastable α -CsPbI₃ Perovskite by Nanoconfined Growth. *Small* **15**, 1900219 (2019).
- R5 Yeom J, Santos US, Chekini M, Cha M, de Moura AF, Kotov NA. Chiro-magnetic nanoparticles and gels. *Science* **359**, 309-314 (2018).
- R6 Jana MK, et al. Organic-to-inorganic structural chirality transfer in a 2D hybrid perovskite and impact on Rashba-Dresselhaus spin-orbit coupling. *Nat. Commun.* **11**, 4699 (2020).

Comment 3:

p.13, Figure 4. The CPPL data are very interesting and complementary. However, both CD and potentially CPPL spectra of AAO templated MBA2PbI can depend on the incident angle. It will likely to change the spectra and thus the small changes in FE peaks can be ascribed to the change in the incident angle.

Author's Response:

We appreciate the reviewer for constructive comments regarding the change of spectra as a function of incident light angle. We understood that reviewer asks us to exclude the effect of incident angle on chiroptical activities associated with free exciton (FE) transition that arises from the experiment condition. However, when we conducted the chiroptical analysis (both CD and CPPL), the measurement condition (including incident light angle as well as relative humidity and temperature) were carefully controlled. Regardless of the substrate conditions, all the CD and CPPL spectra (Fig. 1 and Fig. 4, respectively) for the chiral 2D perovskite were obtained under the same measurement conditions. Therefore, we assume that the different chiroptical behavior (*i.e.*, different anisotropy factor of photoluminescence of 4.7×10^{-3} for planar and 6.48×10^{-2} for AAO template in Fig.4c and d, respectively) is attributed to different charge carrier dynamics induced by nanoconfined growth rather than the incident angle variation of excitation laser. [R7, R8]

To verify our hypothesis and clarify the effect of nanoconfined growth on charge carrier and recombination dynamics, we additionally investigated the CPPL spectra of chiral 2D perovskite in AAO templates with 100 nm pore size by varying the incident angle of excitation laser (*e.g.*, 45° and 60° as shown in Fig.R1a), as reviewer suggested. However, as shown in Fig. R1b, despite of the huge difference of incident angle, the CPPL intensity exhibit only negligible difference in FE transition. Because we precisely controlled the experimental procedure so that CPPL are conducted with the same incident angle, the variation of incidence angle during the CPPL measurement, if presented, would be in the range of a few degrees or less. Therefore, we assume that the different chiroptical behavior in FE transition presented in Fig. 4 is attributed to different charge carrier dynamics induced by nanoconfined growth rather than the incident angle variation of excitation laser.

Fig. R1. a. A photograph of CPPL experimental setup with varying the incident angle of excitation laser. **b.** The CPPL spectra of $R\text{-MBA}_2\text{PbI}_{4(1-x)}\text{Br}_{4x}$ ($x = 0.325$) depending on the incident angle of excitation laser.

References cited in this response:

- R7 Kwon, H. C., Kim, A., Lee, H., Lee, D., Jeong, S., Moon, J. Parallelized Nanopillar Perovskites for Semitransparent Solar Cells Using an Anodized Aluminum Oxide Scaffold. *Adv. Energy Mater.* **6**, 1601055 (2016).
- R8 Ma, S., Kim, S. H., Jeong, B., Kwon, H. C., Yun, S. C., Jang, G., *et al.* Strain-Mediated Phase Stabilization: a New Strategy for Ultrastable $\alpha\text{-CsPbI}_3$ Perovskite by Nanoconfined Growth. *Small* **15**, 1900219 (2019).

Revision made (colored in blue):

(in Page 16)

… Interestingly, the intensity ratio, I_{FE}/I_{STE} , drastically increased from 1.43 for the planar chiral 2D OIHPs to 3.19 for the 100 nm pore-sized AAO templated chiral 2D OIHPs, implying that the PbX_4 inorganic framework structure is not distorted by the micro-strain imposed during nanoconfined growth. It is possible that the CPPL spectra (different emission rate between the RCP and LCP) and charge carrier dynamics can be changed with the incident angle of excitation laser. To verify the effect of nanoconfined growth on charge carrier dynamics, we additionally investigated the CPPL spectra of chiral 2D perovskite in AAO templates with 100 nm pore size by varying the incident angle of excitation laser (Supplementary Fig. S15a). As

shown in Fig. S15b, despite of the huge difference of incident angle (between 45° and 60°), the CPPL intensity exhibit only negligible difference in FE transition, implying that the carrier and recombination dynamics is independent on the incident angle of excitation laser. Therefore, we assume that the different chiroptical behavior (*i.e.*, different anisotropy factor of photoluminescence of 4.7×10^{-3} for planar and 6.48×10^{-2} for AAO template in Fig.4c and d, respectively) is attributed to different charge carrier dynamics induced by nanoconfined growth rather than the incident angle variation of excitation laser.

(Supplementary; Fig. S15 was added)

Fig. S15. **a.** A photograph of CPPL experimental setup with varying the incident angle of excitation laser. **b.** The CPPL spectra of $R\text{-MBA}_2\text{PbI}_{4(1-x)}\text{Br}_{4x}$ ($x = 0.325$) depending on the incident angle of excitation laser.

<Reviewer 2>

In this manuscript, Ma and coauthors have investigated the chirality transfer mechanism in chiral perovskites. They concluded that the multi-polar interaction between chiral molecular spacers and the inorganic framework plays a key role in promoting the chiroptical activity of chiral perovskites via the in-depth investigation of chiroptical phenomena-based oscillator coupling theory and theoretical calculations. Chiral perovskites would find promising applications in spintronic- and polarization-based optoelectronic devices and understanding the chirality transfer mechanism would be essential for the material design and device construction. Therefore, this study is important to the chiral perovskite community. Nevertheless, the conclusions cannot fully supported by the provided experimental evidences. I cannot recommend its publication at the current form.

Remark:

We would like to gratefully thank the reviewer for reviewing and evaluating our work. We believe that the reviewer's comments highly improve the quality of our manuscript. Our response to the reviewer's comments can be found below.

Comment 1:

The detailed material characterizations for the strain-imposed chiral 2D perovskites are missing. Are those strain-imposed chiral 2D perovskites in the pores of AAO single crystals? What is the size distribution of the as-grown 2D perovskites on AAO substrates? The authors also need to provide the morphology information on the as-grown samples.

Author's Response:

We appreciate the reviewer's critical comment on the material characterizations for the confined grown 2D perovskites. The additional crystallographic and morphological analyses have been conducted to characterize the crystalline properties. First, to investigate whether the strain-imposed chiral 2D perovskites are single crystal, high-resolution transmission electron microscope (HRTEM) was applied to analyze the structure and morphology of the crystalline grains. Since the overlayer grown on the top of the AAO template will disturb our experimental subject (*i.e.*, nanoconfined chiral 2D perovskite inside the pore of AAO template), it is necessary to prevent the formation of such an overlayer by using a precursor solution with low

concentration. Due to the low precursor concentration, the cross-sectional HRTEM image in Fig. R2 reveals that the pores of AAO template are partially filled with conjoined chiral 2D perovskite nanocrystals. There are no grain boundaries in conjoined chiral 2D perovskites, confirming that **strain-imposed chiral 2D perovskites grew as a single crystal**, which is also observed in our previous report [R9, R10]. Although there is large empty space in the pores of AAO templates, the horizontal growth of chiral 2D perovskite (parallel to substrate) is sufficient to fill the pores in the horizontal direction. This implies that the growth of chiral 2D perovskite along the horizontal direction is hampered by the pore wall, inducing the micro-strain into the lattice along the c-axis (nano-confined growth direction) which is parallel to the preferred orientation of chiral 2D perovskite ((002*l*) plane). These results, which are consistent with our findings of the induced micro-strain direction in the X-ray diffraction (XRD) patterns (Fig. 1d) and modified Williamson-Hall method (Fig. 2b), can also justify our interpretation on DFT calculations that focus on the shrinkage range in the out-of-plane direction (*e.g.*, negative uniaxial strain; *i.e.*, yellow region in Fig. 3c in our manuscript).

Since the interaction direction between the pore wall of AAO templates and chiral 2D perovskites is parallel to substrate, the average lateral size of single crystal (parallel to AAO substrates), which can determine the degree of micro-strain along the c-axis, is the main concern of our investigation. As the lateral size of single crystal is limited by the pore size of AAO templates, we can calculate the average horizontal size of single crystal by analyzing the pore size distribution. By using ImageJ software (Wayne Rasband, National Institutes of Health, USA), we estimated the size distribution for three different AAO templates as 66.4 ± 1.3 , 100.3 ± 3.9 , and 112.7 ± 5.2 nm. **The image analysis confirms that the pore size of AAO templates exhibits a very narrow distribution.** This narrow distribution of pore size results in coherent distribution of average horizontal sizes of nanoconfined chiral 2D perovskites in AAO templates, ensuring the reliable experimental results obtained from the micro-strain analysis. We thank the reviewer for improving the reliability and quality of our work.

Fig. R2. The morphology of chiral 2D OIHPs grown inside AAO templates and high-resolution transmission electron microscopy (HRTEM) image of chiral 2D OIHPs confined in AAO templates with pore size of 66 nm condition.

References cited in this response:

- R9 Kwon, H. C., Kim, A., Lee, H., Lee, D., Jeong, S., Moon, J. Parallelized Nanopillar Perovskites for Semitransparent Solar Cells Using an Anodized Aluminum Oxide Scaffold. *Adv. Energy Mater.* **6**, 1601055 (2016).
- R10 Ma, S., Kim, S. H., Jeong, B., Kwon, H. C., Yun, S. C., Jang, G., *et al.* Strain-Mediated Phase Stabilization: a New Strategy for Ultrastable α -CsPbI₃ Perovskite by Nanoconfined Growth. *Small* **15**, 1900219 (2019).

Revision made (colored in blue):

(in Page 4-5)

... The same precursor solutions were deposited on a glass substrate (hereafter denoted as planar) and AAO templates with different pore sizes, followed by spin coating and a subsequent annealing process. It is worth to note that the overlayer grown on top of the AAO template will disturb our experimental subject (*i.e.*, nanoconfined chiral 2D OIHPs in the pore of AAO template: imposing the micro-strain into the lattice of chiral 2D OIHPs). Therefore, it is necessary to prevent the formation of such an overlayer by using a precursor solution with low concentration. As shown in Supplementary Fig. S2, nanoconfined chiral 2D OIHPs in AAO templates grew as a single crystal without grain boundary, which is consistent with our observation in previous report.³¹ The horizontal growth (parallel to the substrate) of single crystalline chiral 2D OIHPs is effectively hampered by the pore walls of AAO templates as expected.

(in Page 13)

... ~~In addition, the~~ The observed peak shift in the iodide-determinant phase (toward higher 2θ degree) suggests that confined growth induces lattice shrinkage along the c -axis. In addition, the horizontal growth of chiral 2D OIHPs is effectively inhibited by the pore wall (parallel to substrate) (as shown in Fig.S2 Supplementary), the direction of imposed micro-strain is out-of-plane direction (*i.e.*, parallel to pore wall). Therefore, we need to focus on the shrinkage range in the out-of-plane direction (*i.e.*, negative uniaxial strain and positive biaxial strain; yellow region in Fig. 3c and d) to properly interpret our DFT calculations.

(Supplementary; Fig. S2 and related comments were added)

Fig. S2. The morphology of chiral 2D OIHPs grown inside AAO templates and high-resolution transmission electron microscopy (HRTEM) image of chiral 2D OIHPs confined in AAO templates with pore size of 66 nm condition.

As the lateral size of single crystal is limited by the pore size of AAO templates, we can calculate the average horizontal size of single crystal by analyzing the pore size distribution. By using ImageJ software (Wayne Rasband, National Institutes of Health, USA), we estimated the size distribution for three different AAO templates as 66.4 ± 1.3 , 100.3 ± 3.9 , and 112.7 ± 5.2 nm (Supplementary Fig. S2). The image analysis confirms that the pore size of AAO templates (*i.e.*, lateral size of single crystalline chiral 2D OIHPs) exhibits a very narrow distribution.

Comment 2:

If the as-grown chiral 2D perovskite samples are not single crystals, how the results the authors obtained will change? How the non-uniform size influences the conclusions the authors obtained?

Author's Response:

We appreciate the reviewer's critical comment on the statistical analysis of crystalline size of chiral 2D OIHPs with respect to the reliability of our experimental finding. As we explained before (in our Response to **Comment 1** for *Reviewer 2*), the pore size in AAO templates as well as lateral size of single crystal chiral 2D OIHPs exhibits very narrow size distribution. In addition, the micro-strain values by the modified Williamson-Hall method were statistically calculated from the several repetitions of XRD measurement (average calculated strain values and standard deviation were presented in Fig. 2b). Therefore, we believe that the concern resulted from the size distribution or non-uniform size can be excluded from our conclusion.

Comment 3:

The authors claimed that there is strain in the samples grown on AAO substrates. Can the authors directly provide the experimental evidences and estimate the strain from them? For example, PL spectrum would change under external strain.

Author's Response:

We appreciate the constructive comments raised by the reviewer. In previous literature, the spectral peak position of PL could shift to lower energy as pressure increases [R11]. Furthermore, we observed the lattice shrinkage along the c-axis in nanoconfined chiral 2D OIHPs, which can lead to the bandgap narrowing. Therefore, as the reviewer suggested, we performed photoluminescence (PL) measurement with the chiral 2D OIHPs grown in different substrate conditions (*e.g.*, planar, 66 nm-, 100 nm-, and 112 nm-pore sized AAO templates) to provide direct experimental evidence regarding imposed strain in chiral 2D OIHPs. As shown in Fig. R3, the PL spectra of chiral 2D OIHPs reveal completely different emission behavior depending upon the grown substrates. The chiral 2D OIHPs grown in AAO templates (regardless of pore sizes) exhibit significantly enhanced PL intensity than planar substrate condition. It is attributed to the reduced defect density and suppressed trap states due to the well-controlled crystallization during the confined growth.

To correlate the spectral peak shift of PL according to the induced micro-strain in the lattice of chiral 2D OIHPs, we carefully calculated the energy of PL emission from each substrate condition. The PL emission energy as a function of AAO template pore sizes was plotted in Fig. R4a. For clear comparison, the plot of calculated micro-strain values versus pore size (Fig. 2b in our manuscript) is also presented as Fig. R4b. Interestingly, the plot of PL emission energy also exhibits zigzag tendency as a function of the template pore size, which is similar to the calculated micro-strain plot. It is worth noting that the direction of y-axis in both plots (*i.e.*, PL emission energy in Fig. R4a and calculated micro-strain in Fig. R4b) represents the same physical meaning; bandgap narrowing and lattice shrinkage direction (from top to bottom). Consequently, it is concluded that the PL emission shift can be attributed to the imposed micro-strain in the lattice of chiral 2D OIHPs and such a coincident tendency (*i.e.*, similar zigzag tendency in both PL emission energy and calculated micro-strain) can provide direct evidence regarding micro-strain imposed by AAO templates.

Fig. R3. Steady-state photoluminescence (PL) spectra of $R\text{-MBA}_2\text{PbBr}_x\text{I}_{4-x}$ ($x=0.325$) grown in various substrate conditions when excited by laser with wavelength of 325 nm.

Fig. R4. The correlation between the AAO template pore size and induced micro-strain in chiral 2D OIHPs. **a**, The PL emission peak energy plot obtained from the steady-state PL spectra, and **b**, calculated micro-strain plot calculated by Williamson-Hall method as a function of AAO template pore sizes.

References cited in this response:

R11 Liu S, et al. Manipulating efficient light emission in two-dimensional perovskite crystals by pressure-induced anisotropic deformation. *Sci. Adv.* **5**, eaav9445 (2019).

Revision made (colored in blue):

(in Page 9-10)

... conformation change during the nanoconfined growth inside AAO templates (see Supplementary Note 3 and Supplementary Fig. S4 and Fig. S5 for calculation details).

In previous literature, it is well known that the spectral peak position of photoluminescence (PL) could shift with varying the degree of micro-strain.⁴² Therefore, to confirm the existence of the micro-strain in the lattice of chiral 2D OIHPs induced by the AAO templates, we conducted PL measurement with the chiral 2D OIHPs grown in different substrate conditions (*e.g.*, planar, 66 nm-, 100 nm-, and 112 nm-pore sized AAO templates). As shown in Supplementary Fig. S8a, the PL spectra of chiral 2D OIHPs reveal completely different emission behavior depending upon the grown substrate. The chiral 2D OIHPs grown in AAO templates (regardless of pore sizes) exhibit significantly enhanced PL intensity than planar substrate condition due to the reduced defect density and suppressed trap states. These results indicate that, as previously reported, the confined growth in AAO templates efficiently controls the crystallization kinetic of OIHPs, resulting in a high-quality single crystal in AAO templates.^{31,38} Furthermore, to correlate the spectral peak shift of PL according to the induced micro-strain in the lattice of chiral 2D OIHPs, we carefully calculated the energy of PL emission obtained from each substrate condition. The PL emission energy as a function of AAO template pore sizes was plotted in Supplementary Fig. S8b. Interestingly, the plot of PL emission energy also exhibits zigzag tendency as a function of the template pore size, which is similar to the calculated micro-strain plot. This implies that the PL emission shift is originated from the imposed micro-strain in the lattice of chiral 2D OIHPs. Such a coincident tendency (*i.e.*, similar zigzag tendency in both PL emission energy and calculated micro-strain) reconfirmed the existence of the micro-strain imposed by AAO templates.

(in Page 12)

... Interestingly, in the entire composition range (from $x = 0.325$ to $x = 0.400$), the corresponding variation in Davydov splitting also exhibited a zigzag tendency, which is similar

to the calculated micro-strain results (Fig. 2b) and PL emission shift (Supplementary Fig. S8b). Such a coincident tendency (*i.e.*, similar zigzag tendency in calculated micro-strain, PL emission shift as well as Davydov splitting) can support that unprecedentedly observed chiroptical conversion behavior in AAO templated chiral 2D OIHPs results from facilitated chirality transfer phenomena from chiral organic cations (MBA^+) to achiral inorganic framework (lead halide) induced by the micro-strain.

(Supplementary; Fig. S8 was added)

Fig. S8. The correlation between the AAO template pore sizes and induced micro-strain in chiral 2D OIHPs. **a**, Steady-state photoluminescence (PL) spectra of $R\text{-MBA}_2\text{PbBr}_x\text{I}_{4-x}$ ($x=0.325$) grown in various substrate conditions when excited by laser with wavelength of 325 nm. **b**, Corresponding the PL emission peak energy plot obtained from the steady-state PL spectra as a function of AAO template pore sizes.

Comment 4:

In addition to strain, there might be other factors that alter the chiroptical behaviors such as quantum confinement effect itself. How can the authors exclude other possibilities?

Author's Response:

We thank the reviewer for the important comment on the origin of the unprecedented chiroptical activity in chiral 2D OIHPs. We understood that reviewer asks us to consider the other possible origin of the chiroptical behavior that could arise from the confined growth in AAO templates, such as quantum confinement effect. As the reviewer commented, strong quantum confinement effect might cause bandgap widening and change of electronic structure as well as carrier transport dynamics. However, the quantum confinement effect can be observed when the size of particles is less than 10 nm (*e.g.*, quantum dots (QDs) or nanoparticles (NPs) whose size is comparable to the wavelength of electron. Even the smallest single crystal size of chiral 2D OIHPs is about 60 nm as defined by the pore size of AAO templates, which far exceeds the electron wavelength. Furthermore, our DFT calculation results suggested that the interaction between the inorganic perovskites slabs and the chiral organic spacer was enhanced by the micro-strain induced due to the lattice shrinkage along the *c*-axis. It is well known that strong interaction between the inorganic frameworks and the chiral organic spacer cation weakens the quantum confinement effects in 2D OIHPs.[R12, R13] Therefore, we can exclude the quantum confinement effect on the chiroptical activity of chiral 2D OIHPs.

References cited in this response:

- R12 Zhou YL, et al. Optical Coupling Between Chiral Biomolecules and Semiconductor Nanoparticles: Size-Dependent Circular Dichroism Absorption. *Angew. Chem. Int. Edit.* **50**, 11456-11459 (2011).
- R13 Liu C, *et al.* Imidazolium Ionic Liquid as Organic Spacer for Tuning the Excitonic Structure of 2D Perovskite Materials. *ACS Energy Lett.* **5**, 3617-3627 (2020).

Comment 5:

My most concern is the optical anisotropy of the samples grown on AAO substrates. For those samples, large optical anisotropy could be expected, which would greatly affect the accuracy of the measured CD signals and CPL. If this occurs, any conclusion obtained in this manuscript needs to be reconsidered. Or how can the authors exclude the influence from the optical anisotropy of the samples?

Author's Response:

We appreciate the reviewer's valuable comment on the optical anisotropy, which can be originated from the AAO substrate itself. As the reviewer commented, the optical anisotropy due to the AAO substrate itself may interfere with the origin of chiroptical activity in chiral 2D OIHPs grown inside AAO templates. In order to eliminate the interference induced by the optical anisotropic properties of AAO templates, circular dichroism (CD) measurement was carefully investigated with various empty AAO substrates. Fig. R5a and b show the linear absorption and CD spectra of bare AAO templates with various pore size, respectively. Due to its high transparency of AAO template, which consists of the wide bandgap metal oxide such as Al₂O₃, TiO₂ and fluorine-doped SnO₂, bare AAO substrates have no optical activity in the region longer than 300 nm wavelength in which we are interested (the region where the chiral 2D OIHPs exhibit chiroptical activity; *i.e.*, wavelength range from 425 nm to 525 nm). Although the CD spectra of empty AAO templates exhibit huge CD signal (nearly 100 mdeg), those optical activity is originated from the limitation of transmitted CD measurement. Because the transmitted CD signal includes both absorption- (A) and scattering-contribution (R) from the sample (*i.e.*, $CD=(A_L-A_R)+(S_L-S_R)$), the CD signal can be overestimated by the scattering contribution, which is common in nanostructured materials with definite spatial orientations.[R14] However, as mentioned above, as the bare AAO substrates show no CD signal in the wavelength range longer than 350 nm, so that the effect of the optical anisotropy from the bare AAO templates can be completely excluded from our experimental results, interpretation, and conclusion.

Fig. R5. Optical activity of bare AAO substrates with various pore size. a. Linear absorption spectra and b. CD spectra obtained from the empty AAO substrates without chiral 2D OIHPs.

References cited in this response:

R14 Duan YY, *et al.* Optically Active Nanostructured ZnO Films. *Angew. Chem. Int. Edit.* **54**, 15170-15175 (2015).

Revision made (colored in blue):

(in Page 6)

... different chromophores are located nearby in space and have a proper chiral mutual orientation.^{33,34} In order to eliminate the interference induced by the optical anisotropic properties of AAO templates, CD measurement was also carefully investigated with empty AAO substrates with various pore sizes (Supplementary Fig. S3). Although the CD spectra of bare AAO templates exhibit huge CD signal (nearly 100 mdeg) due to the overestimated scattering contribution of transmitted CD measurement, which is common in nanostructured materials with definite spatial orientations,³⁵ the empty AAO templates show no CD signal in the wavelength range above 350 nm. This implies that the effect of the optical anisotropy from the bare AAO templates can be completely excluded. Therefore, the observed sign conversion and spectral shape change of the Cotton effect imply that spatial confinement by AAO templates might induce the coupling modulation between the chiral chromophores, resulting in the reconstruction of the chiroptical band structure.

(References; 35 is added)

35. Duan YY, *et al.* Optically Active Nanostructured ZnO Films. *Angew. Chem. Int. Edit.* **54**, 15170-15175 (2015).

(Supplementary; Fig. S3 was added)

Fig. S3. Optical activity of bare AAO substrates with various pore size. a. Linear absorption spectra and b. CD spectra obtained from the empty AAO substrate without chiral 2D OIHPs.

Comment 6:

The CD spectra of perovskite grown on planar substrates exhibit unisgnated CD signal at the first excitonic transition. Can the author explain the reason for uncommon behavior (not bisgnated Cotton effect but unisgnated CD)?

Author's Response:

We thank the reviewer for the comment. However, it is not uncommon behavior to exhibit unisgnated CD signal at the first excitonic transition in chiral OIHPs.[R15,R16,R17] In chiral OIHPs, the different absorption between left-handed circularly polarized light (LCP) and right-handed circularly polarized light (RCP) occurs when the degeneracy of electronic state is slightly lifted by the perturbation due to the chirality of the chiral organic molecules; energy state with one spin can go relatively higher energy compared to the other spin state. In this case, the intensity of the CD spectra can be estimated by rotational strength, which is directly proportional to the scalar product as expressed by equation (1)

$$R_{ab} \approx |\mu_{ab}| \cdot |m_{ab}| \cos \theta \quad (1)$$

where R_{ab} describes the rotational strength related to the transition from state a to b , μ_{ab} , m_{ab} , and θ are the electrical transition dipole moment (ETDM), magnetic transition dipole moment (MTDM), and angle between the ETDM and MTDM. However, the magnitude of transition dipole moment (whether ETDM or MTDM) as well as angle between them are not the same at given wavelength where each transition occurs (absorption of LCP or RCP). Therefore, positive and negative Cotton effect centered on the zero cross point cannot be completely symmetrical. For example, as shown in Fig. R7 (reproduced from Fig. 1a in our manuscript), the area of positive Cotton effect was larger than those of negative Cotton effect. In this case, one might then imagine the case where the total intensity of Cotton effect (both positive and negative) is gradually decreased. At the certain point, the negative Cotton effect firstly disappeared in the CD spectra, while the positive Cotton effect still remained. This is consistent with our previous report on spectral change in CD signal induced by the halide composition engineering.[R15]

Figure R6. Asymmetric behavior of CD signal. Typical CD spectra obtained from the R-configuration chiral 2D OIHPs with two asymmetric Cotton effect (small positive Cotton effect *versus* large negative Cotton effect).

References cited in this response:

- R15 Ahn, J., Ma, S., Kim, J. Y., Kyhm, J., Yang, W., Lim, J. A., *et al.* Chiral 2D Organic Inorganic Hybrid Perovskite with Circular Dichroism Tunable Over Wide Wavelength range. *J. Am. Chem. Soc.* **142**, 4206–4212 (2020).
- R16 Ma S, Ahn J, Moon J. Chiral Perovskites for Next-Generation Photonics: From Chirality Transfer to Chiroptical Activity. *Adv. Mater.* **33**, 2005760 (2021).
- R17 Chen C, *et al.* Circularly Polarized Light Detection Using Chiral Hybrid Perovskite. *Nat. Commun.* **10**, 1927 (2019).

Comment 7:

The authors are suggested to provide the results of rac-MBA samples grown on both planar and AAO substrate for comparisons.

Author's Response:

Thank you for important comment. We understood that the reviewer ask us to rule out the possible effect of AAO substrate on the chiroptical activity of 2D OIHPS by comparing optical activity of *R*- and *S*-configurations with that of racemic compound. Therefore, during the revision work, we additionally conducted CD measurement and XRD analysis with rac-MBA₂PbBr_xI_{4-x} ($x=0.325$) to clearly confirm the effect of nanoconfined growth on the chiroptical activity of chiral 2D OIHPS. As shown in Fig. R7, the CD spectra of racemic samples do not exhibit any notable chiroptical response in the range of 425–525 nm regardless of the grown substrate conditions (*e.g.*, planar, 66 nm-, 100 nm-, 112 nm-pore sized AAO templates). Considering the aforementioned CD results of bare AAO substrates (in our Response to **Comment 5** for ***Reviewer 2***), we verify that unprecedented chiroptical phenomena of nanoconfined chiral 2D OIHPS in AAO substrates is not due to the optical anisotropy of the AAO substrate itself; rather, the amplified CD signal, sign conversion of Cotton effect, and huge difference in CPPL intensity are originated from the micro-strain induced by nanoconfined growth in AAO templates. Furthermore, the XRD spectra of racemic samples also confirmed that the nanoconfined growth in AAO templates do not affect the crystallization process without impairing the preferred orientation and quality of chiral 2D OIHPS (Fig. R8).

Fig. R7. Optical properties of racemic OIHPs grown in different substrate conditions. a. CD spectra and b. linear absorption spectra of racemic compound $R\text{-MBA}_2\text{PbBr}_x\text{I}_{4-x}$ ($x=0.325$) grown in different substrate conditions.

Fig. R8. XRD spectra of racemic compound $R\text{-MBA}_2\text{PbBr}_x\text{I}_{4-x}$ ($x=0.325$) grown in different substrate conditions.

Revision made (colored in blue):

(in Page 10-11)

… Interestingly, the abnormal chiroptical behavior (*i.e.*, amplified CD signal, sign conversion, and spectra shape change in the Cotton effect) was also observed for the chiral 2D OIHPs thin films with all the compositions (from $x = 0.350$ to 0.400 as well as $x = 0.325$) (Supplementary Fig. S6). We also conducted the CD measurement and XRD analysis with racemic $\text{MBA}_2\text{PbBr}_x\text{I}_{4-x}$ ($x=0.325$) to clearly confirm the effect of nanoconfined growth on the chiroptical activity of chiral 2D OIHPs. As shown in Fig. S10 (Supplementary), the CD spectra of racemic compound do not exhibit any notable chiroptical response in the range of 425–525 nm regardless of the grown substrate conditions (*e.g.*, planar, 66 nm-, 100 nm-, 112 nm-pore sized AAO templates), implying that the abnormal chiroptical behavior in chiral 2D OIHPs is not due to the optical anisotropy of the AAO substrate itself, but rather resulted from the promoted chirality transfer from organic spacer cations to achiral inorganic frameworks. The XRD spectra of racemic samples with different AAO templates condition also show no noticeable difference, suggesting that the nanoconfined growth in AAO templates do not influence on the crystallization process without impairing the preferred orientation and quality of chiral 2D OIHPs (Supplementary Fig. S11). These observations imply that the chirality transfer could be effectively promoted by nanoconfined growth in AAO templates, resulting in the modulation of chiroptical activity in chiral 2D OIHPs even at higher bromide composition.

(in Supplementary Information; Fig. S10 and Fig. S11 were added)

Fig. S10. Optical properties of racemic OIHPs grown in different substrate conditions. **a**, CD spectra and **b**, linear absorption spectra of racemic compound $R\text{-MBA}_2\text{PbBr}_x\text{I}_{4-x}$ ($x=0.325$) grown in different substrate conditions.

Fig. S11. XRD spectra of racemic compound $R\text{-MBA}_2\text{PbBr}_x\text{I}_{4-x}$ ($x=0.325$) grown in different substrate conditions.

Comment 8:

I noticed that the authors used circularly polarized light to excite samples for CPPL measurement. Under such case, other effect such as Rashba effect might also contribute to the measured CPPL, which is not the intrinsic CPPL due to the chirality transfer.

Author's Response:

We thank the reviewer for the comment. It is expected that Rashba effect or Rashba-Dresselhaus effects can occur under an external magnetic field even in the achiral OIHPs. As the reviewer commented, similar to the external magnetic field applied condition, excitation by the circularly polarized light also can lift the degeneracy of spin state due to the large spin-orbit coupling (SOC) of OIHPs. If the huge difference PL emission intensity between the RCP and LCP in the CPPL measurement is originated from the Rashba effect and the induced spin degeneracy rather than due to the intrinsic chirality transfer phenomena, the different emission rates of RCP and LCP should be also detected in racemic compound under the same measurement condition. Therefore, to clarify the origin of different emission rate of RCP and LCP, the CPPL measurement was also performed in the same manner (*i.e.*, using circularly polarized laser as an excitation source) for racemic compounds grown on AAO templates with pore size of 100 nm.

As shown in Fig. R9a, the racemic compounds grown in AAO templates with 100 nm pore do not show any different emission behavior between the RCP and LCP. For clear comparison, the CPPL spectra of R-configuration grown in AAO templates with 100 nm (Fig. 4b in our manuscript) is also presented as Fig. R9b with same scale. The CPPL spectra of racemic compounds grown on AAO templates suggested that the Rashba effect and coherent lifting of the spin degeneracy by the circularly polarized light did not occur in the absence of chirality transfer phenomena (*i.e.*, in the absence of chiral organic molecules). This implies that enhanced asymmetric factor of CPPL (g_{CPPL}) in chiral 2D OIHPs results from the facilitated chirality transfer phenomena rather than Rashba effect itself (by the excitation using circular polarized light). Very recently, Mitzi group found that the Rashba-Dresselhaus spin-splitting is a consequence of the chirality transfer phenomena.[R18] Based on their findings and experimental results, the other effects such as Rashba effect or Rashba-Dresselhaus spin-splitting (whether it occurs or not), which might contribute to the measured CPPL, cannot be explained separately. Rather, such effects should be included as a consequence of the chirality transfer phenomena facilitated by nanoconfined growth in AAO templates.

Fig. R9. a, CPPL spectra of racemic-MBA₂PbI_{4(1-x)}Br_{4x} (x=0.325) grown in AAO template with 100 nm pore size. **b**, CPPL spectra of R-MBA₂PbI_{4(1-x)}Br_{4x} (x=0.325) grown in AAO template with 100 nm pore size.

References cited in this response:

R18 Jana MK, *et al.* Structural descriptor for enhanced spin-splitting in 2D hybrid perovskites. *Nat. Commun.* **12**, 4982 (2021).

Revision made (colored in blue):

(in Page 15-16)

To exclude the effect of Rashba splitting on CPPL spectra, which might arise from the experimental procedure (because the excitation by the circularly polarized light can lift the degeneracy of spin state due to the large spin-orbit coupling (SOC) of OIHPs), and to clarify the origin of different emission rate of RCP and LCP, the CPPL measurement was also performed in the same manner (using circular polarized light as a excitation source) for racemic compounds grown on AAO templates with pore size of 100 nm. As shown in Fig. S14 (Supplementary), the racemic compounds grown in AAO templates do not show any different emission behavior between the RCP and LCP. The CPPL spectra of racemic compounds grown on AAO templates suggested that the Rashba effect and coherent lifting of the spin degeneracy by the circularly polarized light did not occur in the absence of chirality transfer phenomena (*i.e.*, in the absence of chiral organic molecules). This implies that enhanced asymmetric factor

of CPPL (g_{CPPL}) in chiral 2D OIHPs results from the facilitated chirality transfer phenomena rather than Rashba effect itself (by the excitation using circular polarized light). Very recently, Mitzi group found that the Rashba-Dresselhaus spin-splitting is a consequence of the chirality transfer phenomena.⁴⁴ Based on their findings and experimental results, the other effects such as Rashba effect or Rashba-Dresselhaus spin-splitting (whether it occurs or not), which might contribute to the measured CPPL, cannot be explained separately. Rather, such effects should be included as a consequence of the chirality transfer phenomena facilitated by nanoconfined growth in AAO templates.

(Supplementary; Fig. S14 was added)

Fig. S14. CPPL spectra of $\text{racemic-MBA}_2\text{PbI}_{4(1-x)}\text{Br}_{4x}$ ($x=0.325$) grown in AAO template with 100 nm pore size.

Comment 9:

Some important recent works on chiral 2D perovskites are missing. The authors are suggested to introduce and cite them in the revised manuscript.

Author's Response:

We thank the reviewer for the comment. We have added the references of important recent works on chiral 2D perovskites in our manuscript.

Revision made (colored in blue):

(References were added)

2. Ma, S., Ahn, J., Moon, J. Chiral Perovskites for Next-Generation Photonics: From Chirality Transfer to Chiroptical Activity. *Adv. Mater.* **33**, 2005760 (2021).
37. Lu Y, *et al.* Spin-Dependent Charge Transport in 1D Chiral Hybrid Lead-Bromide Perovskite with High Stability. *Adv. Funct. Mater.* **31**, 2104605 (2021).
44. Jana MK, *et al.* Structural Descriptor for Enhanced Spin-Splitting in 2D Hybrid Perovskites. *Nat. Commun.* **12**, 4982 (2021).

<Reviewer 3>

I was excited by the title and abstract of this manuscript. The authors attempt to address a difficult and important problem (the origin/mechanism or chiro-optical phenomena in chiral metal halide systems) and the abstract promises analysis in terms of a very interesting new idea, namely, multipolar interactions involving the chiral cations in these systems. Not mentioned in the abstract was the technique used of growth in nano-pore substrates; this is a very clever idea. However after reading the work I find that the claims made are not well supported by the work, it seems very preliminary. In addition, the manuscript is missing some essential information and the presentation needs to be substantially improved. Here I summarize what I see as the major technical issues; further below I will make comments on the presentation of the manuscript and list other minor technical issues I noted while reading the manuscript.

Remark:

We would like to thank the reviewer for evaluating our work as an interesting new idea to clarify the origin/mechanism of chiroptical activities in metal halide systems. Our response to the reviewer's comments can be found below.

Comment 1:

Analysis of micro-strain of films grown on AAO nanopore substrates. Fig 2b shows the micro-strain determined for films grown on AAO nanopore substrates. The authors describe in the SI the procedure whereby these substrates were fabricated. But no-where is there a definition of an AAO substrate with pore size of 0 nm as shown in Fig 2b.

a. I assume that a pore size of 0 nm means "there are no pores".

b. *How then is it possible that the micro-strain shown for the 0nm pore size in Fig 2b is -6%?*

It is difficult to believe that this is a correct analysis. If I understand correctly what the 0 nm pore size means, then this sample comprises a planar film. Given the weak non-covalent interaction between layers it is highly unlikely that these films are epitaxial so the -6% strain of this "0 nm pore" sample is quite a mystery. Indeed, there is no correlation between the derived micro-strain and the nano-pore size (see plot in Fig2b). The Authors' own description on line 167, is that the nano-pore size is of "unexpected irrelevance".

c. Yet the interpretation of the XRD pattern in terms of the micro-strain is a foundational

element of the author's interpretation and claims.

Author's Response:

We appreciate the reviewer's comment. We understood that the reviewer asks us to provide more detail information of the micro-strain analysis and to demonstrate the validity of experimental results from the modified Williamson-Hall methods. First of all, as the reviewer assumed, the mention "pore size of 0 nm" is correct to mean "there are no pores". The sample with pore size of 0 nm represents thin films of chiral 2D OIHPs grown on planar substrates without porous template. In order to eliminate the possible confusion and misunderstanding, we have added the definition describing the chiral 2D OIHPs grown on "substrate with pore size of 0 nm".

For the chiral 2D OIHPs grown on substrate with pore size of 0 nm (*i.e.*, planar substrate), we observed the presence of -6% micro-strain in the lattice of OIHPs, even though the thin films were grown freely without spatial confinement. Reviewer pointed out, "*How then is it possible that the micro-strain shown for the 0 nm pore size in Fig 2b is -6%?*" However, it is not uncommon that the local lattice strain exists in OIHPs thin films grown on planar substrate condition due to the lattice mismatch, atomic size misfit, mismatch of thermal expansion, or lattice defects.[R19, R20, R21] **Since we have calculated the degree of micro-strain by the comparison with the strain-free single crystalline data (*i.e.*, MBA_2PbI_4 single crystal as standard material),** the obtained value of -6% micro-strain from the thin films of chiral 2D OIHPs grown in planar substrate is reasonable. We are sorry that the experimental procedure of micro-strain analysis in Supplementary did not provide sufficient information on how the micro-strain was calculated. Therefore, we have added and modified the experimental procedure to include the corresponding calculation process and single crystalline data in the revised manuscript.

Furthermore, as the reviewer pointed out, we mentioned the calculated micro-strain values exhibit "unexpected irrelevance" with the pore size of AAO templates in our manuscripts. However, when we use the term of "unexpected irrelevance", it was not intended that there is no correlation between the derived micro-strain and the nano-pore size. Since the calculated micro-strain values are inversely proportional to the pore size of AAO templates in our previous reports regarding 3D OIHPs, the expression "unexpected irrelevance" rather just comprehensively implies unexpected behavior (not inversely proportional to the pore size of AAO templates). For clarification, the expression "unexpected irrelevance" has been replaced

with “unexpected non-linear behavior” in our revised manuscript.

To elucidate the relationship between the pore size and resulting micro-strain, the calculation of the external stress exerted by the pore wall at each spatial confinement condition as a function of pore sizes must be preceded. However, as the optimal π - π stacking structure of 2D OIHPs varies with the given spatial restriction condition, it is hard to directly calculate the magnitude of external stress exerted by the pore wall. Therefore, to provide direct experimental evidence of imposed strain induced by pore wall and to correlate micro-strain values depending upon the pore size of AAO templates, we additionally conducted the PL measurement of chiral 2D OIHPs grown on different pore-sized AAO templates (please see our Response to **Comment 3** for *Reviewer 2*). Remarkably, the plot of PL emission energy also exhibits zigzag tendency as a function of the template pore sizes (as shown in Fig. R4), which is similar to that of the calculated micro-strain plot. Such a coincident behavior implies that the shift of PL emission not only provides the direct evidence of micro-strain imposed by AAO templates but also demonstrates the validity of the modified Williamson-Hall methods. Consequently, the correlation between the deviation of chiroptical activities and the resulting micro-strain induced by the nanoconfined growth, which is a core objective of this paper, was well supported by experimental results and strain analysis.

References cited in this response:

- R19 Li W, et al. Phase Segregation Enhanced Ion Movement in Efficient Inorganic CsPbIBr₂ Solar Cells. *Adv. Energy Mater.* **7**, 1700946 (2017).
- R20 Tsai H, et al. Light-Induced Lattice Expansion Leads to High-Efficiency Perovskite Solar Cells. *Science* **360**, 67-70 (2018).
- R21 Zheng XJ, Wu CC, Jha SK, Li Z, Zhu K, Priya S. Improved Phase Stability of Formamidinium Lead Triiodide Perovskite by Strain Relaxation. *ACS Energy Lett.* **1**, 1014-1020 (2016).

Revision made (colored in blue):

Description of detail experimental procedure of strain-analysis and definition describing the chiral 2D OIHPs grown on planar substrate (with 0 nm pore size) has been added.

(in page 4)

... The same precursor solutions were deposited on a glass substrate (hereafter denoted as planar or substrate with 0 nm pore size) and AAO templates with different pore sizes

(in page 7)

... For the accurate assessment of the micro-strain imposed by AAO templates, the XRD patterns for chiral 2D OIHPs ~~with various halide composition (e.g., MBA_2PbI_4 as well as $\text{MBA}_2\text{PbI}_{4(1-x)}\text{Br}_{4x}$ with a composition of $x = 0.325, 0.350$)~~ with single iodine halide composition (*i.e.*, MBA_2PbI_4) grown in different substrate conditions were obtained. The degree of micro-strain values imposed by different substrate conditions (planar; *i.e.*, substrate with 0 nm pore size, AAO template with pore size of 66 nm, 100nm, and 112 nm) were carefully calculated by the comparison with the strain-freely grown single crystalline MBA_2PbI_4 . Detailed information on strain analysis is provided in Supplementary Note 2 and Supplementary Fig. S4.

(in page 9)

... the magnitude of micro-strain rather revealed a zigzag tendency, as shown in Fig. 2b, than a linear dependency on the pore size. It is worth to note that the presence of -6% micro-strain in the lattice of OIHPs was observed in 0 nm pore size condition, even though the thin films were freely grown on planar substrate without spatial confinement. However, it is not uncommon that the local lattice strain exists in OIHPs thin films grown on planar substrate due to the lattice mismatch, atomic size misfit, mismatch of thermal expansion, or lattice defects.^{39,40,41} As mentioned above, we have calculated the degree of micro-strain by the comparison with the strain-free single crystalline data (*i.e.*, MBA_2PbI_4 single crystal as standard material), the obtained value of -6% micro-strain from the thin films of chiral 2D OIHPs grown in planar substrate is reasonable.

(in caption of Fig. 2)

Fig. 2. Estimation of micro-strain in chiral 2D OIHPs and stacking conformation variation induced by micro-strain. **a**, Schematic illustration of unit cell of R-MBA₂PbI₄ and induced change of *d*-spacing between the (002) planes. **b**, Magnitude of calculated micro-strain values as a function of AAO template pore size. The inset represents the schematic illustration of nanoconfined growth of chiral 2D OIHPs in AAO template. The black arrow indicates the change in the lattice parameter along the *c*-axis. **c**, Deconvolution results obtained from the CD spectra. **d**, Plot of Davydov splitting values versus AAO template pore size. Error bars indicate the standard deviation. *Note that the 0 nm pore size condition represents chiral 2D OIHPs grown on planar substrate without template.*

For clarification, the expression “unexpected irrelevance” has been replaced with “unexpected non-linear behavior” or with appropriate term in context.

(in page 9)

The ~~unexpected irrelevance~~ unexpected non-linear behavior of calculated micro-strain values for AAO templated chiral 2D OIHPs as a function of pore size can be possibly understood by the ...

Comment 2:

CD and dissymmetry spectra are shown in Fig 1 and in the SI, and details of the deconvolution procedure they use to assign transitions are described in the SI (Suppl Note 4). However, no absorption or extinction spectra are shown. It is problematic to evaluate statements about the nature of the spectral features in the CD spectra and how they are de-convolved and assigned without seeing these spectra.

Author's Response:

We thank the reviewer for the critical comment on the CD and dissymmetry spectra together with deconvolution procedure to assign the transition. As the reviewer commented, to assign the signal in the CD spectra to different excitonic transition level, the absorption or extinction spectra should be presented.[R22] We are sorry that the absorption spectra of chiral 2D OIHPs grown in different substrates conditions are missing. In Fig. R10, the CD and corresponding absorption spectra of chiral 2D OIHPs with the composition of $R\text{-MBA}_2\text{PbI}_{4(1-x)}\text{Br}_{4x}$ ($x = 0.325$) are presented. Regardless of the grown substrate conditions, the CD spectra of chiral 2D OIHPs exhibit two CD responses (whether bi-signate or uni-signate) in the wavelength range of 350–550 nm. Although the chiral OIHPs exhibit multiple excitonic transition behaviors (two CD signals), the intensive excitonic transition behavior occurs near the first extinction band edge. In addition, the chiroptical activity of chiral 2D OIHPs due to the chirality transfer process occurs around 475 nm, which is associated with the transition in inorganic framework. Therefore, CD spectrum fitting procedure was mainly focused on the first extinction band edge range. **As a result, the intense CD signal at longer wavelength can be assigned to the first excitonic transition in absorption spectra (red dot line in absorption spectra), which is consistent with the previous report.**[R22] We have added the figure in Supplementary to provide corresponding extinction spectra.

Fig. R10. Deconvolution results obtained from the CD spectra (upper panel) and linear absorption spectra (lower panel) of *R*-MBA₂PbI_{4(1-x)}Br_{4x} chiral OIHPs ($x = 0.325$) grown on different substrate conditions; **a**, planar substrate, **b**, AAO template with a 66 nm pore size, **c**, AAO template with a 100 nm pore size, and **d**, AAO template with a 112 nm pore size. The solid purple line represents obtained CD spectra from the chiral 2D OIHPs. The red and blue dot-line in CD spectra (upper panel) indicate the absorption of LCP and RCP, respectively.

References cited in this response:

- R22 Ben-Moshe A, Teitelboim A, Oron D, Markovich G. Probing the Interaction of Quantum Dots with Chiral Capping Molecules Using Circular Dichroism Spectroscopy. *Nano Lett.* **16**, 7467-7473 (2016).

Revision made (colored in blue):

(Supplementary; Corresponding extinction spectra were presented in Fig. S12)

Fig. S12. Deconvolution results obtained from the CD spectra (upper panel) and linear absorption spectra (lower panel) of R -MBA₂PbI_{4(1-x)}Br_{4x} chiral OIHPs ($x = 0.325$) grown on different substrate conditions; a, planar substrate, b, AAO template with a 66 nm pore size, c, AAO template with a 100 nm pore size, and d, AAO template with a 112 nm pore size. The solid purple line represents obtained CD spectra from the chiral 2D OIHPs. The red and blue dot-line in CD spectra indicate the absorption of LCP and RCP, respectively.

Comment 3:

The analysis in terms of Davydov splitting of the MBA cation transitions is a concept based in Frenkel exciton theory and it is a very interesting idea.

a. But the MBA cation exciton transition is at $\sim 260\text{nm}$ (See Fig S3 in DOI: 10.1039/c7mh00197e from this group); it is never explained adequately here how this transition would be relevant to the exciton transition at $\sim 475\text{nm}$ which is associated with the lead-iodide framework .

b. To be clear, the authors claim on line 275 et seq that Fig 3b “suggests” that the MBA HOMO forms a “resonant state with the inorganic PbI_4 framework” but this is an assertion, and it is certainly not proven. The assertion does not seem to be backed up by the energy level calculations in Fig 3 and it contradicts what the Authors state about the frontier orbitals being associated with the inorganic sub-lattice (in line 262-262).

c. If Davydov splitting in nano-confined MBAPbX_4 is responsible for the emergence of bi-signate CD spectra, as claimed here, why are bi-signate CD spectra (sometimes) observed in planar thin film samples as reported for example in DOI: 10.1039/c7mh00197e from this group?

Author’s Response:

As the reviewer’s pointed out (in **Comment 3a**), Davydov splitting of exciton transition at $\sim 475\text{ nm}$, presented in Fig.1a in our manuscript, is not originated from the MBA cation exciton transition, but rather associated with the transition in lead-iodide framework. Because the MBA cation exciton transition occurs at $\sim 260\text{ nm}$ (the article suggested by reviewer (DOI: 10.1039/c7mh00197e) is the work of our group), it is far from the wavelength region where exciton transition occurs ($\sim 475\text{ nm}$). Therefore, **the calculated Davydov splitting values should be interpreted as a result of excitonic transition behavior in the inorganic framework where the chirality is induced by the chirality transfer phenomena.**

In this manner, the change in Davydov splitting value according to the pore size of AAO templates signify that the efficiency (or degree) of chirality transfer can greatly vary depending upon the imposed micro-strain. Consequently, it can be concluded that the variation of Davydov splitting extracted from the exciton transition at $\sim 475\text{ nm}$ is the result of the stepwise process: i) conformational stacking order of chiral organic cations (*i.e.*, angle and length between the MBA_1 and MBA_2) changes due to the imposed micro-strain, ii) the electronic interaction between the chiral organic molecules and achiral inorganic framework

was enhanced (or reduced), iii) the chirality transfer from the chiral organic cation to inorganic framework was promoted (or suppressed). Since the chiral organic spacer can promote efficient chirality transfer by inducing symmetry-breaking distortion into the inorganic framework *via* hydrogen-bonding interaction,[R23,R24] we suspect that the pore size-dependent behavior of Davydov splitting might be originated from the enhanced electronic interaction (hydrogen-bonding interaction) between the chiral organic spacer and achiral inorganic framework.

Furthermore, as the reviewer mentioned in **Comment 3b**, the electronic states at E_v and E_c of chiral 2D OIHPs mainly result from the strong antibonding interaction between the Pb *s*- and I *p*-orbitals, and from nonbonding Pb *p*-orbital, respectively. Although the HOMO level of MBA becomes closer to VBM of PbI₄ inorganic frameworks as the compressive strain imposed (Fig. 3c and d in our manuscript), it would be overstated that HOMO level of MBA forms a resonant state with PbI₄ frameworks. Therefore, on the basis of our newly obtained findings (described below) during the revision, we propose the modified working mechanism about the origin of chiroptical activities as **facilitated by organic-to-inorganic chirality transfer *via* hydrogen-bonding interaction changes between the chiral organic spacer and inorganic framework.**

To support our scenario of facilitated chirality transfer phenomena, we additionally analyzed structural properties of MBA₂PbI₄ as a function of micro-strain. To correlate the imposed micro-strain and the degree of chirality transfer, the specific structural parameters such as the intra-octahedron distortions (*i.e.* Δd and σ^2) were measured from our DFT-optimized structures. Δd represents the bond length distortion defined as $\Delta d = \sum (d_i - d_0)^2 / 6d_0^2$ (d_i implies the six Pb–I bond lengths and d_0 is the average Pb–I bond length), and σ^2 is the bond angle variance defined as $\sigma^2 = \sum_{i=1}^{12} (\theta_i - 90)^2 / 11$, where θ_i denotes the individual *cis* I–Pb–I bond angles. Remarkably, in the compressive micro-strain imposed region (as highlighted with yellow color in Fig. R11), both Δd and σ^2 increased sharply, implying that the degree of intra-octahedron distortion becomes larger when the lattice shrinkage occurs along the c-axis.

To elucidate how the micro-strain (nanoconfined growth in AAO templates) influences the degree of the intra-octahedron distortion (degree of the chirality transfer), we have also calculated the hydrogen bonding length between NH₃⁺ groups of chiral organic spacer and nearest iodine atom of inorganic framework. It is worth noting that there are four different distinguishable hydrogen bondings in the unit cell of MBA₂PbI₄ (denoted as HB_{top1}, HB_{top2}, HB_{bot1}, and HB_{bot2}). Interestingly, as shown in Fig. R12, **the asymmetric nature of hydrogen bonding (variance and difference between the hydrogen bonding lengths) was amplified**

when the lattice shrinkage occurs along the c-axis (region of negative uniaxial strain and positive biaxial strain). The DFT calculation results support that the asymmetric behavior of hydrogen bonding between the chiral organic molecules and inorganic framework can be amplified depending on the degree of micro-strain, which can effectively promote the chirality transfer process and larger chiral distortion in inorganic framework.

Based on the above discussion and theoretical calculation results, we suggest that the origin of unprecedented chiroptical activities in nanoconfined chiral 2D OIHPs can be explained by the **efficient chirality transfer promoted by asymmetric hydrogen-bonding interaction between the chiral spacer and inorganic frameworks**. We guarantee that our assertion is fully supported by the related calculation results. Addition of Fig. R11 and R12 and related calculation results into the revised manuscript will clarify the relationship between the micro-strain and induced chiroptical activities in chiral 2D OIHPs, while satisfying the reviewer's concern. We gratefully thank the reviewer for helpful comments which significantly improves the quality of our manuscript.

In addition, we believed that the comment of “nano-confined MBAPbX₄ is responsible for the emergence of bi-signate CD spectra, as claimed here, why are bi-signate CD spectra (sometimes) observed in planar thin film samples as reported for example in DOI: 10.1039/c7mh00197e from this group?” likely results from the reviewer's misunderstanding of our explanation. First of all, the bi-signate CD spectra behavior is commonly observed in the chiral 2D OIHPs. The article suggested by reviewer (DOI: 10.1039/c7mh00197e) is the work of our group. In a subsequent study (*J. Am. Chem. Soc.* 2020, 142, 4206–4212), we already observed the conversion of Cotton effect from bi-signate to uni-signate in MBAPbI₄(1-x)Br_{4x} as the bromide ratio increased. As we mentioned above (please see our Response to **Comment 6** for ***Reviewer 2***), since the positive and negative area of Cotton effect cannot be completely symmetrical, the one Cotton effect can be firstly disappeared in the CD spectra, while the opposite Cotton effect still remained, resulting in the uni-signate Cotton effect in CD spectra. Based on our observation and calculation results, this conversion of Cotton effect (from bi-signate to uni-signate) can be also interpreted as change of chirality transfer efficiency by modulation of hydrogen-bonding interaction; *i.e.*, the substitution of iodide anion with bromide anion could weaken the hydrogen-bonding interaction between the chiral organic molecules and inorganic framework. Therefore, the opposite conversion behavior (from uni-signate to bi-signate) of Cotton effect observed in nanoconfined growth should be interpreted

in the same manner; *i.e.*, imposed micro-strain can facilitate the chirality transfer process by enhancing the hydrogen-bonding interaction between the chiral organic spacer and inorganic framework.

Fig. R11. Structural characteristics of chiral OIHPs. **a.** Schematic illustration of specific structural parameters measured from the optimized structure of our DFT-calculation. Corresponding bond length distortion and bond angle variance under uniaxial strain (**b.** and **d.**) and biaxial strain (**c.** and **e.**), respectively. The bright yellow region denotes the range of lattice shrinkage along the *c*-axis.

Fig. R12. Asymmetric hydrogen bonding interaction in chiral OIHPs. a. Illustration of hydrogen bonding between the chiral organic spacer and inorganic framework in the optimized structure based on our DFT-calculation. b. Corresponding hydrogen bonding length as a function of the imposed strain. The bright yellow region denotes the range of lattice shrinkage along the c-axis.

References cited in this response:

- R23 Jana MK, et al. Organic-to-Inorganic Structural Chirality Transfer in a 2D Hybrid Perovskite and Impact on Rashba-Dresselhaus Spin-Orbit Coupling. *Nat. Commun.* **11**, 4699 (2020).
- R24 Jana MK, et al. Structural Descriptor for Enhanced Spin-Splitting in 2D Hybrid Perovskites. *Nat. Commun.* **12**, 4982 (2021).

Revision made (colored in blue):

(in page 11)

... while \vec{e}_i is the corresponding unit vector. It is worth to noting that the abnormal chiroptical behaviors (*i.e.*, amplified CD signal, sign conversion, and spectra shape change in the Cotton effect) observed in AAO templated chiral 2D OIHPs occur at ~ 475 nm (near the band edge extinction of chiral 2D OIHPs), which is far from the wavelength region where the exciton transition of chiral MBA^+ cations occurs (~ 260 nm). Therefore, the CD signal of chiral 2D OIHPs in Fig. 1a should be interpreted as a result of excitonic transition behavior in the lead halide inorganic framework where the chirality was induced by the chirality transfer phenomena. ~~Since the MBA^+ -spacer cations (MBA_1 and MBA_2) are the only possible candidates for generating the CD signal (because the transition dipole moment of two MBA^+ cations cannot be located in the coplanar due to their inherent chirality), it is logical to conclude that the conformational π - π stacking variation of chiral organic spacer MBA^+ cations is the origin of abnormal chiroptical behaviors (*i.e.*, amplified CD signal, sign conversion, and spectra shape change in the Cotton effect) observed in AAO templated chiral 2D OIHPs. It is logical to conclude that the efficiency (or degree) of the chirality transfer can greatly vary depending upon the imposed micro-strain. In this manner, we propose the stepwise chirality transfer mechanism to interpret the unprecedented chiroptical activity of chiral 2D OIHPs in AAO templates: i) conformational stacking order of chiral organic cations (*i.e.*, angle and length between the MBA_1 and MBA_2) changes due to the imposed micro-strain, ii) the electronic interaction between the chiral organic molecules and achiral inorganic framework was enhanced (or reduced), iii) the chirality transfer from the chiral organic cation to inorganic framework was promoted (or suppressed).~~

(in page 14)

To verify the relationship between the imposed micro-strain and the efficiency of chirality transfer in chiral 2D OIHPs ~~π - π stacking conformation variation and the electronic state of chiral 2D OIHPs~~, we evaluated the Davydov splitting values from the deconvoluted CD spectra where a bisignate CD signal appeared around the extinction band edge λ_0

(in page 14)

... ~~Because all the transition dipole moments in chiral 2D OIHPs have an identical magnitude regardless of the grown substrate types (because of the same concentration of MBA^+ cations in chiral 2D OIHPs), the variations in Davydov splitting imply the changes in both the distance~~

between the two transition dipoles and the direction of corresponding unit vectors.³⁴ Therefore, such a coincident tendency (*i.e.*, zigzag tendency in both calculated micro-strain and Davydov splitting) can support that unprecedentedly observed chiroptical conversion behavior in AAO templated chiral 2D OIHPs results from the micro-strain induced variation in the π - π stacking conformation (*i.e.*, distance and angle changes between MBA_{1-} and MBA_{2-} as confirmed by DFT calculation). Such a coincident tendency (*i.e.*, similar zigzag tendency in calculated micro-strain, PL emission shift as well as Davydov splitting) can support that unprecedentedly observed chiroptical conversion behavior in AAO templated chiral 2D OIHPs results from facilitated chirality transfer phenomena from chiral organic cations (MBA^+) to achiral inorganic framework (lead halide) induced by the micro-strain.

(in page 16-18: the statements related to the electronic structure from the DFT calculation were deleted)

To support our scenario of facilitated chirality transfer phenomena by imposed micro-strain, we analyzed structural properties of MBA_2PbI_4 as a function of micro-strain. It is worth mentioning that $\text{MBA}_2\text{PbI}_{4(1-x)}\text{Br}_{4x}$ thin films exhibit sharp XRD diffraction peaks assignable to the (00 l) planes regardless of the growing substrates and bromide composition, indicating the highly preferred orientation along the c -axis (Fig. 1d and Supplementary Fig. S13). The observed peak shift in the iodide-determinant phase (toward higher 2 θ degree) suggests that confined growth induces lattice shrinkage along the c -axis. In addition, the horizontal growth of chiral 2D OIHPs is effectively inhibited by the pore wall (parallel to substrate) (as shown in Fig. S2 Supplementary), the direction of imposed micro-strain is out-of-plane direction (*i.e.*, parallel to pore wall). Consequently, we need to focus on the shrinkage range in the out-of-plane direction (*i.e.*, negative uniaxial strain and positive biaxial strain; yellow region in Fig. 3b and c) to properly interpret our DFT calculations. To correlate the imposed micro-strain and the degree of chirality transfer, the specific structural parameters such as the intra-octahedron distortions (*i.e.*, Δd and σ^2) were measured from our DFT-optimized structures as has been suggested by Mitzi group.⁴³ Δd represents the bond length distortion defined as $\Delta d = \sum (d_i - d_0)^2 / 6d_0^2$ (d_i implies the six Pb-I bond lengths and d_0 is the average Pb-I bond length), and σ^2 is the bond angle variance defined as $\sigma^2 = \sum_{i=1}^{12} (\theta_i - 90)^2 / 11$, where θ_i denotes the individual *cis*-I-Pb-I bond angles (Fig. 3a). Remarkably, in the compressive micro-strain imposed region (as highlighted with yellow color in Fig. 3b), both Δd and σ^2 increased sharply,

implying that the degree of intra-octahedron distortion becomes larger when the lattice shrinkage occurs along the *c*-axis.

To elucidate how the micro-strain (nanoconfined growth in AAO templates) influences the degree of the intra-octahedron distortion (efficiency of the chirality transfer), we have also calculated the hydrogen bonding length between NH_3^+ groups of chiral organic spacer and nearest iodine atom of inorganic framework (Fig. 3a). Notably, there are four different distinguishable hydrogen bondings in the unit cell of MBA_2PbI_4 (denoted as HB_{top1} , HB_{top2} , HB_{bot1} , and HB_{bot2} in Fig. 3a). As shown in Fig. 3c, the asymmetric nature of hydrogen bonding (variance and difference between the hydrogen bonding length) was amplified when the lattice shrinkage occurs along the *c*-axis (region of negative uniaxial strain and positive biaxial strain as highlighted in Fig. 3c). The DFT calculation results support that the asymmetric behavior of hydrogen bonding between the chiral organic molecules and inorganic framework can be amplified depending on the degree of micro-strain, which can promote the efficient chirality transfer process by increasing the chiral distortion in inorganic framework.

The intensity of resulting CD from chiral 2D OIHs depends on the quadruple product as expressed by Eq. (3):

$$V_{12}[\vec{r}_{12} \cdot (\vec{\mu}_1 \times \vec{\mu}_2)] \quad (3)$$

where V_{12} is the half of the Davydov splitting value in Eq. (2) (See Supplementary Note 5 for detailed derivation procedure). With the consideration of Eq. (3), it is worth noting that the distance and angle between the MBA_1 and MBA_2 can determine the CD signal intensity and Davydov splitting. Therefore, we quantitatively estimated the effects of conformational π - π stacking variation on the CD signal intensity. Because the same precursor solutions were used when preparing the chiral 2D OIHs on different substrates (*i.e.*, the same magnitude of transition dipole moments μ_1 and μ_2 ; the same concentration of chiral organic spacer in chiral 2D OIHs lattice), the only difference arises from the distance and angle between the MBA_1 and MBA_2 . Based on our DFT calculations and Eq. (3), we can approximately evaluate the theoretical values of CD intensity change ($\Delta I_{CD}^{\text{theory}} = I_{CD}^{\text{strain}} / I_{CD}^{\text{planar}}$). However, as shown in Supplementary Fig. S8, regardless of the degree of induced strain, $\Delta I_{CD}^{\text{theory}}$ was remarkably lower than the observed experimental value ($\Delta I_{CD}^{\text{experiment}}$) for AAO templated chiral 2D OIHs. This implies that the conformational difference associated with π - π stacking of chiral organic cations cannot fully explain the enhancement in the CD intensity of chiral 2D OIHs grown in AAO templates. Therefore, we should consider other possible chirality transfer mechanisms.

Very recently, Mitzi *et al.* discovered that the induced CD of chiral 2D OIHPs do not always accompany structural chirality transfer from organic to inorganic layers manifested by centrosymmetric breaking in inorganic frameworks.³⁶ Indeed, dipolar interactions between the polarizable π clouds of chiral spacer cations and inorganic layers are crucial for determining the associated electronic structure of chiral 2D OIHPs, thereby giving rise to chiroptical responses, such as CD and CPPL.

In this regard, we firstly calculated the electronic band structures for strain free chiral 2D OIHPs by DFT calculations. The frontier orbitals at the band edges are attributed to the inorganic sub-lattice, which is consistent with previously reported Pb-based OIHPs (Fig. 3a). The conduction band minimum (CBM) is mainly determined by the Pb 6p orbitals. The valence band maximum (VBM) reveals the hybridization of the I 5p and Pb 6s orbital states. The calculated band gap of MBA_2PbI_4 at its ground state is 2.19 eV (565 nm), which agrees well with the extinction band edge of 2.35 eV (527 nm). It was found that the band gap of MBA_2PbI_4 is reduced when the interlayer spacing between PbI_4 inorganic layers shrinks as shown in Supplementary Fig. S9. The orbital resolved band structure clearly shows that the band edges (VBM and CBM) of MBA_2PbI_4 originated from the PbI_4 inorganic layers (Fig. 3a), and the highest occupied molecular orbital (HOMO) and lowest unoccupied molecular orbital (LUMO) of the MBA molecules form a type I-like band alignment. In addition, to assess the effectiveness of the multi-polar interactions between chiral organic spacers and inorganic frameworks, we further performed theoretical analysis on the distributions of the HOMO of MBA cations in chiral 2D OIHPs. Interestingly, Fig. 3b suggests that the HOMO wave function of MBA cations forms a resonant state with inorganic PbI_4 frameworks rather than exhibiting a strong localization on MBA molecules. Since the HOMO of MBA cations are delocalized over the entire chiral 2D OIHPs unit cell, this theoretical analysis implies that the electronic interaction between the chiral spacer cations and inorganic frameworks could be modulated by changing the degree of micro-strain imposed on the PbI_4 framework.

To explicitly prove our hypothesis that multi-polar interactions between chiral spacer cations and inorganic frameworks could be a key for interpreting the chirality transfer phenomena and chiroptical activities of chiral 2D OIHPs, we calculated the associated changes in electronic band structures for confined grown chiral 2D OIHPs when the micro-strain is applied to the lattice of chiral 2D OIHPs. Remarkably, the HOMO level of MBA goes up toward the VBM of PbI_4 inorganic frameworks upon compressive strain imposed along the c -axis (see yellow regions in Fig. 3c and d). This observation may give rise to the enhanced

multi-polar interactions between MBA molecules and PbI_4 inorganic frameworks, as the electrons belonging to MBA and PbI_4 become closer not only in spatial distance but also in energy level. It is worth mentioning that $\text{MBA}_2\text{PbI}_4(1-x)\text{Br}_{4x}$ thin films exhibit sharp XRD diffraction peaks assignable to the (002) planes regardless of the growing substrates and bromide composition, indicating the highly preferred orientation along the c -axis (Fig. 1d and Supplementary Fig. S10). In addition, the observed peak shift in the iodide determinant phase (toward higher 2θ degree) suggests that confined growth induces lattice shrinkage along the c -axis. Consequently, we need to focus on the shrinkage range in the out-of-plane direction (*i.e.*, negative uniaxial strain and positive biaxial strain; yellow region in Fig. 3c and d) to properly interpret our DFT calculations. The calculated electronic band structures imply that the amplified chiroptical phenomena of AAO templated chiral 2D OIHPs can possibly originate from the enhanced multi-polar interaction between MBA cations (*i.e.*, MBA_1 - and MBA_2 -) and inorganic framework induced by lattice shrinkage along the c -axis.

(Fig. 3 was substituted with the calculation results of specific structural parameters)

Fig. 3. Structure properties of chiral 2D OIHPs and the influence of micro-strain on the specific structural parameters by theoretical calculation. **a**, Schematic illustration of intra-octahedron distortions and hydrogen bondings in DFT-optimized structure. **b**, bond length distortion (upper panel) and bond angle variance (lower panel) as a function of micro-strain. **c**, hydrogen bonding length between the four different distinguishable hydrogen bondings in the unit cell of MBA_2PbI_4 (denoted as $\text{HB}_{\text{top}1}$, $\text{HB}_{\text{top}2}$, $\text{HB}_{\text{bot}1}$, and $\text{HB}_{\text{bot}2}$) and inorganic framework under uniaxial strain (left panel) and biaxial strain (right panel). Highlighted yellow regions indicate the range where the lattice shrinkage occurs along the c -axis.

Comment 4:

The authors claim that the CD “intensity” is given by Eq. 3. I do not understand the basis for this statement. It is not explained or derived and I am not aware of a literature precedent.

Author’s Response:

We thank the reviewer for kind comment of calculation process and equation based on Frenkel exciton theory. We are sorry for the difficulty in understanding the meaning of Eq. 3 due to the insufficient explanation and derivation. Therefore, we provide the detail derivation procedure for CD intensity calculation in the revised manuscript.

Intensity of CD signal can be calculated by the rotational strength (R) from the experimental results given by Eq. 1

$$R_{ab} = c \int \frac{CD(\lambda)}{\lambda} d\lambda \quad \text{Eq. 1}$$

where R_{ab} represents the rotational strength related to the transition from state a to state b , c is constant given by $\frac{3hc}{8\pi^3N_0}$, and $CD(\lambda)$ is the circular dichroisms as a function of the wavelength.

Eq. 1 implies that the large intensity of CD can be attributed to the large rotational strength (R_{ab}). However, in the chiral 2D OIHPs, the interaction between the dipole moments (static dipole moment of chiral organic cation and transition dipole moment in inorganic framework) provides the most significant contributions to the CD spectra.[R25] As a result of the coupling between the dipole moments, the degenerated excited states split into two sub levels separated by a quantity $2V_{12}$ (Davydov splitting). The strength of the dipole interaction can be calculated by the Coulomb dipole–dipole given by Eq. 2:

$$V_{12} = \frac{\mu_1\mu_2}{r_{12}^3} [\vec{e}_1 \cdot \vec{e}_2 - 3(\vec{e}_1 \cdot \vec{e}_{12})(\vec{e}_2 \cdot \vec{e}_{12})] \quad \text{Eq. 2}$$

where μ_1 , μ_2 and r_{12} are the intensity of each transition dipole and distance between the two transition dipoles, while \vec{e}_i is the corresponding unit vector. Here, it is worth to note that the most significant case arises with strong electric-dipole allowed transitions (not magnetic-dipole allowed transition) couple to each other (exciton coupling) due to the huge difference of magnitude between the electric-dipole and magnetic-dipole (*i.e.*, $\mu_{ab} \gg m_{ab}$).[R25] Therefore, the corresponding rotational strength in chiral 2D OIHPs can be approximated by Eq. 3:

$$R_{1,2} \approx \pm \vec{r}_{1,2} \cdot \vec{\mu}_1 \times \vec{\mu}_2 \quad \text{Eq. 3}$$

Combining the Eq.2 and Eq.3, the resulting CD signal from the chiral 2D OIHPs, which is proportional to rotational strength of dipole interaction ($R_{1,2}$), can be estimated by the

following derivation procedure and Eq. 4:

$$\int \frac{CD(\lambda)}{\lambda} d\lambda \propto \Gamma(\lambda, \lambda_0) V_{12} \vec{r}_{1,2} \cdot \vec{\mu}_1 \times \vec{\mu}_2 \quad \text{Eq. 4}$$

where $\Gamma(\lambda, \lambda_0)$ represents the factor that accounts for the shape of CD signal.[R26] Consequently, based on the Eq. 4, the CD intensity of chiral 2D OIHPs resulting from the dipole interaction between the static dipole moment of chiral organic cation and transition dipole moment of inorganic framework depends on the triple product as expressed by $V_{12}[\vec{r}_{1,2} \cdot (\vec{\mu}_1 \times \vec{\mu}_2)]$ scaled by V_{12} . Because the statements related with the calculation of CD intensity were deleted in manuscript, no revision was made by this question

References cited in this response:

- R25 Aaron, F. *et al.* Induced Chirality in Halide Perovskite Clusters Through Surface Chemistry *J. Phys. Chem. Lett.* **13**, 686–693 (2022).
- R26 Berova, N., Di Bari, L. & Pescitelli, G. Application of Electronic Circular Dichroism in Configurational and Conformational Analysis of Organic Compounds. *Chem. Soc. Rev.* 2007, **36**, 914–931 (2007).

Comments on the presentation and additional technical issues:

In this section I comment on aspects of the manuscript presentation that I think ought to be corrected; as well as some other technical issues I noted while reading the manuscript.

Comment 1)

Given the importance of the nano-confined growth in nanopore substrates in this study, this rather clever technique ought to be mentioned at least in the abstract-- if not in the title; however, nothing is said about the technique until page 4 of the manuscript.

Comment 2)

The actual material system studied (chiral $\text{MBA}_2\text{PbI}_4(1-x)\text{Br}_4x$) is not stated until page 6 of the manuscript: It is not stated in the abstract or title; leading one to wonder what x in line 107 on page 4 refers to. The impression given in the title is that a very general analysis will be forth-coming; but that was not the case.

Comment 3)

Line 34 in the abstract is rather unclear: “(different absorption of 2.0×10^{-3} and distinct photoluminescence of 6.4×10^{-2} for left- and right-handed circularly polarized light)” There are terms for what the authors are trying to describe here: “dissymmetry” and “anisotropy factor”. I suggest these terms be used.

Comment 4)

Line 78 why “inconspicuous”?

Comment 5)

Line 79 “it is” -> “it was”

Comment 6)

Page 4 line 122: “intensities” is not a correct term to use here.

Author’s Response:

As per the reviewer’s comment, we have corrected it.

Revision made (colored in blue):

(Title of manuscript)

~~Multi Polar Interaction: The Origin of Chiroptical Activity in Chiral 2D Perovskites~~

Elucidating the origin of chiroptical activity in chiral 2D perovskites through nano-confined growth

(Abstract, line 4)

... Here, through the nano-confined growth of chiral perovskites ($\text{MBA}_2\text{PbI}_{4(1-x)}\text{Br}_{4x}$) ~~the in-depth investigation of chiroptical phenomena-based oscillator coupling theory~~ and theoretical calculations, ...

(Abstract, line 8–9)

... we observed remarkable asymmetry behavior (~~different absorption dissymmetry~~ of 2.0×10^{-3} and ~~distinct anisotropy factor~~ of photoluminescence of 6.4×10^{-2} for left- and right-handed circularly polarized light) in nanoconfined chiral perovskites even at room temperature. ...

(In page 3)

... (i.e., mechanism (iv), which is ~~inconspicuous and~~ less studied) should also be elaborately scrutinized. ...

(In page 3)

... Very recently, it ~~is was~~ demonstrated that a large π bond (Π_6^0) with delocalized electrons of the organic spacer could ...

(In page 5)

... where CD and extinction are ~~the intensities~~ obtained from the CD and absorption spectra, ...

Comment 7)

Page 6, line 148-152. I find it hard to accept the general statement made, when only one XRD peak is shown. Properly, an analysis would need to be made of the XRD patterns to actually determine the space groups before making such a broad claim: “This observation implies that unprecedented chiroptical phenomena of templated chiral 2D OIHPs cannot be explained in terms of the dichotomy between optically active chiral space group of $P212121$ (iodide-determinant phase) and nonchiral space group (bromidedeterminant phase)”

Author’s Response:

Based on the previous reports, for bromide ion, it is difficult to form 2D corner-sharing PbBr_6 layers like the structure in MBA_2PbI_4 . [R27,R28] Therefore, the reduced optical activity in

MBA₂Pb_{4x}Br_{4(1-x)} with higher bromide composition can be interpreted as a result of the decreased proportion of optically active phase (iodide-determinant phase) in thin films. In order to avoid misunderstanding, we correct the term for description of the bromide-determinant phase.

References cited in this response:

- R27 Dang YY, Liu XL, Sun YJ, Song JW, Hu WP, Tao XT. Bulk Chiral Halide Perovskite Single Crystals for Active Circular Dichroism and Circularly Polarized Luminescence. *J. Phys. Chem. Lett.* **11**, 1689-1696 (2020).
- R28 Lu Y, *et al.* Spin-Dependent Charge Transport in 1D Chiral Hybrid Lead-Bromide Perovskite with High Stability. *Adv. Funct. Mater.* **31**, 2104605 (2021).

Revision made (colored in blue):

(In page 7)

... This observation implies that unprecedented chiroptical phenomena of templated chiral 2D OIHPs cannot be explained in terms of the dichotomy between **optically active iodide-determinant phase (chiral space group of $P2_12_12_1$)** (~~iodide-determinant phase~~) and **optically non-active bromide-determinant phase (thermodynamically unfavorable phase)** ~~nonchiral space group (bromide-determinant phase)~~, which is based on the prevailing crystal structure-dependent chirality transfer mechanism.

Comment 8)

The analysis of micro-strain based on XRD linewidths, shown in Fig 2, is described in the SI in Supplementary Note 2. There the authors state that Scherrer broadening is neglected since chiral OIHPs “have grain size larger than a few hundreds of nanometers”. How is that statement relevant to chiral OIHP films grown in nanopore substrates with pore sizes less than ~100nm?

Author’s Response:

We thank the reviewer for kind comment about the validity of strain-analysis based on modified Williamson-Hall method. In our manuscript, we mentioned that the “Scherrer broadening would be only significant when the grains are in the nanoscale range. Therefore, we do not expect Scherrer broadening to be a significant contribution in chiral OIHPs that have a grain size larger than a few hundreds of nanometers.” As the reviewer pointed out, this description is invalid for the chiral OIHPs films grown in AAO templates with pore size of 66 nm. However, despite of the pore size (< 100 nm), there are several reasons for excluding the effect of Scherrer broadening when the micro-strain analysis is performed. At first, the Scherrer broadening has a significant effect only if the size of crystalline is the main contribution and all other possible causes for micro-strain are negligible.[R29,R30] However, in the case of chiral 2D OIHPs in AAO templates, the main source of peak broadening is micro-stress imposed by pore wall of AAO templates. Furthermore, although the lateral crystalline size of chiral 2D OIHPs in AAO templates with pore size of 66 nm is less than 100 nm due to the nanoconfinement, the vertical size of the crystal is larger than 100 nm (please see our Response to **Comment 1** for ***Reviewer 2***). Therefore, we can conclude that as $\Delta d_{strain} \gg \Delta d_{size}$, the micro-strain value of chiral 2D OIHPs imposed by nanoconfinement growth can be calculated by the following equation: $(\Delta d_{obs}^2 - \Delta d_{inst}^2)^{1/2} \approx \Delta d_{strain} = \varepsilon \cdot d$.

References cited in this response:

- R29 Parrott ES, Patel JB, Haghghirad AA, Snaith HJ, Johnston MB, Herz LM. Growth Modes and Quantum Confinement in Ultrathin Vapour-Deposited MAPbI(3) Films. *Nanoscale* **11**, 14276-14284 (2019).
- R30 Wang JTW, *et al.* Efficient Perovskite Solar Cells by Metal Ion Doping. *Energ. Environ. Sci.* **9**, 2892-2901 (2016).

Revision made (colored in blue):

(Supplementary Note 2)

... We note that Scherrer broadening would only be significant ~~when the grains are in the nanoscale range.~~ when the size of crystal is the main contribution of peak broadening and all other possible causes for micro-strain are negligible. In addition, as shown in Supplementary Fig. S2, the vertical size of the crystal is larger than 100 nm. Therefore, we do not expect Scherrer broadening to be a significant contribution in chiral OIHPs., ~~which have a grain size larger than a few hundreds of nanometers.~~ Micro-strain is the relative change in the size of materials with respect to its thermodynamic ideal size (or size before experiencing an external force). The micro-strain in a crystalline material is the result of small fluctuations in the lattice spacing induced by crystal imperfections, structural defects, including dislocations, vacancies, stacking faults, interstitials, twinning, and grain boundaries. **In the case of chiral 2D OIHPs in AAO templates, the main source of peak broadening is micro-stress imposed by pore wall of AAO templates.** ...

Comment 9)

Page 9 and Fig 2 show the “experimentally determined Davydov splitting”. But Fig 2c and Fig S7 are not properly labelled and are not properly explained.

Author’s Response:

Thanks for the comments. We corrected it in the revised manuscript as the reviewer suggested. However, we partially agree with the comment about the physical information from the CD and CPPL. The absorption process (CD) starts from a selected subset of vibrational states in ground state and proceeds to many vibrational states in an excited electronic state (thus containing much information on the excited electronic state's vibrational properties). In contrast, the emission process (CPPL) starts from a selected subset of vibrational states of the electronic excited state and proceeds to many possible vibrational states in the ground electron state's vibrational manifold (thus providing information on the ground state's vibrational structure). From this perspective, we have mentioned that “The CPPL spectra give useful information on the ground state of chiral 2D OIHPs, whereas CD spectroscopy can provide information regarding the electronic structure of the excited state of materials” in our manuscript. Of course, as the reviewer pointed out, the CPPL spectra can be highly influenced by the spin and energy relaxation process, making it difficult to interpret meaningfully. Therefore, to probe the electronic structure of chiral 2D OIHPs, it is necessary **to investigate these complementary spectroscopies (both CD and CPPL) rather than using each separately.**

Revision made (colored in blue):

(Fig.2c and caption of Fig. 2 was modified)

Fig. 2. Estimation of micro-strain in chiral 2D OIHPs and stacking conformation variation induced by micro-strain. **a**, Schematic illustration of unit cell of R-MBA₂PbI₄ and induced change of *d*-spacing between the (002) planes. **b**, Magnitude of calculated micro-strain values as a function of AAO template pore size. The inset represents the schematic illustration of nanoconfined growth of chiral 2D OIHPs in AAO template. The black arrow indicates the change in the lattice parameter along the *c*-axis. **c**, Deconvolution results obtained from the CD spectra. The solid purple line represents obtained CD spectra from the chiral 2D OIHPs. The red and blue dot-line indicate the absorption of LCP and RCP, respectively. **d**, Plot of Davydov splitting values versus AAO template pore size. Error bars indicate the standard deviation. Note that the 0 nm pore size condition represents chiral 2D OIHPs grown on planar substrate.

(Supplementary; Fig. S7 were modified, and related comments were added in caption)

Fig. S7. Evaluation of the Davydov splitting values from the deconvoluted CD spectra. Deconvolution results obtained from the CD spectra (upper panel) and corresponding extinction spectra (lower panel) of $R\text{-MBA}_2\text{PbI}_{4(1-x)}\text{Br}_{4x}$ chiral OIHPs ($x = 0.325$) grown on different substrate conditions; a, planar substrate, b, AAO template with a 66 nm pore size, c, AAO template with a 100 nm pore size, and d, AAO template with a 112 nm pore size. The solid purple line represents obtained CD spectra from the chiral 2D OIHPs. The red and blue dot-line indicate the absorption of LCP and RCP, respectively.

(In page 14)

The CPPL spectra give useful information on the ground state of chiral 2D OIHPs, whereas CD spectroscopy can provide information regarding the electronic structure of the excited state of materials.³⁴ However, as the CPPL spectra can be highly influenced by the spin and energy relaxation process, it is difficult to derive the information of electronic structure by itself alone. ~~Therefore, these complementary spectroscopies are powerful tools for probing the electronic structure of chiral 2D OIHPs.~~ Therefore, to probe the electronic structure of chiral 2D OIHPs, it is necessary to investigate these complementary spectroscopies (both CD and CPPL) rather than using each separately.

<Reviewer 4>

In the article "Multi-Polar Interaction: The Origin of Chiroptical Activity in Chiral 2D Perovskites" the authors investigated 2D hybrid organic-inorganic perovskites having chiroptical activity. In particular they focus on $\text{MBA}_2\text{PbI}_4(1-x)\text{Br}_4x$ systems. The study involves both experimental characterization and theoretical explanations, mainly based on Density Functional Theory (DFT) calculations.

Starting from experimental characterization of the optical activity due to the chirality of the structure, they proposed a theoretical interpretation in terms of an indirect interaction (there is no chemical bonding) between organic spacers (chiral molecular cations) and the framework. We discussions seem appropriate and supported both from experiments and calculations.

I believe that the article will be interesting for the large community working on hybrid perovskites, and certainly, 2D Chiral HIOPs are an hot topic in material science. Therefore, I recommend publication in Nat. Comm.

Remark:

We would like to thank the reviewer for evaluating our work as an interesting new idea to clarify the origin/mechanism of chiroptical activities in metal halide systems. Our response to the reviewer's comments can be found below.

Comment 1:

Formula (3), line 239, should not be called quadruple product.

A quadruple product involves 4 vectors, and it can be either scalar or vector quadruple product. In this case, V_{12} is simply a scalar quantity (Davydov splitting), which acts as a scale factor. Therefore, I would refer to a triple product (the quantity in square brackets), scaled by V_{12} .

Author's Response:

Thanks for the comments. Because the statements related with the calculation of CD intensity were deleted in manuscript, no revision was made by this question

Response to Technical Comments

Comments:

In addition to the above, you must comply with the following editorial requests; we will not be able to proceed with your revised manuscript otherwise. Please also see the Nature Communications formatting instructions, which you may find useful while preparing your revised manuscript.

Author's response

Thank you for the comment. The manuscript was revised according to the formatting instructions.

Revision made (colored in blue):

The manuscript was divided into several sections.

ABSTRACT / INTRODUCTION / RESULTS / DISCUSSION / METHODS

The **RESULTS** section was also divided into several subsections.

REVIEWER COMMENTS

Reviewer #1 (Remarks to the Author):

The authors addressed the comments to my satisfaction.
Good to go.

Reviewer #2 (Remarks to the Author):

The authors have answered most of my questions except my comment #5. The optical anisotropy I mentioned is from the chiral perovskite nanowires rather than AAO template. In this case, the authors are suggested to consider how the optical anisotropy of perovskite nanowires affects the CD signal. After the authors addressed this question, I would like to recommend its publication.

Reviewer #3 (Remarks to the Author):

Please see attached review file.

Manuscript number: Nature Communications manuscript NCOMMS-21-30935-A

Title: Elucidating the origin of chiroptical activity in chiral 2D perovskites through nano-confined growth

Authors: Sunihl Ma et al.

Synopsis of response to the first referee reports

1. The title and abstract and abstract of the revised submission are much more appropriate in the resubmitted version.
2. The revised discussion of the strain and strain analysis is helpful.
3. The addition of absorption spectra are useful and appreciated additions.
4. The Authors have significantly revised and clarified the discussion pertaining to the Davydov splitting which previously seemed to imply that the CD response at the exciton peak was due to the interaction of the MBA1 and MBA2 transition dipoles and the involvement of a resonant state involving the MBA HOMO and the PbI4 framework- this discussion has now been deleted and a revised explanation involving “organic-to-inorganic chirality transfer *via* hydrogen-bonding interaction changes between the chiral organic spacer and inorganic framework” has been put forward, which seems more plausible.
5. The additions made in response to Referee#2’s comment 5 & 7 regarding the possible role of optical anisotropy either associated with, or induced by, the AAO templates, and the need for measurements on racemic samples grown in AAO templates, comprise very important control experiments and are an important addition.
6. I thank the Authors for their explanation of Eq. 3 in the original submission, -- I now understand that this is an expression derived for chiral molecules with coupled

chromophores with electric dipole allowed/magnetic dipole forbidden transitions; with a twist in their relative orientation. Eq 3 and the discussion of it have been deleted in the resubmitted version. Nevertheless, their explanation greatly clarified (for me) the meaning and context of Fig 2c.

Some issues that the authors ought to address in my opinion:

1. In their response to Referee 2 comment 6, and to Referee 3 comment 3 the Authors state that “ the positive and negative area of Cotton effect cannot be completely symmetrical.”

The Authors have cited Berova, *Chem. Soc. Rev.* 2007, **36**, 914–931 (2007), R26 in their rebuttal letter. So, how does this statement square against the CD sum rule, as articulated by Berova? See Eq. 6 and the discussion of it in the Berova paper. Berova et al. state: “the integral of CD over the whole electromagnetic spectrum is zero” and go on to state: “According to this rule if we see a positive CD band in a spectral region, somewhere else in the spectrum we can expect on or more bands of negative sign” ...although it may be hidden in an overlooked region of the spectrum to paraphrase. That seems unlikely to be the case here.

2. Following on from the last point, “apparent CD” (as opposed to true CD) due to optical anisotropy has a mono-signate characteristic (see Salij et al., <https://doi.org/10.1021/jacs.1c06752>).

While the control experiments made in response to the comments of Referee #2 (measurements on AAO substrates and on racemic samples grown on AAO substrates) seem rather compelling, have the Authors tried the simple expedient of flipping the sample orientation in the CD spectrometer to confirm that the sign of the CD does not change upon flipping the sample? That would be the tell-tale test that what is being measured is true CD versus apparent CD induced by optical anisotropy. If this were my experiment, I would do this test.

3. Eq 3 has been deleted but it appears subsequent equations were not renumbered.
4. A comment on the use and definition of the term “multi-polar interaction” .

The Authors use the term “multi-polar interaction” several places in the resubmitted manuscript (line 354, line 428, line 431, line 440, Fig 4). This term pervaded the original submission starting with the title... But I don't see it defined.

It appears to me that the Authors largely replaced the term “multipolar interactions” with the term “asymmetric hydrogen-bonding interaction”. (...but did so only in roughly the first half of the manuscript). Is this what the authors mean by the term “multipolar interactions”?

For example, in the original submission, the abstract stated,

“Here, through the nano-confined growth of chiral perovskites (MBA2PbI4(1-x)Br4x), we verified that the **multi-polar interaction** between chiral molecular spacers and the inorganic framework plays a key role in promoting the chiroptical activity of chiral perovskites.”.

This sentence has been replaced in the resubmission by,

“Here, through the nano-confined growth of chiral perovskites (MBA2PbI4(1-x)Br4x), we verified that the **asymmetric hydrogen-bonding interaction** between chiral molecular spacers and the inorganic framework plays a key role in promoting the chiroptical activity of chiral perovskites”

Similar replacement in line 92, etc.

So: if the Authors intend that “multipolar interactions” means “asymmetric hydrogen-bonding interaction”, could the Authors simply state that this is the case? If *not*, could the Authors please clarify?

Then: What does the term “multi-polar interaction” have to do with the contents of Fig 4, which is titled: **Effects of multi-polar interaction on the electronic structure of chiral 2D OIHPs** . Fig 4 shows measured CPPL on chiral 2D OIHPs thin films with different growing substrates. It is an inference to connect that to “multi-polar interaction” since Fig 4 adds nothing to understanding these “multipolar interactions” per se.

Since the term “multi-polar interaction” is important enough to highlight it in the Discussion section and in Fig. 4: I suggest the Authors define precisely what they mean by the term. OR replace it if by this term they actually mean “asymmetric hydrogen-bonding interaction” as seems to be the case.

5. Regarding the discussion of the Davydov splitting Eq. 2 and Fig. 2c: This concept and discussion pertains to one particular mechanism of CD, pertaining to the interaction of independent chromophores whose excitonic transitions are purely electric dipole allowed, magnetic dipole forbidden, and that interact via the Davydov mechanism. This is as opposed, for example, to the Rosenfeld mechanism where the CD is proportional to $\frac{\mu \cdot m}{r^3}$ where μ is the electric dipole moment and m is the magnetic dipole moment.

While its applicability could be debated (since it was developed for Frenkel exciton systems, yet they are applying it in a system that supports delocalized Wannier type excitons)... the correlations shown in Fig 2 are certainly interesting/evocative and will provoke discussion.

It would be helpful to readers (like me) to cite some of the literature on the Davydov/exciton model--- since I didn't get the context originally without additional response from the authors and additional study of the literature.

Summary:

The resubmitted manuscript is greatly improved and contains quite interesting data and inferences. In my opinion the manuscript would be a good publication however the questions above ought to be addressed by the authors.

Reviewer #4 (Remarks to the Author):

The authors reported an joint experimental and theoretical study focussing on a booming subfield of hybrid perovskites, i.e. chiral perovskites.

The authors replied in great details and extensively to all the referees' comments and questions.

The article has been significantly improved.

There are reasons to believe that this article will be a reference study for future studies on the same topics.

I suggest publication in Nature Communications.

Response Letter

Journal: Nature Communication

Manuscript number: NCOMMS-21-30935A

Title: “Multi-Polar Interaction: The Origin of Chiroptical Activity in Chiral 2D Perovskites”

Author(s): Sunihl Ma¹, Young-Kwang Jung¹, Jihoon Ahn¹, Jihoon Kyhm², Jeiwan Tan,¹ Hyungsoo Lee,¹ Gyumin Jang,¹ Chan Uk Lee,¹ Aron Walsh^{1,3}, and Jooho Moon^{1,*}

<Reviewer 1>

The authors addressed the comments to my satisfaction.

Good to go.

Remark:

We would like to thank the reviewer for evaluating our work. We believe that the reviewer's comments highly improve the quality of our manuscript.

<Reviewer 2>

The authors have answered most of my questions except my comment #5. The optical anisotropy I mentioned is from the chiral perovskite nanowires rather than AAO template. In this case, the authors are suggested to consider how the optical anisotropy of perovskite nanowires affects the CD signal. After the authors addressed this question, I would like to recommend its publication.

Remark:

Our response to the reviewer's comments can be found below.

Comment 1:

My most concern is the optical anisotropy of the samples grown on AAO substrates. For those samples, large optical anisotropy could be expected, which would greatly affect the accuracy of the measured CD signals and CPL. If this occurs, any conclusion obtained in this manuscript needs to be reconsidered. Or how can the authors exclude the influence from the optical anisotropy of the samples?

Author's Response:

We appreciate the reviewer's valuable comment on the optical anisotropy which can be originated from the macroscopic anisotropy of perovskite grown in AAO templates. As the reviewer commented, the inherent macroscopic optical anisotropy of perovskite grown in AAO templates may interfere with the origin of chiroptical activity in chiral 2D OIHP. However, it is worth to note that the strain-imposed chiral 2D perovskites grew as an ellipsoid-shaped single crystal rather than nanowire as shown in Fig. R1 (reproduced from Fig. S2 in our revised manuscript). Although the macroscopic anisotropy of ellipsoid single crystal is not expected to significantly affect the accuracy of chiroptical measurement, circular dichroism (CD) measurement was carefully investigated to eliminate the possible interference induced by the macroscopic nature of grown chiral 2D perovskites. Very recently, Di Bari and co-workers have reported that several organic thin films with macroscopic anisotropy can exhibit unexpected CD signal with a strong dependence on the light propagation direction (angle of incident light during the chiroptical measurement).[R1,R2,R3] The observed optoelectronic behavior stems from the optical interference of thin film's linear birefringence (LB) and linear dichroism (LD) (hereafter LDLB effect), rather than excitonic effects.

When investigating the chiroptical activities of thin films with macroscopic anisotropy, we need to recall a basic concept of Mueller matrix analysis; because the observed CD signal (CD_{obs}) is the sum of various contributions, which can be represented by the equation (1):

$$CD_{\text{obs}} \approx CD_{\text{true}} + \frac{1}{2}(LD' \cdot LB - LD \cdot LB') \quad (1)$$

where the first term refers to genuine CD, while the second term accounts for LDLB effect contributions (the signal of which is taken along an arbitrary axis defined in the laboratory frame and where the prime indicates a 45° axis rotation).[R1] As previous studies have reported that significant contribution of LDLB effect can contaminate the true chiroptical response in crystalline samples with macroscopic anisotropy, we need to exclude the influence of LDLB contribution to explain the true effect of spatial confined growth on chiroptical activity of chiral 2D perovskites. Since the LDLB effect contribution inverts upon sample flipping (*i.e.*, flipping the sample by 180° with respect to the light propagation axis), the CD_{true} and LDLB contribution term can be obtained separately by taking semi-sum and semi-difference of the two CD spectra with different measurement direction, (*i.e.*, front and back).

$$CD_{\text{true}} = 0.5 \times (CD_{\text{obs, front}} + CD_{\text{obs, back}}) \quad (2)$$

$$LDLB = 0.5 \times (CD_{\text{obs, front}} - CD_{\text{obs, back}}) \quad (3)$$

To eliminate the undesirable contamination from the LDLB effect and clarify the effect of nanoconfined growth on chiroptical phenomena, we have additionally conducted the CD measurement on chiral 2D perovskite in AAO templates with 100 nm pore size by varying the incident light direction (*i.e.*, front and back). Interestingly, as shown in Fig. R2, both of chiral 2D perovskites thin films exhibited the huge increment in CD signal under the backward measurement condition, regardless of the grown substrate conditions (*e.g.*, planar and 100 nm-pore sized AAO template). Furthermore, we isolated CD_{true} and LDLB contribution by using equation (2) and (3), demonstrating that the effect of nanoconfined growth in AAO template (*i.e.*, huge amplification of CD signal) can be also clearly observed in CD_{true} spectra (Fig. R3b). As the CD_{true} and LDLB contributions for AAO sample have opposite sign near the first extinction band edge (around 475 nm), the effect of nanoconfined growth on chiroptical phenomena in our manuscript was underestimated rather than overestimated; the difference of apparent CD spectra (CD_{obs}) between the planar and AAO template conditions (Fig. R3a, which is reproduced from Fig. 1a in our manuscripts) is smaller than that of genuine CD signal (CD_{true} in Fig. R3b). This result demonstrates that the effect of the optical anisotropy resulted from the

macroscopic nature can be completely excluded from our experimental results, interpretation, and conclusion. Consequently, it can be concluded that correlation between the deviation of chiroptical activities and the resulting micro-strain induced by the nanoconfined growth, which is a core objective of this paper, is well supported by experimental results and strain analysis.

Fig. R1. The morphology of chiral 2D OIHPs grown inside AAO templates and high-resolution transmission electron microscopy (HRTEM) image of chiral 2D OIHPs confined in AAO templates with pore size of 66 nm condition.

Fig. R2. Optical activity of chiral 2D perovskite grown on different substrate conditions. CD spectra obtained by varying the light direction (forward and backward) **a**, planar substrate condition and **b**, AAO substrate condition.

Fig. R3. The observed and calculated CD and LDLB spectra of chiral 2D perovskite grown on different substrate conditions. **a**, Apparent CD signal, **b**, genuine CD signal, and **c**, LDLB effect contribution calculated from the light direction dependent CD measurement.

References cited in this response:

- R1 Salij A, Goldsmith RH, Tempelaar R. Theory of Apparent Circular Dichroism Reveals the Origin of Inverted and Noninverted Chiroptical Response under Sample Flipping. *J. Am. Chem. Soc.* **143**, 21519-21531 (2021).
- R2 Albano G, Salerno F, Portus L, Porzio W, Aronica LA, Di Bari L. Outstanding Chiroptical Features of Thin Films of Chiral Oligothiophenes. *Chemnanomat* **4**, 1059-1070 (2018).
- R3 Albano G, Lissia M, Pescitelli G, Aronica LA, Di Bari L. Chiroptical response inversion upon sample flipping in thin films of a chiral benzo[1,2-b:4,5-b']-dithiophene-based oligothiophene. *Mater. Chem. Front.* **1**, 2047-2056 (2017).

<Reviewer 3>

1. The title and abstract and abstract of the revised submission are much more appropriate in the resubmitted version.
2. The revised discussion of the strain and strain analysis is helpful.
3. The addition of absorption spectra are useful and appreciated additions.
4. The Authors have significantly revised and clarified the discussion pertaining to the Davydov splitting which previously seemed to imply that the CD response at the exciton peak was due to the interaction of the MBA1 and MBA2 transition dipoles and the involvement of a resonant state involving the MBA HOMO and the PBI4 framework- this discussion has now been deleted and a revised explanation involving organic-to-inorganic chirality transfer via hydrogen-bonding interaction changes between the chiral organic spacer and inorganic framework has been put forward, which seems more plausible.
5. The additions made in response to Referee#2's comment 5 & 7 regarding the possible role of optical anisotropy either associated with, or induced by, the AAO templates, and the need for measurements on racemic samples grown in AAO templates, comprise very important control experiments and are an important addition.
6. I thank the Authors for their explanation of Eq. 3 in the original submission, -- I now understand that this is an expression derived for chiral molecules with coupled chromophores with electric dipole allowed/magnetic dipole forbidden transitions; with a twist in their relative orientation. Eq 3 and the discussion of it have been deleted in the resubmitted version. Nevertheless, their explanation greatly clarified (for me) the meaning and context of Fig 2c.

Remark:

We would like to thank the reviewer for improving our work by suggesting an interesting new idea and possible explanation based on theoretical basis. Our response to the reviewer's comments can be found below.

Comment 1:

In their response to Referee 2 comment 6, and to Referee 3 comment 3 the Authors state that "the positive and negative area of Cotton effect cannot be completely symmetrical."

The Authors have cited Berova, Chem. Soc. Rev. 2007, 36, 914–931 (2007), R26 in their

rebuttal letter. So, how does this statement square against the CD sum rule, as articulated by Berova? See Eq. 6 and the discussion of it in the Berova paper. Berova et al. state: “the integral of CD over the whole electromagnetic spectrum is zero” and go on to state: “According to this rule if we see a positive CD band in a spectral region, somewhere else in the spectrum we can expect one or more bands of negative sign” ...although it may be hidden in an overlooked region of the spectrum to paraphrase. That seems unlikely to be the case here.

Author’s Response:

As the reviewer commented, according to the CD sum rule, if we see a positive CD band in a spectral region, somewhere else in the spectrum we can expect one or more bands of negative sign. Therefore, the discrepancy between the area of the positive and negative Cotton effect around the first extinction band edge in chiral 2D perovskite (around 475 nm) should be compensated for somewhere in the electromagnetic spectrum. As shown in Fig. R4, of course, the expanded CD spectra of chiral 2D perovskites showed additional CD signal and followed the CD sum rule as Berova mentioned. The additional CD signal results from the excitonic transition state of the perovskite, which is far from the MBA cation exciton transition (~ 260 nm). Since we mainly focused on the chiroptical phenomena near the first extinction band edge, where the intensive excitonic transition behavior occurs, and it is quite natural to satisfy the CD sum rule in electromagnetic spectrum, no revision was made by this question.

Fig. R4. Optical activity of chiral 2D perovskite grown on different substrate conditions. CD spectra near the **a**, first extinction band edge of chiral 2D OIHPs and **b**, expanded wavelength region.

Comment 2:

Following on from the last point, “apparent CD” (as opposed to true CD) due to optical anisotropy has a mono-signate characteristic (see Salij *et al.*). While the control experiments made in response to the comments of Referee #2 (measurements on AAO substrates and on racemic samples grown on AAO substrates) seem rather compelling, have the Authors tried the simple expedient of flipping the sample orientation in the CD spectrometer to confirm that the sign of the CD does not change upon flipping the sample? That would be the tell-tale test that what is being measured is true CD versus apparent CD induced by optical anisotropy. If this were my experiment, I would do this test.

Author’s Response:

We thank the reviewer for the critical comment on the interpretation and validity of CD spectra. As the reviewer suggested, to clarify the true CD associated with the excitonic transition and exclude the LDLB contribution term induced by macroscopic optical anisotropy, we have additionally conducted CD measurement by flipping the samples. Interestingly, as shown in Fig. R2, both of chiral 2D perovskites thin films exhibited the huge increment in CD signal under the backward measurement condition, regardless of the grown substrate conditions (*e.g.*, planar and 100 nm-pore sized AAO template) (please see our Response to **Comment 1** for **Reviewer 2**). Furthermore, in our chiral 2D perovskite grown in AAO templates, since the LDLB effect negatively contributed to the apparent CD, the calculated true CD spectra revealed that the effect of nanoconfined growth by AAO template on chiroptical activities was underestimated in the apparent CD spectra (Fig. 1 in our manuscript).

To quantitatively compare the contribution between CD_{true} and LDLB effect to the apparent CD signal,[R4] the absolute integral area value of each spectrum (CD_{true} and LDLB) was calculated, and the corresponding values were shown in Table R1. Although the LDLB effect contribution can be clearly recognized in chiral 2D perovskite grown in AAO templates, the integral ratio of LDLB/ CD_{true} (0.46 for planar and 0.47 for AAO template) confirmed that CD_{true} term has more contribution to apparent CD spectra than LDLB effect. In addition, the integral ratio of two samples was very similar, implying that observed LDLB effect stems from the preferential orientation of chiral 2D perovskite itself (*i.e.*, preferred orientation of (002 l) plane) rather than optical anisotropic nature of grown substrates. We appreciate the reviewer for suggesting the possible direction of the future work on exploiting the LDLB signal and genuine light-mater interaction. Please understand that the results of chiroptical measurement,

including the sample flipping experiment, are not added in the revised manuscript, because the origin of existing LDLB effect and contribution to apparent CD observed in chiral 2D perovskite will be an interesting future research topic.

Sample Condition	Absolute integral of CD _{true}	Absolute integral of LDLB	Area ratio of LDLB/CD _{true}
Planar	264.80	121.72	0.46
AAO	1104.93	524.00	0.47

Table R1. Integral areas of the absolute values of CD_{true} and LDLB effect for chiral 2D perovskite samples grown in different substrate conditions, which is calculated between 425 nm and 525 nm.

References cited in this response:

R4 Albano G, Salerno F, Portus L, Porzio W, Aronica LA, Di Bari L. Outstanding Chiroptical Features of Thin Films of Chiral Oligothiophenes. *Chemnanomat* **4**, 1059-1070 (2018).

Comment 3:

Eq 3 has been deleted but it appears subsequent equations were not renumbered.

Author's Response:

As the reviewer's pointed out, subsequent equations were renumbered. Since the equation, which is related with the approximation of rotational strength, was added as equation (3), the number of following definition about asymmetry factor of CPPL was not changed.

Revision made (colored in blue):

(in page 13)

... following definition:

$$g_{\text{CPPL}} = \frac{I_L - I_R}{\frac{1}{2}(I_L + I_R)} \quad (4)$$

where I_L and I_R are the intensities of LCP and RCP light photoluminescence, respectively. ...

Comment 4:

A comment on the use and definition of the term “multi-polar interaction”. The Authors use the term “multi-polar interaction” several places in the resubmitted manuscript (line 354, line 428, line 431, line 440, Fig 4). This term pervaded the original submission starting with the title... But I don't see it defined. It appears to me that the Authors largely replaced the term “multipolar interactions” with the term “asymmetric hydrogen-bonding interaction”. (...but did so only in roughly the first half of the manuscript). Is this what the authors mean by the term “multipolar interactions”?

Author's Response:

Thank you for the comment. When we used the term of *multi-polar interaction*, it was intended to mention the type of electronic interaction between the two building blocks (*i.e.*, chiral molecules and achiral inorganic frameworks). However, in our resubmitted manuscript, it is clear that the origin of chiroptical behavior in chiral 2D perovskites is related with the asymmetric hydrogen-bonding interaction rather than multi-polar interaction (or resonant electronic interaction between the chiral molecules and inorganic framework). Therefore, in order to remove the possibility of confusion, we have replaced the term of multi-polar interaction with the term of *asymmetric hydrogen-bonding interaction*.

Revision made (colored in blue):

(in page 14)

... to investigate the effects of ~~multi-polar interaction~~ asymmetric hydrogen-bonding induced by confined growth on the electronic structure of chiral 2D OIHPs ...

(in page 17)

Consequently, it can be concluded that the amplified chiroptical activity in AAO templated chiral 2D OIHPs is more affected by the ~~multi-polar interaction~~ asymmetric hydrogen-bonding interaction between MBA cations and the inorganic framework rather than the structural distortion in the inorganic framework itself. Our findings suggest that the degree of chirality transfer (efficient chirality transfer phenomena) from chiral organic spacers to the achiral inorganic framework can be facilitated by enhancing the asymmetric nature of hydrogen-bonding interaction between chiral organic molecules and inorganic frameworks. ~~interpreted as a multi-polar interaction.~~ ...

(in page 18)

... The induced conformational stacking variation promoted **the asymmetric hydrogen-bonding interaction ~~multi-polar interaction~~** between the chiral organic cations and inorganic framework,...

(Caption of Fig.4)

Fig. 4. Effects of asymmetric hydrogen-bonding interaction ~~multi-polar interaction~~ on the electronic structure of chiral 2D OIHPs. **a**, Schematic illustration of optical setup for measuring the CPPL of chiral 2D OIHPs thin films. **b**, Magnified CPPL spectra (near FE emission wavelength region) of $R\text{-MBA}_2\text{PbI}_{4(1-x)}\text{Br}_{4x}$ chiral 2D OIHPs ($x = 0.325$) grown in the AAO template with a 100 nm pore size. Full range of CPPL spectra (including FE and STE emission wavelength region) of $R\text{-MBA}_2\text{PbI}_{4(1-x)}\text{Br}_{4x}$ chiral 2D OIHPs ($x = 0.325$) grown on **c**, planar substrate condition, and **d**, AAO with 100 nm pore size.

Comment 5:

Regarding the discussion of the Davydov splitting Eq. 2 and Fig. 2c: This concept and discussion pertains to one particular mechanism of CD, pertaining to the interaction of independent chromophores whose excitonic transitions are purely electric dipole allowed, magnetic dipole forbidden, and that interact via the Davydov mechanism. This is as opposed, for example, to the Rosenfeld mechanism where the CD is proportional to $\mu \cdot m$ where μ is the electric dipole moment and m is the magnetic dipole moment. While its applicability could be debated (since it was developed for Frenkel exciton systems, yet they are applying it in a system that supports delocalized Wannier type excitons)... the correlations shown in Fig 2 are certainly interesting/evocative and will provoke discussion. It would be helpful to readers (like me) to cite some of the literature on the Davydov/exciton model--- since I didn't get the context originally without additional response from the authors and additional study of the literature.

Author's Response:

Thank you for the comment. As reviewer pointed out, Davydov model considers an ideal system, which consist of independent chromophores with degenerated excited states. When those independent chromophores are spatially close enough to one another, the interaction between the transition moments of the component chromophores occurs, resulting in coupling of the chromophores.[R5] Therefore, we agreed with the comment that applicability of Davydov model in chiral perovskite system with Wannier type excitons could be controversial. In order to diminish the possibility of misunderstanding and to secure the versatility of interpretations applicable to various system with optical activities, we have substituted the term of "Davydov splitting" with the more general term of "excited state splitting".[R6] By applying perturbation theory, the originally degenerated excited level splits into the two states separated by $2V_{12}$, which is now referred to as the excited state splitting. In this respect, the intensity of characteristic CD signal is proportional to the rotational strength (R) by the following equation (4), which considers the interaction between the electric dipole moments μ_1 and μ_2 and also includes the terms relating to the coupling of μ and m by Rosenfield mechanism.[R7,R8]

$$R \propto r_{1,2} \mu_1 \times \mu_2 + \text{Im}\{(\mu_1 \mp \mu_2) \cdot (m_1 \mp m_2)\} \quad (4)$$

We modified the expression, which is related to Davydov splitting and some of literature about the interpretation of chiroptical activities in exciton model were added in revised manuscript as reviewer suggested.

References cited in this response:

- R5 Greenfield JL, Wade J, Brandt JR, Shi XY, Penfold TJ, Fuchter MJ. Pathways to increase the dissymmetry in the interaction of chiral light and chiral molecules. *Chem Sci.* **12**, 8589 (2021).
- R6 Castro-Fernandez S, Pena-Gallego A, Mosquera RA, Alonso-Gomez JL. Chiroptical Symmetry Analysis: Exciton Chirality-Based Formulae to Understand the Chiroptical Responses of C-n and D-n Symmetric Systems. *Molecules* **24**, 141 (2019).
- R7 Berova N. *et al.* *Comprehensive chiroptical spectroscopy* (Wiley, Hoboken, 2012).
- R8 Bruhn T, *et al.* Axially Chiral BODIPY DYEmers: An Apparent Exception to the Exciton Chirality Rule. *Angew. Chem. Int. Edit.* **53**, 14592-14595 (2014).

Revision made (colored in blue):

(in page 11)

... If two transition dipole moments are located close enough in space but are not coplanar, the coupling between two exciton transitions generate the splitting of excited states into two levels separated by $2V_{12}$, ~~also called Davydov splitting~~ which is referred to as the excited state splitting.^{43,44} ...

(in page 11)

... while \vec{e}_i is the corresponding unit vector. In this respect, the intensity of characteristic CD signal is proportional to the rotational strength (R) by the following equation (4), which considers the interaction between the electric transition dipole moments (μ_1 and μ_2) and also includes the terms relating to the coupling of electric transition dipole moment (μ) and magnetic transition dipole moment (m) by Rosenfield mechanism.^{45,46}

$$R \propto r_{1,2} \mu_1 \times \mu_2 + \text{Im}\{(\mu_1 \mp \mu_2) \cdot (m_1 \mp m_2)\} \quad (3)$$

It is worth to noting that the abnormal chiroptical behaviors ...

(in page 12)

To verify the relationship between the imposed micro-strain and the efficiency of chirality transfer in chiral 2D OIHPs, we evaluated the ~~excited state splitting~~ ~~Davydov splitting~~ values from the deconvoluted CD spectra where a bisignate CD signal appeared around the extinction band edge λ_0 . Using a multiple-peak fitting function, several peaks in chiral 2D OIHPs with various halide composition were identifiable (Supplementary Fig. S12; see Supplementary Note 4 for the detailed procedures and validity). The experimentally determined ~~Davydov splitting~~ ~~excited state splitting~~ values for chiral 2D OIHPs, $2V_{12}$, are plotted in Fig. 2d as a function of the pore size of the AAO template (pore size = 0 for the planar substrate). Interestingly, in the entire composition range (from $x = 0.325$ to $x = 0.400$), the corresponding variation in ~~Davydov splitting~~ ~~excited state splitting~~ also exhibited a zigzag tendency, which is similar to the calculated micro-strain results (Fig. 2b) and PL emission shift (Supplementary Fig. S8b). Such a coincident tendency (*i.e.*, similar zigzag tendency in calculated micro-strain, PL emission shift as well as ~~Davydov splitting~~ ~~excited state splitting~~) can support...

(Fig.2 and Caption of Fig.2 were changed)

Fig. 2. Estimation of micro-strain in chiral 2D OIHPs and stacking conformation variation induced by micro-strain. **a**, Schematic illustration of unit cell of R-MBA₂PbI₄ and induced change of *d*-spacing between the (002) planes. **b**, Magnitude of calculated micro-strain values as a function of AAO template pore size. The inset represents the schematic illustration of nanoconfined growth of chiral 2D OIHPs in AAO template. The black arrow indicates the change in the lattice parameter along the c-axis. **c**, Deconvolution results obtained from the CD spectra. The solid purple line represents obtained CD spectra from the chiral 2D OIHPs. The red and blue dot-line indicate the absorption of LCP and RCP, respectively. **d**, Plot of Davydov splitting excited state splitting values versus AAO template pore size. Error bars indicate the standard deviation. Note that the 0 nm pore size condition represents chiral 2D OIHPs grown on planar substrate without template.

(Caption of Fig. S12 were changed)

Fig. S12. Evaluation of the excited state splitting ~~Davydov-splitting~~ values from the deconvoluted CD spectra. Deconvolution results obtained from the CD spectra (upper panel) and corresponding extinction spectra (lower panel) of *R*-MBA₂PbI_{4(1-x)}Br_{4x} chiral OIHPs (x = 0.325) grown on different substrate conditions; a, planar substrate, b, AAO template with a 66 nm pore size, c, AAO template with a 100 nm pore size, and d, AAO template with a 112 nm pore size. The solid purple line represents obtained CD spectra from the chiral 2D OIHPs. The red and blue dot-line indicate the absorption of LCP and RCP, respectively.

(References were added)

- 43 Greenfield JL, Wade J, Brandt JR, Shi XY, Penfold TJ, Fuchter MJ. Pathways to increase the dissymmetry in the interaction of chiral light and chiral molecules. *Chem Sci.* **12**, 8589 (2021).
- 44 Castro-Fernandez S, Pena-Gallego A, Mosquera RA, Alonso-Gomez JL. Chiroptical Symmetry Analysis: Exciton Chirality-Based Formulae to Understand the Chiroptical Responses of C-n and D-n Symmetric Systems. *Molecules* **24**, 141 (2019).
- 45 Berova N. *et al.* *Comprehensive chiroptical spectroscopy* (Wiley, Hoboken, 2012).
- 46 Bruhn T, *et al.* Axially Chiral BODIPY DYEmers: An Apparent Exception to the Exciton Chirality Rule. *Angew. Chem. Int. Edit.* **53**, 14592-14595 (2014).

<Reviewer 4>

The authors reported an joint experimental and theoretical study focussing on a booming subfield of hybrid perovskites, i.e. chiral perovskites. The authors replied in great details and extensively to all the referees' comments and questions. The article has been significantly improved. There are reasons to believe that this article will be a reference study for future studies on the same topics. I suggest publication in Nature Communications.

Remark:

We would like to thank the reviewer for evaluating our work as a reference study for future research on the same topics.

REVIEWERS' COMMENTS

Reviewer #2 (Remarks to the Author):

I have no further question and would like to recommend its publication at the current form.

Reviewer #3 (Remarks to the Author):

Please see attached PDF containing my comments.

Manuscript number: Nature Communications manuscript NCOMMS-21-30935B

Title: Elucidating the origin of chiroptical activity in chiral 2D perovskites through nano-confined growth

Authors: Sunihl Ma et al.

Synopsis:

I have reviewed Authors' response, re-read the previous correspondence and read the new revised manuscript. The main features of the work are, to summarize,

1. Mixed anion chiral 2D perovskite samples are grown at a composition which is on the edge of a phase transition between a chiral phase (iodine rich) and a nonchiral phase (Br rich) (J. Am. Chem. Soc. 2020, 142);
2. Growth of this material in AAO substrates results in oriented nanoconfined growth of single domain nanocrystals that are in a state of negative uniaxial strain and positive biaxial strain;
3. The degree of micro-strain in the samples appears to correlate with the excited state splitting when the bi-signate CD spectra are interpreted in terms of a molecular Frenkel exciton model (Fig 2);
4. The AAO samples exhibit rather high degree of circularly polarized PL (Fig 4)
5. DFT calculations are performed which suggest the compressive micro-strain is connected with intra-octahedral distortions perhaps due to changes in hydrogen bonding, pointing to a mechanism for strain enhancement of chirality transfer from the chiral organic cations to the inorganic framework.

This is all good, and in my opinion will be really interesting for the community.

A couple issues:

- 1) Whether to include the apparent-CD data that came out in the back and forth over the referee comments.

In the last round of review I had suggested (following on to a question about optical anisotropy first posed by Referee #2) that the authors examine the possibility that the

measured CD signal in their chiral perovskite samples grown in AAO templates may be due to the phenomenon of “apparent CD”- which can be easily tested by the simple expedient of comparing the observed CD response measuring the sample from the front side versus the backside directions. If true CD is being measured, the spectra should be identical whether measured forward or backward. If the sign of the observed CD changes, then there is an “apparent CD” response which originates from an interference of linear dichroism and linear birefringence (the so called “LD/LB” signal).

The authors performed this test and reported back the results in Fig R2 and R3. The results were surprising: The CD amplitude measured in the backwards direction is roughly 6 times larger than measured in the forward direction (Fig R2). The authors define “true CD” and non-CD “LD/LB” spectra as the sum and difference of the forward and backward measured spectra (this is the common way this would be done) and then show bi-signate LD/LB spectra which are opposite in sign to the bi-signate “true CD” response measured in the forward direction.

They argue that the implication is therefore that the CD enhancement due to growth in micropores was previously under-reported, and that since the true CD response is greater than 50% of the signal (Table R1 gives the integrated area of the LB/LB spectra as 47%), none of the conclusions really change. They then state that they did not add these “apparent CD” data to the revised manuscript because this will be a subject of future research, writing:

“We appreciate the reviewer for suggesting the possible direction of the future work on exploiting the LDLB signal and genuine light-mater interaction. **Please understand that the results of chiroptical measurement including the sample flipping experiment, are not added in the revised manuscript**, because the origin of existing LDLB effect and contribution to apparent CD observed in chiral 2D perovskite will be an interesting future research topic.”

In my opinion, this data should be included. Just my opinion but, the question addressed in this manuscript is basically: “what is the mechanism for the enhanced CD response observed in the chiral perovskites grown in nanopore templates, and what, if anything does this contribute to the general understanding of the chiroptic response at the exciton line in chiral 2D perovskite semiconductors, which is so far essentially not understood?”. The directional dependence of the enhanced response is unexpected and is totally relevant to the question. It may be awkward to include this data, because it is hard to explain what the authors observed (i.e. its not understood at this point) but it is certainly relevant to complete picture of/understanding of the work.

2. General clarity: I suggest that the authors go through the paper once more and make that everything is described in consistent and clear terms since the theoretical model and some of explanations have evolved quite a bit in the course of the referee comments & discussion... The issue I raised about “multipolar interactions” vs hydrogen bonding interactions is an example

which the authors addressed in this new revision at my request. There are a few other statements that have crept in that I think are rather unclear or questionable, such as, on page 16,

To exclude the effect of Rashba splitting on CPPL spectra, which might arise from the experimental procedure (because the excitation by the circularly polarized light can lift the degeneracy of spin state due to the large spin-orbit coupling (SOC) of OIHPs),

And, coherent lifting of the spin degeneracy by the circularly polarized light.

These statements need references as it is not at all clear (to me) what they are taking about; are the authors referring to the work by Adarsh or ? Conventional 2D-Rashba splitting in 2D systems produces spin textures that are not compatible with chiroptic effects, so such statements should be clarified with additional prose or references that indicate what they mean.

Response Letter

Journal: Nature Communication

Manuscript number: NCOMMS-21-30935B

Title: “Multi-Polar Interaction: The Origin of Chiroptical Activity in Chiral 2D Perovskites”

Author(s): Sunihl Ma¹, Young-Kwang Jung¹, Jihoon Ahn¹, Jihoon Kyhm², Jeiwan Tan,¹ Hyungsoo Lee,¹ Gyumin Jang,¹ Chan Uk Lee,¹ Aron Walsh^{1,3}, and Jooho Moon^{1,*}

<Reviewer 3>

Comment 1:

Whether to include the apparent-CD data that came out in the back and forth over the referee comments. In my opinion, this data should be included. Just my opinion but, the question addressed in this manuscript is basically: “what is the mechanism for the enhanced CD response observed in the chiral perovskites grown in nanopore templates, and what, if anything does this contribute to the general understanding of the chiroptic response at the exciton line in chiral 2D perovskite semiconductors, which is so far essentially not understood?”. The directional dependence of the enhanced response is unexpected and is totally relevant to the question. It may be awkward to include this data, because it is hard to explain what the authors observed (i.e. its not understood at this point) but it is certainly relevant to complete picture of/understanding of the work.

Author’s Response:

We appreciate the reviewer’s valuable comment on the apparent-CD data observed in the chiral OIHPs grown inside the AAO templates. Although it is difficult to explain the directional dependence of the enhanced chiroptical activity in chiral OIHPs grown in the AAO templates, we decided to provide the apparent CD data and related explanations in our revised manuscript as the reviewer suggested. We would like to thank the reviewer for improving our work.

Revision made (colored in blue):

(in page 6)

… This implies that the effect of the optical anisotropy from the bare AAO templates can be completely excluded.

Recently, Di Bari and co-workers have reported that several organic thin films with macroscopic anisotropy can exhibit unexpected CD signal with a strong dependence on the light propagation direction (angle of incident light during the chiroptical measurement),^{36,37} which stems from the optical interference of thin film's linear birefringence (LB) and linear dichroism (LD) (hereafter LDLB effect) rather than excitonic effects. Therefore, when investigating the chiroptical activities of thin films with macroscopic anisotropy, we need to consider a basic concept of Mueller matrix analysis; because the observed CD signal (CD_{obs}) is the sum of various contributions as represented by the equation (2):

$$CD_{obs} \approx CD_{true} + \frac{1}{2}(LD' \cdot LB - LD \cdot LB') \quad (2)$$

where the first term refers to genuine CD, while the second term accounts for LDLB effect contribution (the signal of which is taken along an arbitrary axis defined in the laboratory frame and where the prime indicates a 45° axis rotation). It is necessary to exclude the influence of LDLB contribution to explain the true effect of spatial confined growth on chiroptical activity of chiral 2D perovskites. Since the LDLB effect contribution is inverted upon sample flipping (i.e., flipping the sample by 180° with respect to the light propagation axis), the CD_{true} term can be separately obtained by taking semi-sum of the two CD spectra with different measurement directions, (i.e., front and back).

$$CD_{true} = 0.5 \times (CD_{obs, front} + CD_{obs, back}) \quad (3)$$

The effect of nanoconfined growth in AAO template (i.e., huge amplification of CD signal) can be clearly observed in CD_{true} spectra (Supplementary Fig. S4b), where the effect of the optical anisotropy resulted from the macroscopic nature is completely excluded. Consequently, it can be concluded that the observed chiroptical activities in the AAO templated chiral 2D OIHs (e.g., huge amplification of the absolute g_{CD} value, sign conversion, and spectral shape change of Cotton effect) are attributed to effect of spatial confined growth of chiral 2D OIHs rather than optical anisotropy from the bare AAO templates and macroscopic anisotropy of chiral 2D OIHs. ~~Therefore, the observed sign conversion and spectral shape change of the Cotton effect imply that spatial confinement by AAO templates might induce the coupling~~

modulation between the chiral chromophores, resulting in the reconstruction of the chiroptical band structure.

(Apparent-CD measurement data were added in Supplementary as Fig. S4)

Fig. S4. The observed and calculated CD and LDLB spectra of chiral 2D perovskite grown on different substrate conditions. **a**, Apparent CD signal, **b**, genuine CD signal, and **c**, LDLB effect contribution calculated from the light direction dependent CD measurement.

(Related References were added)

- 36 Albano G, Salerno F, Portus L, Porzio W, Aronica LA, Di Bari L. Outstanding Chiroptical Features of Thin Films of Chiral Oligothiophenes. *Chemnanomat* **4**, 1059-1070 (2018).
- 37 Albano G, Lissia M, Pescitelli G, Aronica LA, Di Bari L. Chiroptical response inversion upon sample flipping in thin films of a chiral benzo[1,2-b:4,5-b']-dithiophene-based oligothiophene. *Mater. Chem. Front.* **1**, 2047-2056 (2017).

Comment 2:

General clarity: I suggest that the authors go through the paper once more and make that everything is described in consistent and clear terms since the theoretical model and some of explanations have evolved quite a bit in the course of the referee comments & discussion... The issue I raised about “multipolar interactions” vs hydrogen bonding interactions is an example which the authors addressed in this new revision at my request. There are a few other statements that have crept in that I think are rather unclear or questionable, such as, on page 16,

To exclude the effect of Rashba splitting on CPPL spectra, which might arise from the experimental procedure (because the excitation by the circularly polarized light can lift the degeneracy of spin state due to the large spin-orbit coupling (SOC) of OIHPs),

And,

coherent lifting of the spin degeneracy by the circularly polarized light.

These statement need references as it is not at all clear (to me) what they are taking about; are the authors are referring to the work by Adarsh or ? Conventional 2D-Rashba splitting in 2D systems produces spin textures that are not compatible with chiroptic effects, so such statements should be clarified with additional prose or references that indicate what they mean.

Author’s Response:

As the reviewer commented, we are sorry that there are a few unclear statements about the Rashba splitting and CPPL measurement conditions. Due to the heavy atoms in OIHPs, large spin-orbit coupling (SOC) of OIHPs can lift the degeneracy of spin state and lead to large Rashba splitting if the structure lacks inversion symmetry.[R1,R2] The effect of SOC may be enhanced in reduced dimension OIHPs including chiral 2D OIHPs with MQW structures.[R3] Furthermore, in the presence of applied magnetic field (about 1T – 5T), the circularly polarized light (CPPL) can be observed even in racemic compounds and 3D OIHPs without chirality transfer phenomena due to the field-induced population changes among the spin sublevels.[R4,R5] Compare to the unpolarized light, which consist of many electromagnetic waves polarized in different directions (*i.e.*, net electric and magnetic field are zero), circularly

polarized light (CPL) is polarized only in one direction by passing through the polarizing filter, so that both of the electric and magnetic field exist. Although the effective magnitude of external magnetic field for magneto-CPPL is quite large (as aforementioned; 1T – 5T) compared to the magnetic field in CPL, the excitation by CPL source can give rise to CPPL due to the field-effect rather chirality transfer phenomena. Therefore, to exclude all the potential possibilities of field-effect induced CPPL, which might arise from the CPL excitation source, the CPPL measurement was also performed in the same manner (using circular polarized light as an excitation source) for racemic compounds grown inside AAO templates with pore size of 100 nm as presented in Fig. S14 (Supplementary). In order to diminish the possibility of misunderstanding and to secure the reliability of interpretations on CPPL measurement using CPL excitation source, we have modified relevant statements and added additional reference for clarity as reviewer suggested.

References cited in this response:

- R1 Niesner D, *et al.* Structural fluctuations cause spin-split states in tetragonal (CH₃NH₃)PbI₃ as evidenced by the circular photogalvanic effect. *Proc. Natl. Acad. Sci. USA* **115**, 9509-9514 (2018).
- R2 Poonia AK, Shrivastava M, Mir WJ, Aneesh J, Nag A, Adarsh KV. Intervalley polaronic biexcitons in metal halide perovskite quantum dots. *Phys. Rev. B* **104**, L161407 (2021).
- R3 Zhai YX, *et al.* Giant Rashba splitting in 2D organic-inorganic halide perovskites measured by transient spectroscopies. *Sci. Adv.* **3**, e1700704 (2017).
- R4 Long GK, *et al.* Spin control in reduced-dimensional chiral perovskites. *Nat. Photonics* **12**, 528-533 (2018).
- R5 Zhang C, *et al.* Magnetic field effects in hybrid perovskite devices. *Nat. Phys.* **11**, 428-435 (2015).

Revision made (colored in blue):

(in page 16)

Due to the heavy atoms in OIHPs, large spin-orbit coupling (SOC) of OIHPs can lift the degeneracy of spin state and lead to large Rashba splitting if the structure lacks inversion symmetry.^{50,51} Furthermore, in the presence of applied magnetic field (about 1T – 5T), the CPPL can be observed even in racemic compounds and 3D OIHPs without chirality transfer phenomena due to the field-induced population changes among the spin sublevels.^{18,21} Compare to the unpolarized light, which consist of many electromagnetic waves polarized in different directions (i.e., net electric and magnetic field are zero), CPL is polarized only in one direction by passing through the polarizing filter, so that both of the electric and magnetic field exist. Although the effective magnitude of external magnetic field for magneto-CPPL is quite large (as aforementioned; 1T – 5T) compared to the magnetic field in CPL, the excitation by CPL source can give rise to CPPL due to the field-effect rather chirality transfer phenomena. To exclude the effect of Rashba splitting due to large SOC of OIHPs on CPPL spectra, which might arise from the experimental procedure (because of the magnetic field in CPL excitation by the circularly polarized light can lift the degeneracy of spin state due to the large spin-orbit coupling (SOC) of OIHPs), and to clarify the origin of different emission rate of RCP and LCP, the CPPL measurement was also performed in the same manner (using circular polarized light as a excitation source) for racemic compounds grown on AAO templates with pore size of 100 nm. As shown in Fig. S15 (Supplementary), the racemic compounds grown in AAO templates do not show any different emission behavior between the RCP and LCP. The CPPL spectra of racemic compounds grown on AAO templates suggested that the Rashba effect and coherent lifting of the spin degeneracy induced by the SOC of OIHPs circularly polarized light did not occur in the absence of chirality transfer phenomena (i.e., in the absence of chiral organic molecules). This implies that enhanced asymmetric factor of CPPL (g_{CPPL}) in chiral 2D OIHPs results from the facilitated chirality transfer phenomena rather than Rashba effect itself (induced by the large SOC of OIHPs excitation using circular polarized light). Very recently, Mitzi group found that the Rashba-Dresselhaus spin-splitting is a consequence of the chirality transfer phenomena.⁵² ...

(Related References were added)

- 50 Niesner D, *et al.* Structural fluctuations cause spin-split states in tetragonal $(\text{CH}_3\text{NH}_3)\text{PbI}_3$ as evidenced by the circular photogalvanic effect. *Proc. Natl. Acad. Sci. USA* **115**, 9509-9514 (2018).
- 51 Zhai YX, *et al.* Giant Rashba splitting in 2D organic-inorganic halide perovskites measured by transient spectroscopies. *Sci. Adv.* **3**, e1700704 (2017).